

# The Met Office Unified Model Global Atmosphere 7.0/7.1 and JULES Global Land 7.0 configurations

David Walters[1], Anthony Baran[1], Ian Boutle[1], Malcolm Brooks[1], Paul Earnshaw[1], John Edwards[1], Kalli Furtado[1], Peter Hill[2], Adrian Lock[1], James Manners[1], Cyril Morcrette[1], Jane Mulcahy[1], Claudio Sanchez[1], Chris Smith[1], Rachel Stratton[1], Warren Tennant[1], Lorenzo Tomassini[1], Kwinten Van Weverberg[1], Simon Vosper[1], Martin Willett[1], Jo Browse[3], Andrew Bushell[1], Mohit Dalvi[1], Richard Essery[4], Nicola Gedney[5], Steven Hardiman[1], Ben Johnson[1], Colin Johnson[1], Andy Jones[1], Graham Mann[6,7], Sean Milton[1], Heather Rumbold[1], Alistair Sellar[1], Masashi Ujiie[8], Michael Whitall[1], Keith Williams[1], and Mohamed Zerroukat[1]

[1]Met Office, FitzRoy Road, Exeter, EX1 3PB, UK
[2]Department of Meteorology, University of Reading, Reading, RG6 6BB, UK
[3]Centre for Geography, Society and the Environment, University of Exeter, Penryn Campus, Cornwall, TR10 9EZ, UK
[4]School of Geosciences, University of Edinburgh, Edinburgh, EH8 9XP UK
[5]Met Office, Joint Centre for Hydrometeorological Research, Maclean Building, Wallingford, OX10 8BB, UK
[6]School of Earth and Environment, University of Leeds, Leeds, LS2 9JT, UK
[7]National Centre for Atmospheric Science, University of Leeds, Leeds, LS2 9JT, UK
[8]Numerical Prediction Division, Japan Meteorological Agency 1-3-4 Otemachi Chiyoda-ku Tokyo 100-8122, Japan

*Correspondence to:* David Walters (david.walters@metoffice.gov.uk)

**Abstract.** We describe Global Atmosphere 7.0 and Global Land 7.0 (GA7.0/GL7.0): the latest science configurations of the Met Office Unified Model and JULES land surface model developed for use across weather and climate timescales. GA7.0 and GL7.0 include incremental developments and targeted improvements that between them address four critical errors identified in previous configurations: excessive precipitation biases over India, warm/moist biases in the tropical tropopause layer, a source
5 of energy non-conservation in the advection scheme and excessive surface radiation biases over the Southern Ocean. They also include two new parametrisations, namely the UKCA Glomap-mode aerosol scheme and the JULES multi-layer snow scheme, which improve the fidelity of the simulation and were required for the coupled climate models and Earth system models that will use these configurations in submissions to CMIP6.

In addition, we describe the GA7.1 branch configuration, which reduces an overly negative anthropogenic aerosol effective
10 radiative forcing in GA7.0, whilst maintaining the quality of simulations of the present-day climate. GA7.1/GL7.0 will form the physical atmosphere/land component in the HadGEM3-GC3.1 and UKESM1 climate models.





## 1   Introduction

In this paper, we document the Global Atmosphere 7.0 configuration (GA7.0) of the Met Office Unified Model (UM, Brown et al., 2012) and the Global Land 7.0 configuration (GL7.0) of the JULES land surface model (Best et al., 2011; Clark et al., 2011). These are the latest iterations in the line of GA/GL configurations developed for use in global atmosphere/land and

coupled modelling systems across weather and climate time scales. This development is a continual process made up of small incremental changes to parameters and options within existing parametrisation schemes, the implementation of new schemes and options, and less frequent major changes to the structure of the model and the framework in which it is built. The Global Atmosphere 6.0 configuration (GA6.0, Walters et al., 2017) fell into the latter category, as it included a once-in-a-decade replacement of the model's dynamical core. To allow the configuration developers to concentrate on that change, the inclusion

of other changes was limited to those that were known to be necessary alongside the dynamical core, or to significantly improve system performance measures so as to make the dynamical core implementation easier. For this reason, GA7 sees the inclusion of a number of bottom-up developments to the atmospheric parametrisation schemes developed over several years that improve the fidelity and internal consistency of the model. These include an improved treatment of gaseous absorption in the radiation scheme, improvements to the treatment of warm rain and ice clouds and an improvement to the numerics in

the model's convection scheme. It also includes a number of top-down developments motivated by the findings of Process Evaluation Groups (PEGs), which are tasked with understanding the root causes of model error. These changes include further developments in the model's microphysics and incremental improvements to our implementation of the dynamical core. In combination with the bottom-up developments discussed previously, these lead to large reductions in our four critical model errors, namely rainfall deficits over India during the South Asian monsoon, temperature and humidity biases in the tropical

tropopause layer, deficiencies in the model's numerical conservation and surface flux biases over the Southern Ocean. Finally, GA7 and GL7 include new parametrisation schemes, which increase the complexity and fidelity of the model and introduce new functionality that was deemed necessary for the next generation climate modelling systems in which they will be used, and which will form the UK's contribution to the 6[th] Coupled Model Intercomparison Project (CMIP6, Eyring et al., 2015). These new capabilities include a multi-moment modal representation of prognostic tropospheric aerosols, a multi-layer snow

scheme and a seamless stochastic physics package, which will see the inclusion of stochastic physics terms in production UM climate simulations for the first time.

In Sect. 2 we describe GA7.0 and GL7.0, whilst in Sect. 3 we document how these differ from the last documented configurations: GA6.0 and GL6.0[1,2]. The development of these changes is documented using "trac" issue tracking software, so for consistency with that documentation, we list the trac ticket numbers (denoted by trac's # character) along with these descriptions.

Section 4 includes an assessment of the configuration's performance in global weather prediction and atmosphere/land-only climate simulations. This illustrates the reduction of the critical model errors noted above, and highlights some improvements

---

[1]Where the configurations remain unchanged from GA6.0/GL6.0 and their predecessors, Sect. 2 contains material which is unaltered from the documentation papers for those releases (i.e. Walters et al., 2011, 2014, 2017).

[2]In addition to the material herein, the Supplement to this paper includes a short list of model settings outside the GA/GL definition that are dependent on either model resolution or system application.





in simple weather prediction tests, but suggests that improvements are needed in the interaction between the model and its data assimilation before implementation for operational forecasting. In Sect. 5 we briefly describe GA7.1, which is based on the GA7.0 "trunk" configuration, but includes a minimal set of changes to address the excessive aerosol forcing discussed in Sect. 4.5. As a result of this work, GA7.1 and GL7.0 are suitable for use as the physical atmosphere and land components in

the HadGEM3-GC3.1 and UKESM1 climate models that will be submitted to CMIP6.

## 2   Global Atmosphere 7.0 and Global Land 7.0

### 2.1   Dynamical formulation and discretisation

The UM's ENDGame dynamical core uses a semi-implicit semi-Lagrangian formulation to solve the non-hydrostatic, fully-compressible deep-atmosphere equations of motion (Wood et al., 2014). The primary atmospheric prognostics are the three-

dimensional wind components, virtual dry potential temperature, Exner pressure, and dry density, whilst moist prognostics such as the mass mixing ratio of water vapour and prognostic cloud fields as well as other atmospheric loadings are advected as free tracers. These prognostic fields are discretised horizontally onto a regular longitude/latitude grid with Arakawa C-grid staggering (Arakawa and Lamb, 1977), whilst the vertical discretisation utilises a Charney-Phillips staggering (Charney and Phillips, 1953) using terrain-following hybrid height coordinates. The discretised equations are solved using a nested iterative

approach centred about solving a linear Helmholtz equation. By convention, global configurations are defined on $2N$ longitudes and $1.5N$ latitudes of scalar grid-points, with the meridional wind variable held at the north and south poles and scalar and zonal wind variables first stored half a grid length away from the poles. This choice makes the grid-spacing approximately isotropic in the mid-latitudes and means that the integer $N$, which represents the maximum number of zonal 2 grid-point waves that can be represented by the model, uniquely defines its horizontal resolution; a model with $N = 96$ is said to be N96

resolution. Limited-area configurations use a rotated longitude/latitude grid with the pole rotated so that the grid's equator runs through the centre of the model domain. In the vertical, the majority of climate configurations use an 85 level set labelled $L85(50_t, 35_s)_{85}$, which has 50 levels below $18\,\mathrm{km}$ (and hence at least sometimes in the troposphere), 35 levels above this (and hence solely in or above the stratosphere) and a fixed model lid $85\,\mathrm{km}$ above sea level. Limited area climate simulations use a reduced 63 level set, $L63(50_t, 13_s)_{40}$, which has the same 50 levels below $18\,\mathrm{km}$, with only 13 above and a lower model top

at $40\,\mathrm{km}$. Finally, numerical weather prediction (NWP) configurations use a 70 level set, $L70(50_t, 20_s)_{80}$ which has an almost identical 50 levels below $18\,\mathrm{km}$, a model lid at $80\,\mathrm{km}$, but has a reduced stratospheric resolution compared to $L85(50_t, 35_s)_{85}$. Although we use a range of vertical resolutions in the stratosphere, a consistent tropospheric vertical resolution is currently used for a given GA configuration. A more detailed description of these level sets is included in the supplementary material to this paper.

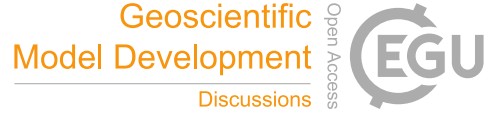

## 2.2 Structure of the atmospheric model time step

With ENDGame, the UM uses a nested iterative structure for each atmospheric time step within which processes are split into an outer loop and an inner loop. The semi-Lagrangian departure point equations are solved within the outer loop using the latest estimates for the wind variables. Appropriate fields are then interpolated to the updated departure points. Within the inner

loop, the Coriolis, orographic and non-linear terms are solved along with a linear Helmholtz problem to obtain the pressure increment. Latest estimates for all variables are then obtained from the pressure increment via a back-substitution process; see Wood et al. (2014) for details. The physical parametrisations are split into slow processes (radiation, large-scale precipitation and gravity wave drag) and fast processes (atmospheric boundary layer turbulence, convection and land surface coupling). The slow processes are treated in parallel and are computed once per time step before the outer loop. The source terms from the

slow processes are then added on to the appropriate fields before interpolation. The fast processes are treated sequentially and are computed in the outer loop using the latest predicted estimate for the required variables at the next, $n+1$ time step. A summary of the atmospheric time step is given in Algorithm 1. In practice two iterations are used for each of the outer and inner loops so that the Helmholtz problem is solved four times per time step. When using the prognostic aerosol scheme, this is included via a call to the UK Chemistry and Aerosol (UKCA) code after the main atmospheric time step; this call is currently

performed once per hour.

## 2.3 Solar and terrestrial radiation

Shortwave (SW) radiation from the Sun is absorbed and reflected in the atmosphere and at the Earth's surface and provides energy to drive the atmospheric circulation. Longwave (LW) radiation is emitted from the planet and interacts with the atmosphere, redistributing heat, before being emitted into space. These processes are parametrised via the radiation scheme,

which provides prognostic atmospheric temperature increments, prognostic surface fluxes and additional diagnostic fluxes. The SOCRATES [3] radiative transfer scheme (Edwards and Slingo, 1996; Manners et al., 2015) is used with a new configuration for GA7. Solar radiation is treated in 6 SW bands and thermal radiation in 9 LW bands, as outlined in Table 1). Gaseous absorption uses the correlated-$k$ method with newly derived coefficients for all gases (except where indicated below) based on the HITRAN 2012 spectroscopic database (Rothman et al., 2013). Scaling of absorption coefficients uses a look-up table

of 59 pressures with 5 temperatures per pressure level based around a mid-latitude summer profile. The method of equivalent extinction (Edwards, 1996; Amundsen et al., 2017) is used for minor gases in each band. The water vapour continuum is represented using laboratory results from the CAVIAR project (Continuum Absorption at Visible and Infrared wavelengths and its Atmospheric Relevance) between 1 and 5 μm (Ptashnik et al., 2011, 2012) and version 2.5 of the Mlawer–Tobin_Clough–Kneizys–Davies (MT_CKD-2.5) model (Mlawer et al., 2012) at other wavelengths.

Forty-one (41) $k$ terms are used for the major gases in the SW bands. Absorption by water vapour ($H_2O$), carbon dioxide ($CO_2$), ozone ($O_3$), oxygen ($O_2$), nitrous oxide ($N_2O$) and methane ($CH_4$) is included. Ozone cross-sections for the ultra-violet and visible come from Serdyuchenko et al. (2014) and Gorshelev et al. (2014), along with Brion-Daumont-Malicet (Daumont

---

[3]https://code.metoffice.gov.uk/trac/socrates

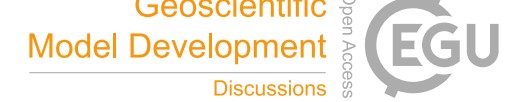



---

**Algorithm 1** Iterative structure of time step $n+1$. Here, we use two inner and two outer loops ($L = 2$, $M = 2$).

1: Given the solution at time step $n$, let the first estimate for a prognostic variable $F$ at time level $n+1$ be $F^{n+1} = F^n$

2: Compute slow parametrised processes and time level $n$ forcings $R_F^n$

3: **for** $m = 1, M$ **do** {*departure (outer-loop) iteration*}

4:     Solve the trajectory equations to compute the next estimate of the departure points using the time level $n$ and the latest estimate for time level $n+1$ wind fields

5:     Interpolate $R_F^n$ to departure points

6:     Compute time level $n+1$ predictors $F^*$

7:     Compute fast parametrised processes using latest $n+1$ predictor $F^*$

8:     Evaluate time level $n$ component of Helmholtz right hand side $\mathfrak{R}^n$

9:     **for** $l = 1, L$ **do** {*non-linear (inner-loop) iteration*}

10:         Evaluate non-linear and Coriolis terms $R_F^*$

11:         Evaluate time level $n+1$ component of Helmholtz right hand side $\mathfrak{R}^*$

12:         Solve the Helmholtz problem for the pressure increment $\pi'$ and hence obtain the next estimate for $\pi^{n+1} \equiv \pi^n + \pi'$

13:         Obtain the other prognostic variables at time level $n+1$ via back-substitution

14:     **end for**

15: **end for**

---

**Table 1.** Spectral bands for the treatment of incoming solar (SW) radiation (left) and thermal (LW) radiation (right).

| SW Band | Wavelength (nm) | LW Band | Wavenumber (cm$^{-1}$) | Wavelength ($\mu$m) |
|---|---|---|---|---|
| 1 | 200 - 320 | 1 | 1 - 400 | 25 - 10000 |
| 2 | 320 - 505 | 2 | 400 - 550 | 18.18 - 25 |
| 3 | 505 - 690 | 3 | 550 - 590 and 750 - 800 | 12.5 - 13.33 and 16.95 - 18.18 |
| 4 | 690 - 1190 | 4 | 590 - 750 | 13.33 - 16.95 |
| 5 | 1190 - 2380 | 5 | 800 - 990 and 1120 - 1200 | 8.33 - 8.93 and 10.10 - 12.5 |
| 6 | 2380 - 10000 | 6 | 990 - 1120 | 8.93 - 10.10 |
| - | - | 7 | 1200 - 1330 | 7.52 - 8.33 |
| - | - | 8 | 1330 - 1500 | 6.67 - 7.52 |
| - | - | 9 | 1500 - 2995 | 3.34 - 6.67 |

et al., 1992; Malicet et al., 1995) for the far-UV. In the first SW band, a single $k$-term is calculated for each 20 nm sub-interval from 200 to 320nm, and in band 2, a single $k$-term is calculated for each of the sub-intervals 320-400 nm and 400-505 nm.



This allows the incoming solar flux to be supplied on these finer wavelength bands for experiments concerning solar spectral variability. The solar spectrum uses data from NRLSSI (Lean et al., 2005) as recommended by the SPARC/SOLARIS [4] group. A mean solar spectrum for the period 2000-2011 is used when a varying spectrum is not invoked.

Eighty-one (81) $k$ terms are used for the major gases in the LW bands. Absorption by $H_2O$, $O_3$, $CO_2$, $CH_4$, $N_2O$, CFC-11

($CCl_3F$), CFC-12 ($CCl_2F_2$) and HFC134a ($CH_2FCF_3$) is included. For climate simulations, the atmospheric concentrations of CFC-12 and HFC134a are adjusted to represent absorption by all the remaining trace halocarbons. The treatment of $CO_2$ absorption for the peak of the $15\mu m$ band (LW band 4) is as described in Zhong and Haigh (2000). An improved representation of $CO_2$ absorption in the "window" region (8 - $13\mu m$) provides a better forcing response to increases in $CO_2$ (Pincus et al., 2015). The method of "hybrid" scattering is used in the LW which runs full scattering calculations for 27 of the major gas

$k$-terms (where their nominal optical depth is less than 10 in a mid-latitude summer atmosphere). For the remaining 54 $k$-terms (optical depth $> 10$) much cheaper non-scattering calculations are run.

Of the major gases considered, only $H_2O$ is prognostic; $O_3$ uses a zonally symmetric climatology, whilst other gases are prescribed using either fixed or time-varying mass mixing ratios and assumed to be well mixed.

Absorption and scattering by the following prognostic aerosol species are included in both the SW and LW using the

UKCA-Radaer scheme: sulphate, black carbon, organic carbon and sea salt. The aerosol scattering and absorption coefficients and asymmetry parameters are precomputed for a wide range of plausible Mie parameters and stored in look-up tables for use during run-time when the atmospheric chemical composition, including mean aerosol particle radius and water content are known. As the aerosol species are internally mixed within the modal aerosol scheme (see Table 3) the refractive indices of each mode are calculated online as a volume weighted mean of the component species contributing to that mode. The component

refractive indices are documented in the Appendix of Bellouin et al. (2013). Nucleation mode particles are neglected as they are not expected to contribute significantly to the atmospheric optical properties. The parametrisation of cloud droplets is described in Edwards and Slingo (1996) using the method of "thick averaging". Padé fits are used for the variation with effective radius, which is computed from the number of cloud droplets. In configurations using prognostic aerosol, cloud droplet number concentrations are not calculated within the radiation scheme itself but are calculated by the UKCA-Activate scheme (West

et al., 2014), which is based on the activation scheme of Abdul-Razzak and Ghan (2000). Note that in simulations using climatological rather than prognostic aerosol, the approach described here is not yet available and instead we use CLASSIC (Coupled Large-scale Aerosol Simulator for Studies in Climate, Bellouin et al. (2011)) aerosol climatologies and the calculation of optical properties and cloud droplet concentrations described in Sect. 2.3 of Walters et al. (2017). Both prognostic and climatological simulations of mineral dust also use the CLASSIC scheme. This is discussed in more detail in Sect. 3.8. The

parametrisation of ice crystals is described in Baran et al. (2016). Full treatment of scattering is used in both the SW and LW. The sub-grid cloud structure is represented using the Monte Carlo Independent Column Approximation (McICA) as described in Hill et al. (2011), with the parametrisation of subgrid-scale water content variability described in Hill et al. (2015b).

Full radiation calculations are made every hour using the instantaneous cloud fields and a mean solar zenith angle for the following 1 h period. Corrections are made for the change in solar zenith angle on every model time step as described in

---

[4]http://solarisheppa.geomar.de/ccmi





Manners et al. (2009). The emissivity and the albedo of the surface are set by the land surface model. The direct SW flux at the surface is corrected for the angle and aspect of the topographic slope as described in Manners et al. (2012).

## 2.4 Large-scale precipitation

The formation and evolution of precipitation due to grid scale processes is the responsibility of the large-scale precipitation — or microphysics — scheme, whilst small-scale precipitating events are handled by the convection scheme. The microphysics scheme has prognostic input fields of temperature, moisture, cloud and precipitation from the end of the previous time step, which it modifies in turn. The microphysics used is based on Wilson and Ballard (1999), with extensive modifications. The warm-rain scheme is based on Boutle et al. (2014b), and includes a prognostic rain formulation, which allows three-dimensional advection of the precipitation mass mixing ratio, and an explicit representation of the affect of sub-grid variability on autoconversion and accretion rates (Boutle et al., 2014a). We use the rain-rate dependent particle size distribution of Abel and Boutle (2012) and fall velocities of Abel and Shipway (2007), which combine to allow a better representation of the sedimentation and evaporation of small droplets. We also make use of multiple sub-time steps of the precipitation scheme, as in Posselt and Lohmann (2008) with one sub-time step for every two minutes of the model time step to achieve a realistic treatment of in-column evaporation. With prognostic aerosol, we use the UKCA-Activate aerosol activation scheme (West et al., 2014) to provide the cloud droplet number for autoconversion, where only soluble aerosol species (which can be composed of sulphate, sea salt, black carbon and organic carbon) contribute to the droplet number. When using climatological aerosol, the cloud droplet number is the same as that used in the radiation scheme. Ice cloud parametrisations use the generic size distribution of Field et al. (2007) and mass-diameter relations of Cotton et al. (2013).

## 2.5 Large-scale cloud

Clouds appear on sub-grid scales well before the humidity averaged over the size of a model grid box reaches saturation. A cloud parametrisation scheme is therefore required to determine the fraction of the grid box which is covered by cloud and the amount and phase of condensed water contained in those clouds. The formation of clouds will convert water vapour into liquid or ice and release latent heat. The cloud cover and liquid and ice water contents are then used by the radiation scheme to calculate the radiative impact of the clouds and by the large-scale precipitation scheme to calculate whether any precipitation has formed.

The parametrisation used is the prognostic cloud fraction and prognostic condensate (PC2) scheme (Wilson et al., 2008a, b) along with the cloud erosion parametrisation described by Morcrette (2012) and critical relative humidity parametrisation described in Van Weverberg et al. (2016). PC2 uses three prognostic variables for water mixing ratio — vapour, liquid and ice — and a further three prognostic variables for cloud fraction: liquid, ice and mixed-phase. The following atmospheric processes can modify the cloud fields: SW radiation, LW radiation, boundary layer processes, convection, precipitation, small-scale mixing (cloud erosion), advection and changes in atmospheric pressure. The convection scheme calculates increments to the prognostic liquid and ice water contents by detraining condensate from the convective plume, whilst the cloud fractions are updated using the non-uniform forcing method of Bushell et al. (2003). One advantage of the prognostic approach is that





clouds can be transported away from where they were created. For example, anvils detrained from convection can persist and be advected downstream long after the convection itself has ceased. The radiative impact of convective cores, which have not detrained into the large-scale variables, is represented by diagnosing a convective cloud amount (CCA) and convective cloud water (CCW) where the convection is active on a particular time-step. The CCA and CCW then get combined with the PC2

cloud fraction and condensate variables before these get passed to McICA to calculate the radiative impact of the combined cloud fields. Finally, the production of supercooled liquid water in a turbulent environment is parametrised following Furtado et al. (2016).

## 2.6  Sub-grid orographic drag

The effect of local and mesoscale orographic features not resolved by the mean orography, from individual hills through to

small mountain ranges, must be parametrised. The smallest scales, where buoyancy effects are not important, are represented by an effective roughness parametrisation in which the roughness length for momentum is increased above the surface roughness to account for the additional stress due to the sub-grid orography (Wood and Mason, 1993). The effects of the remainder of the sub-grid orography (on scales where buoyancy effects are important) are parametrised by a drag scheme which represents the effects of low-level flow blocking and the drag associated with stationary gravity waves (mountain waves). This is based on the

scheme described by Lott and Miller (1997), but with some important differences, described in more detail in Vosper (2015).

The sub-grid orography is assumed to consist of uniformly distributed elliptical mountains within the grid box, described in terms of a height amplitude, which is proportional to the grid box standard deviation of the source orography data, anisotropy (the extent to which the sub-grid orography is ridge-like, as opposed to circular), the alignment of the major axis and the mean slope along the major axis. The scheme is based on two different frameworks for the drag mechanisms: bluff body dynamics

for the flow-blocking and linear gravity waves for the mountain wave drag component.

The degree to which the flow is blocked and so passes around, rather than over the mountains is determined by the Froude number, $F = U/(NH)$ where $H$ is the assumed sub-grid mountain height (proportional to the sub-grid standard deviation of the source orography data) and $N$ and $U$ are respectively measures of the buoyancy frequency and wind speed of the low-level flow. When $F$ is less than the critical value, $F_c$, a fraction of the flow is assumed to pass around the sides of the orography, and

a drag is applied to the flow within this blocked layer. Mountain waves are generated by the remaining proportion of the layer, which the orography pierces through. The acceleration of the flow due to wave stress divergence is exerted at levels where wave breaking is diagnosed. The kinetic energy dissipated through the flow-blocking drag, the mountain-wave drag and the non-orographic gravity wave drag (see Sect. 2.7 below) is returned to the atmosphere as a local heating term.

## 2.7  Non-orographic gravity wave drag

Non-orographic sources — such as convection, fronts and jets — can force gravity waves with non-zero phase speed. These waves break in the upper stratosphere and mesosphere, depositing momentum, which contributes to driving the zonal mean wind and temperature structures away from radiative equilibrium. Waves on scales too small for the model to sustain explicitly are represented by a spectral sub-grid parametrisation scheme (Scaife et al., 2002), which by contributing to the deposited





momentum leads to a more realistic tropical quasi-biennial oscillation. The scheme, described in more detail in Walters et al. (2011), represents processes of wave generation, conservative propagation and dissipation by critical-level filtering and wave saturation acting on a vertical wavenumber spectrum of gravity wave fluxes following Warner and McIntyre (2001). Momentum conservation is enforced at launch in the lower troposphere, where isotropic fluxes guarantee zero net momentum, and by

imposing a condition of zero vertical wave flux at the model's upper boundary. In between, momentum deposition occurs in each layer where reduced integrated flux results from erosion of the launch spectrum, after transformation by conservative propagation, to match the locally evaluated saturation spectrum.

## 2.8 Atmospheric boundary layer

Turbulent motions in the atmosphere are not resolved by global atmospheric models, but are important to parametrise in order

to give realistic vertical structure in the thermodynamic and wind profiles. Although referred to as the "boundary layer" scheme, this parametrisation represents mixing over the full depth of the troposphere. The scheme is that of Lock et al. (2000) with the modifications described in Lock (2001) and Brown et al. (2008). It is a first-order turbulence closure mixing adiabatically conserved heat and moisture variables, momentum and tracers. For unstable boundary layers, diffusion coefficients ($K$ profiles) are specified functions of height within the boundary layer, related to the strength of the turbulence forcing. Two separate

$K$ profiles are used, one for surface sources of turbulence (surface heating and wind shear) and one for cloud-top sources (radiative and evaporative cooling). The existence and depth of unstable layers is diagnosed initially by moist adiabatic parcels and then adjusted to ensure that the magnitude of the buoyancy consumption of turbulence kinetic energy is limited to a specified fraction of buoyancy production, integrated across the boundary layer. This can permit the cloud layer to decouple from the surface (Nicholls, 1984). This same energetic diagnosis is used to limit the vertical extent of the surface-driven

$K$ profile when cumulus convection is diagnosed (through comparison of cloud and sub-cloud layer moisture gradients), except that in this case no condensation is included in the diagnosed buoyancy flux because that part of the distribution is handled by the convection scheme (which is triggered at cloud base). Mixing across the top of the boundary layer is through an explicit entrainment parametrisation that can either be resolved across a diagnosed inversion thickness or, if too thin, is coupled to the radiative fluxes and the dynamics through a sub-grid inversion diagnosis. If the thermodynamic conditions are right, cumulus

penetration into a stratocumulus layer can generate additional turbulence and cloud-top entrainment in the stratocumulus by enhancing evaporative cooling at cloud top. There are additional non-local fluxes of heat and momentum in order to generate more vertically uniform potential temperature and wind profiles in convective boundary layers. For stable boundary layers and in the free troposphere, we use a local Richardson number scheme based on Smith (1990). Its stable stability dependence is given by the "sharp" function over sea and by the "MES-tail" function over land (which matches linearly between an enhanced

mixing function at the surface and "sharp" at $200\,\mathrm{m}$ and above). This additional near-surface mixing is motivated by the effects of surface heterogeneity, such as those described in McCabe and Brown (2007). The resulting diffusion equation is solved implicitly using the monotonically damping, second-order-accurate, unconditionally stable numerical scheme of Wood et al. (2007). The kinetic energy dissipated through the turbulent shear stresses is returned to the atmosphere as a local heating term.



## 2.9 Convection

The convection scheme represents the sub-grid scale transport of heat, moisture and momentum associated with cumulus clouds within a grid box. The UM uses a mass flux convection scheme based on Gregory and Rowntree (1990) with various extensions to include down-draughts (Gregory and Allen, 1991) and convective momentum transport (CMT). The current scheme consists

of three stages: (i) convective diagnosis to determine whether convection is possible from the boundary layer; (ii) a call to the shallow or deep convection scheme for all points diagnosed deep or shallow by the first step; and (iii) a call to the mid-level convection scheme for all grid points.

The diagnosis of shallow and deep convection is based on an undilute parcel ascent from the near surface for grid boxes where the surface layer is unstable and forms part of the boundary layer diagnosis (Lock et al., 2000). Shallow convection is

then diagnosed if the following conditions are met: (i) the parcel attains neutral buoyancy below $2.5\,\mathrm{km}$ or below the freezing level, whichever is higher, and (ii) the air in model levels forming a layer of order $1500\,\mathrm{m}$ above this has a mean upward vertical velocity less than $0.02\,\mathrm{m\,s^{-1}}$. Otherwise, convection diagnosed from the boundary layer is defined as deep.

The deep convection scheme differs from the original Gregory and Rowntree (1990) scheme in using a convective available potential energy (CAPE) closure based on Fritsch and Chappell (1980). Mixing detrainment rates now depend on relative

humidity and forced detrainment rates adapt to the buoyancy of the convective plume (Derbyshire et al., 2011). The CMT scheme uses a flux gradient approach (Stratton et al., 2009).

The shallow convection scheme uses a closure based on Grant (2001) and has larger entrainment rates than the deep scheme consistent with cloud-resolving model (CRM) simulations of shallow convection. The shallow CMT uses flux–gradient relationships derived from CRM simulations of shallow convection (Grant and Brown, 1999).

The mid-level scheme operates on any instabilities found in a column above the top of deep or shallow convection or above the lifting condensation level. The scheme is largely unchanged from Gregory and Rowntree (1990), but uses the Gregory et al. (1997) CMT scheme and a CAPE closure. The mid-level scheme operates mainly either overnight over land when convection from the stable boundary layer is no longer possible or in the region of mid-latitude storms. Other cases of mid-level convection tend to remove instabilities over a few levels and do not produce much precipitation.

The timescale for the CAPE closure, which is used for deep and mid-level convection schemes, varies according to the large-scale vertical velocity. The values used vary from a shortest value equal to the convection time step when the ascent is strongest, to a maximum of either $4\,\mathrm{h}$ for mid-level convection, or the minimum of either $4\,\mathrm{h}$ or a time scale from a surface flux closure for deep convection.

## 2.10 Atmospheric aerosols and chemistry

As discussed in Walters et al. (2011), the precise details of the modelling of atmospheric aerosols and chemistry is considered as a separate component of the full Earth system and remains outside the scope of this document. The aerosol species represented and their interaction with the atmospheric parametrisations is, however, part of the Global Atmosphere component and is therefore included. Systems including prognostic aerosol modelling do so using the GLOMAP-mode (Global Model of



Aerosol Processes) aerosol scheme described in Mann et al. (2010) with updates described in Mulcahy et al. (in prep.a), which is included in the UM as part of the UKCA coupled chemistry and aerosol code. The scheme simulates speciated aerosol mass and number in 4 soluble modes covering the sub-micron to super-micron aerosol size ranges (nucleation, Aitken, accumulation and coarse modes) as well as an insoluble Aitken mode. The prognostic aerosol species represented are sulphate, black

carbon, organic carbon and sea salt. For more details see Sect. 3.8. Mineral dust is simulated using the CLASSIC dust scheme described in Woodward (2011). Systems not including prognostic aerosols use a three-dimensional monthly climatology for each aerosol species to model both the direct and indirect aerosol effects. Ideally, this should use the same aerosol species and parametrisation of the direct and indirect aerosol effects as we use for the prognostic scheme. As this capability has not yet been developed for GLOMAP-mode, however, we continue to use climatologies based on the CLASSIC aerosol scheme (Bel-

louin et al., 2011) as described in (Walters et al., 2017). In addition to the treatment of these tropospheric aerosols, we include a simple stratospheric aerosol climatology based on Cusack et al. (1998). We also include the production of stratospheric water vapour via a simple methane oxidation parametrisation (Untch and Simmons, 1999).

## 2.11  Land surface and hydrology: Global Land 7.0

The exchange of fluxes between the land surface and the atmosphere is an important mechanism for heating and moistening the

atmospheric boundary layer. In addition, the exchange of $CO_2$ and other greenhouse gases plays a significant role in the climate system. The hydrological state of the land surface contributes to impacts such as flooding and drought as well as providing freshwater fluxes to the ocean, which influences ocean circulation. Therefore, a land surface model needs to be able to represent this wide range of processes over all surface types that are present on the Earth.

The Global Land configuration uses a community land surface model, JULES (Best et al., 2011; Clark et al., 2011), to

model all of the processes at the land surface and in the sub-surface soil. A tile approach is used to represent sub-grid scale heterogeneity (Essery et al., 2003b), with the surface of each land grid box subdivided into five types of vegetation (broadleaf trees, needle-leaved trees, temperate C3 grass, tropical C4 grass and shrubs) and four non-vegetated surface types (urban areas, inland water, bare soil and land ice). The ground beneath vegetation is coupled to the vegetation canopy by longwave radiation and turbulent sensible heat exchanges. JULES also uses a canopy radiation scheme to represent the penetration

of light within the vegetation canopy and its subsequent impact on photosynthesis (Mercado et al., 2007). The canopy also interacts with falling snow. Snow buries the canopy for most vegetation types, but the interception of snow by needle-leaved trees is represented with separate snow stores on the canopy and on the ground. This impacts the surface albedo, the snow sublimation and the snow melt (Essery et al., 2003a). The vegetation canopy code has been adapted for use with the urban surface type by defining an "urban canopy" with the thermal properties of concrete (Best, 2005). This has been demonstrated

to give improvements over representing an urban area as a rough bare soil surface. Similarly, this canopy approach has also been adopted for the representation of lakes. The original representation was through a soil surface that could evaporate at the potential rate (i.e. a permanently saturated soil), which has been shown to have incorrect seasonal and diurnal cycles for the surface temperature (Rooney and Jones, 2010). By defining an "inland water canopy" and setting the thermal characteristics to those of a suitable mixed layer depth of water ($\approx 5\,\mathrm{m}$), a better diurnal cycle for the surface temperature is achieved.



Surface fluxes are calculated separately on each tile using surface similarity theory. In stable conditions we use the similarity functions of Beljaars and Holtslag (1991), whilst in unstable conditions we take the functions from Dyer and Hicks (1970). The effects on surface exchange of both boundary layer gustiness (Godfrey and Beljaars, 1991) and deep convective gustiness (Redelsperger et al., 2000) are included. Temperatures at $1.5\,\mathrm{m}$ and winds at $10\,\mathrm{m}$ are interpolated between the model's grid levels using the same similarity functions, but a parametrisation of transitional decoupling in very light winds is included in the calculation of the $1.5\,\mathrm{m}$ temperature.

SW radiation fluxes use a "first guess" snow-free albedo for each each land surface type, which can then be nudged towards an imposed grid box mean value taken from a climatology; this is further modified in the presence of snow. The albedo of the ocean surface is a function of the wavelength, the solar zenith angle, the $10\,\mathrm{m}$ wind speed and the chlorophyll content according to the Jin et al. (2011) parametrisation. The emitted LW radiation is calculated using a prescribed emissivity for each surface type.

Soil processes are represented using a 4-layer scheme for the heat and water fluxes with hydraulic relationships taken from van Genuchten (1980). These four soil layers have thicknesses from the top down of 0.1, 0.25, 0.65 and $2.0\,\mathrm{m}$. The impact of moisture on the thermal characteristics of the soil is represented using a simplification of Johansen (1975), as described in Dharssi et al. (2009). The energetics of water movement within the soil is accounted for, as is the latent heat exchange resulting from the phase change of soil water from liquid to solid states. Sub-grid scale heterogeneity of soil moisture is represented using the Large-Scale Hydrology approach (Gedney and Cox, 2003), which is based on the topography-based rainfall-runoff model TOPMODEL (Beven and Kirkby, 1979). This enables the representation of an interactive water table within the soil that can be used to represent wetland areas, as well as increasing surface runoff through heterogeneity in soil moisture driven by topography.

A river routing scheme is used to route the total runoff from inland grid points both out to the sea and to inland basins, where it can flow back into the soil moisture. Excess water in inland basins is distributed evenly across all sea outflow points. In coupled model simulations the resulting freshwater outflow is passed to the ocean, where it is an important component of the thermohaline circulation, whilst in atmosphere/land-only simulations this ocean outflow is purely diagnostic. River routing calculations are performed using the TRIP (Total Runoff Integrating Pathways) model (Oki and Sud, 1998), which uses a simple advection method (Oki, 1997) to route total runoff along prescribed river channels on a $1\,^\circ \times 1\,^\circ$ grid using a $3\,\mathrm{h}$ time step. Land surface runoff accumulated over this time step is mapped onto the river routing grid prior to the TRIP calculations, after which soil moisture increments and total outflow at river mouths are mapped back to the atmospheric grid (Falloon and Betts, 2006). This river routing model is not currently being used in limited-area or NWP implementations of the Global Atmosphere/Land.

## 2.12 Stochastic physics

A key component of many Ensemble Prediction Systems (EPSs) is the use of stochastic physics schemes to represent model error emerging from unrepresented or coarsely resolved processes such as numerical diffusion or fluctuations in the impact of physical parametrisations on the large-scale fields. The addition of unresolved variability around the deterministic solution adds spread between ensemble members and has been shown to improve ensemble predictions in the medium range (Palmer



et al., 2009; Tennant et al., 2011) as well as on seasonal (Weisheimer et al., 2011) and decadal time scales (Doblas-Reyes et al., 2009). The increase in the model's internal variability also helps to improve the model's climatology, through a noise-drift induced process. In particular, there is strong evidence of the positive impact of stochastic physics schemes on specific processes such as mid-latitude blocking (Berner et al., 2012), the Madden–Julian Oscillation (MJO, Madden and Julian, 1971; Weisheimer et al., 2014) and North Atlantic weather regimes (Dawson and Palmer, 2015).

In GA7, we use a standardised package of stochastic physics schemes (Sanchez et al., 2016) based on an improved version of the Stochastic Kinetic Energy Backscatter scheme version 2 (SKEB2, Tennant et al., 2011) and the Stochastic Perturbation of Tendencies scheme (SPT) with additional constraints designed to conserve energy and water. SKEB2 adds forcing to the large-scale flow to represent the backscatter of small-scale kinetic energy lost via numerical diffusion, whilst SPT stochastically scales the output of physical parametrisations to represent variability about their mean predictions. Despite the positive impact of these stochastic physics schemes on EPS and climate model performance, their formulation lacks a sound physical basis. For this reason, these schemes are not used in deterministic forecast systems, which are designed to forecast the best possible single prediction of the atmosphere's future state.

### 2.13 Global atmospheric energy correction

Long climate simulations of the Unified Model include an energy correction scheme, designed to ensure that numerical errors, inconsistent geometric assumptions and missing processes do not lead to any spurious drift in the atmosphere's total energy. The scheme accumulates the net flux of energy through the upper and lower boundaries of the atmosphere over a period of $1 \, \mathrm{day}$ and calculates the difference between this and the change in the atmosphere's internal energy. Any drift is compensated by the addition of a globally uniform temperature increment, which is applied every time step for the following day.

### 2.14 Ancillary files and forcing data

In the UM, the characteristics of the lower boundary, the values of climatological fields and the distribution of natural and anthropogenic emissions are specified using ancillary files. Use of correct ancillary file inputs can play as important a role in the performance of a system as the correct choice of many options in the parametrisations described above. For this reason, we consider the source data and processing required to create ancillaries as part of the definition of the Global Atmosphere/Land configurations. Table 2 contains the main ancillaries used as well as references to the source data from which they are created.

## 3 Developments since Global Atmosphere/Land 6.0

The previous section provides a general description of the whole of the GA7.0 and GL7.0 configurations. In this section, we describe in more detail how these configurations differ from the previously documented configurations of GA6.0 and GL6.0.

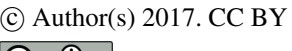



**Table 2.** Source datasets used to create standard ancillary files used in GA7.0/GL7.0. [%]This is expanded to a "zonally symmetric" 3D field in limited area simulations on a rotated pole grid.

| Ancillary field | Source data | Notes |
|---|---|---|
| Land mask/fraction | System dependent | |
| Mean/sub-grid orography | GLOBE 30″; Hastings et al. (1999) | Fields filtered before use |
| Land usage | IGBP; Global Soil Data Task (2000) | Mapped to 9 tile types |
| Soil properties | HWSD; Nachtergaele et al. (2008) | Three datasets blended via optimal interpolation |
| | STATSGO; Miller and White (1998) | |
| | ISRIC-WISE; Batjes (2009) | |
| Leaf area index | MODIS collection 5 | 4 km data (Samanta et al., 2012) mapped to 5 plant types |
| Plant canopy height | IGBP; Global Soil Data Task (2000) | Derived from land usage and mapped to 5 plant types |
| Bare soil albedo | MODIS; Houldcroft et al. (2008) | |
| Snow free surface albedo | GlobAlbedo; Muller et al. (2012) | Spatially complete white sky values |
| TOPMODEL topographic index | Marthews et al. (2015) | |
| SST/sea ice | System/experiment dependent | |
| Sea surface chlorophyll content | GlobColour; Ford et al. (2012) | |
| Ozone | SPARC-II; Cionni et al. (2011) | Zonal mean field used[%] |
| GLOMAP-mode emissions/fields: | | Only required for prognostic aerosol simulations |
|     Main primary emissions | CMIP5; Lamarque et al. (2010) | Includes $SO_2$, DMS (land), black carbon from fossil fuel, organic carbon from fossil fuel |
|     Biomass Burning | GFED3.1; van der Werf et al. (2010) | 10 year monthly means |
|     Volcanic $SO_2$ emissions | Andres and Kasgnoc (1998) | |
|     Gas phase aerosol precursors | UKCA-tropospheric chemistry simulations O'Connor et al. (2014) | |
|     Ocean DMS concentrations | Kettle et al. (1999) | |
| CLASSIC aerosol climatologies | System/experiment dependent | Used when prognostic fields not available |
| TRIP river paths | 1 ° data from Oki and Sud (1998) | Adjusted at coastlines to ensure correct outflow |

## 3.1 Dynamical formulation and discretisation

### Cubic Hermite interpolation and improved conservative advection for moist prognostics (GA ticket #135)

In GA6, the semi-Lagrangian interpolation to the departure point for moist prognostic variables was performed via bi-cubic interpolation in the horizontal and quintic interpolation in the vertical. The latter choice is one that has been made in global UM configurations for some time and was originally chosen to improve the fit to sharp discontinuities around the tropopause. For ENDGame's prognostic temperature variable, virtual dry potential temperature, the vertical interpolation used a cubic Hermite





formulation, which it still does at GA7. This is formed by matching the data and its derivative at the two levels closest to the departure point (rather than using the data at the four closest levels) and results in a spline interpolation with a continuous first derivative. The derivatives are estimated by fitting a quadratic polynomial to the data on three consecutive levels and evaluating its derivative at the central level. Formally, this is lower order than quintic (or even cubic Lagrange) interpolation, such that

the solution will be less accurate in general. The continuity of the first derivative, however, gives advection increments that correctly cancel under small amplitude oscillatory displacement in regions of strong gradients, such at the tropopause. In GA7, we apply this same vertical interpolation algorithm to all moist prognostic variables. The impact of this change is marked as "*q vertical interpolation - advection*" in Fig. 7 of Hardiman et al. (2015), which shows that in an atmosphere/land-only climate simulation at N96 horizontal resolution ($\approx 135\,\mathrm{km}$ in the mid-latitudes), this reduces the bias in lower-stratospheric

water vapour by $\approx 50\%$. This change also improves the dynamical core's internal consistency as it means that we use the same three-dimensional interpolation algorithms for temperature and moisture.

For systems enforcing the mass conservation of moist prognostics (which we formally treat as a "system dependent option" in the Global Atmosphere configuration) we change the algorithm used from that described in Zerroukat (2010) to the Optimized Conservative Filter scheme (OCF, Zerroukat and Allen, 2015). The OCF seeks to find a weighted conservative solution

between the high-order semi-Lagrangian solution discussed above and a lower-order (tri-linear) solution, where the weights are optimised such that the conservative solution stays as close as possible to the high-order one, whilst achieving conservation. This particular change has little impact on the moisture biases in the lower-stratosphere, but makes the conservation algorithm for moisture consistent with that used for atmospheric composition fields (see Sect. 3.8).

**Conservative advection of mass-weighted potential temperature (GA ticket #146)**

For an adiabatic flow, virtual dry potential temperature ($\theta_{vd}$), is constant within a fluid parcel moving with the flow. Additionally, the product of $\theta_{vd}$ and the density of dry air $\rho_d$ is also conserved, i.e.

$$\frac{\partial(\rho_d\theta_{vd})}{\partial t} + \nabla.(\rho_d\theta_{vd}\boldsymbol{u}) = 0. \tag{1}$$

In the ENDGame formulation, however, the fluid flow is not discretised in a conservative form; even in the absence of diabatic sources and sinks, the semi-implicit semi-Lagrangian time step does not satisfy the discrete form of Equation (1), which leads

to a spurious source of energy as discussed in Sect. 5.4.3 of Walters et al. (2017).

In GA7, we address this by applying the same OCF conservation-recovery algorithm discussed above in the context of moist prognostics to the $\theta_{vd}$ field; unlike the conservation of moist prognostics, however, this is not treated as a "system dependent option" and is applied in all systems using GA7. This improves the warm biases in the tropical tropopause layer as discussed in Hardiman et al. (2015). By removing this spurious source of energy, it also reduces the size of (and resolution dependence

in) the global energy correction step used in long climate simulations as described in Sect. 2.13.





**Reduction of solver tolerance in the iterative Helmholtz solver (GA ticket #153)**

As discussed in Sects 2.1 and 2.2, an important part of the ENDGame time step is the iterative solution of the linear Helmholtz problem to determine the model's pressure field. The approach is said to have reached its solution when a global normalised residual term (the solver "norm") is smaller than a pre-determined small value, or "tolerance". The smaller the tolerance, the

more accurate the solution, albeit at the cost of requiring more iterations to reach it. In GA6, the solver tolerance was set to $1 \times 10^{-3}$, which was thought a suitable balance between accuracy and computational cost. At horizontal resolutions at or above about N512 ($\approx 25\,\mathrm{km}$ in the mid-latitudes), however, global GA6 simulations suffered from numerical noise in the meridional wind near the poles in the topmost few levels (i.e. at altitudes of $65\,\mathrm{km}$ and above). Local calculations of the solver norms have also shown that this is largest close to the poles. Although cause and effect is unclear, reducing the global solver tolerance

by two orders of magnitude makes the noise almost imperceptible, but at the cost of increasing model run time by over $50\%$. Reducing by only a single order of magnitude, however, significantly reduces this noise, whilst only increasing run time by $\approx 15\%$. For this reason, in GA7 we have implemented this compromise and use a solver tolerance of $1 \times 10^{-4}$.

## 3.2    Solar and terrestrial radiation

**Improved treatment of gaseous absorption (GA ticket #16)**

GA7 includes an updated treatment of gas absorption with newly derived correlated-$k$ coefficients for all gases as described in section 2.3. Generation and validation of the gas absorption coefficients involved the creation of two configurations: a high wavelength resolution reference configuration (for offline comparison and diagnostic use), and a low resolution broadband configuration for use in the full model. The reference configurations contain 300 bands in the LW and 260 bands in the SW[5] and are based on the same data sources as the broadband files (primarily HITRAN 2012). These were validated against

independent line-by-line codes and subsequently used as a reference to verify the performance of the broadband configurations[6] over a range of atmospheric conditions and greenhouse gas forcing scenarios.

The resulting SW treatment improves the representation of $H_2O$, $CO_2$, $O_3$, and $O_2$ absorption compared to GA6 and also now includes absorption from $N_2O$ and $CH_4$. Changes result in increased atmospheric absorption and reduced surface (clear-sky) fluxes reducing errors compared to reference results from the Continual Intercomparison of Radiation Codes (CIRC,

Oreopoulos et al., 2012).

The new LW treatment improves the representation of all gases resulting in reduced clear-sky outgoing LW radiation and increased downwards surface flux. In particular, improvements to the treatment of the water vapour continuum significantly improve the downwards LW surface fluxes in regions of low humidity. The stratospheric heating rates, and in particular, the stratospheric water vapour forcing are significantly improved, addressing errors described by Maycock and Shine (2012). There

is also a significant improvement in the $CO_2$ forcing, especially for $CO_2$ concentrations of 4 times the present day value and

---

[5]SOCRATES spectral files: sp_lw_300_jm2, sp_sw_260_jm2
[6]SOCRATES spectral files: sp_lw_ga7, sp_sw_ga7





above. Figure 1 compares the errors in LW fluxes for various $CO_2$ concentrations based on the clear-sky atmospheric profiles used for CIRC.

In both the SW and LW regions there is a significant improvement in the band-by-band breakdown of absorption compared to GA6 where cancellation of errors between different bands was important. This should improve the interaction with band-by-band aerosol, cloud, and surface properties such as the albedo of the sea.

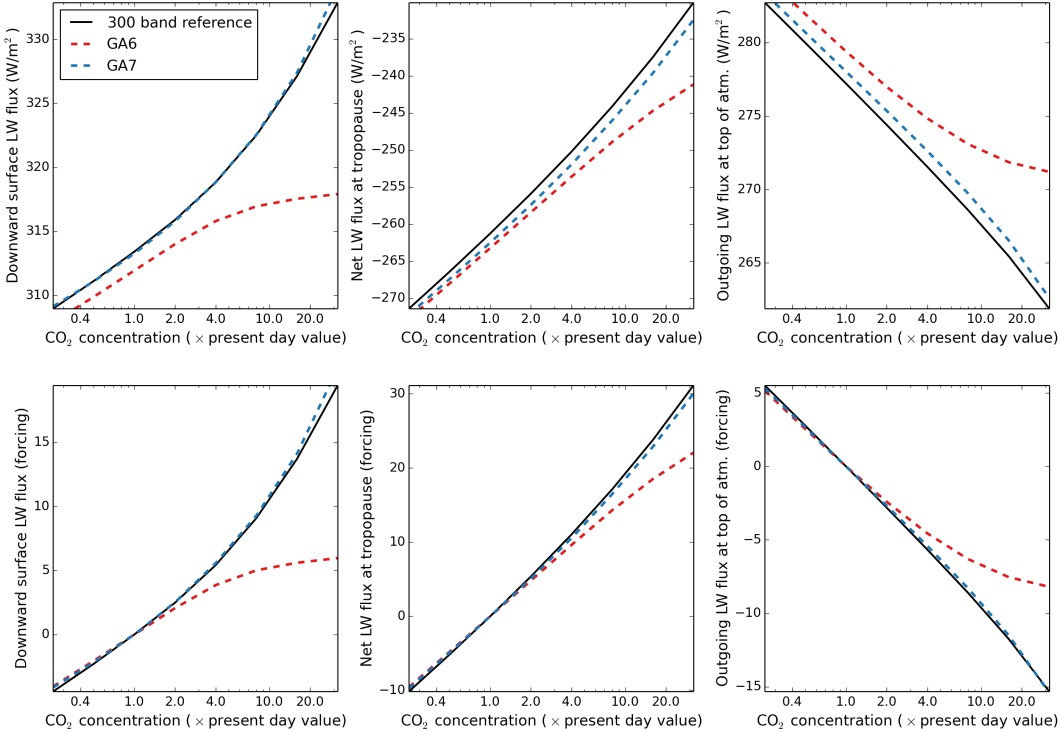

**Figure 1.** Comparison of LW flux errors due to changes in $CO_2$ concentration using GA6 and GA7 gaseous absorption compared to a 300 band LW reference configuration. Plots show an average response over the four clear-sky atmospheric profiles used for CIRC (Oreopoulos et al., 2012) which represent a broad range of water vapour path lengths. The top row shows the actual mean fluxes over the four profiles, whilst the bottom row shows the flux differences compared to a run using present day $CO_2$.

**Improved treatment of sub-grid scale cloud water content variability (GA ticket #15)**

In order to represent the radiative effects of sub-grid scale water content variability, the radiation scheme uses McICA as described in Hill et al. (2011). In McICA, the variability of water content within a grid box is determined by a fractional standard deviation ($f$), which is equal to the standard deviation of cloud water content in a grid box divided by its mean value. The transmission of radiation through a cloud is a convex function of the cloud water content such that increasing the value of $f$ decreases the radiative effect of a cloud, whilst decreasing $f$ has the opposite effect (e.g. Shonk and Hogan,





2010). In GA6, we used a globally constant value of $f = 0.75$, but in reality, the water content variability itself is variable and the magnitude of $f$ has been linked to cloud type, cloud fraction, wind shear and domain size (e.g. Hogan and Illingworth, 2003; Oreopoulos and Cahalan, 2005; Hill et al., 2012). At GA7, we include some of these effects by determining $f$ from the parametrisation of Hill et al. (2015b). In the interests of physical consistency, this parametrisation is also used in the warm rain part of the microphysics scheme. The implementation of the Hill et al. (2015b) parametrisation results in $f$ that depends on cloud fraction, vertical layer thickness and whether or not the cloud is convective, where convective cloud is identified based on the activation of the convection scheme.

The implementation of this scheme in GA7 included one change from that described in Hill et al. (2015b), whereby the grid box size dependency was replaced by a fixed effective resolution of $\approx 100\,\mathrm{km}$. It was discovered during testing that adjusting the sub-grid variability with resolution led to a large resolution-sensitivity in cloud properties, because the model did not resolve extra variability at the same rate as which the parametrisation removed it. This is because the parametrisation is based on observed variability, whilst the model resolves features at an effective resolution far greater than the grid box length (of order $10\Delta x$). Therefore, for GA configurations at resolutions $\geq 10\,\mathrm{km}$, the effective resolution required in the parametrisation is $\approx 100\,\mathrm{km}$. As the parametrisations do not show much change in variability beyond this, and the data used to construct them becomes increasingly sparse, it was felt simplest to use the same value in all GA resolutions.

**Consistent ice optical and microphysical properties (GA ticket #17)**

In GA7, we parametrise the scalar optical properties of ice crystals using the scheme described in Baran et al. (2016). This is based on an ensemble model of ice crystals developed by Baran and Labonnote (2007), where the bulk ice optical properties are derived by averaging habit-dependent scalar optical properties over an assumed particle size distribution function (PSD). This approach has the advantage that it is possible to generate ice optical properties from PSDs with the same microphysical assumptions used in the model's microphysics scheme; the same mass of ice is passed into each scheme and the bulk scalar ice optical properties are parametrised as a function of ice mass and temperature as described in Baran et al. (2016). This improves the self-consistency within the model in a way that is generally not achieved with scalar optical properties determined from an ice crystal effective dimension as was done in GA6 and in most other atmospheric models; as a result, those models usually assume inconsistent PSDs and mass-diameter relations in the microphysics and radiation schemes.

The difference between the Baran et al. (2016) scheme implemented in GA7 and the Baran et al. (2014) scheme that was originally proposed is that in the original scheme, the derived optical properties are fitted to be functions of the spectral band (see Table 1) and the model's prognostic ice water mass mixing ratio only, whilst in GA7, and Baran et al. (2016), there is an additional functional relationship to the atmospheric temperature (fitted to data sampled between -80°C and 0°C). This relationship to temperature was included to improve the temperature error in the tropical tropopause layer, which is highly sensitive to the specification of the scalar ice optical properties (Hardiman et al., 2015).

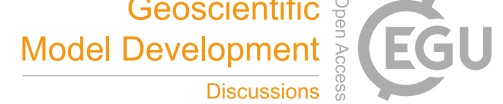

### 3.3 Large-scale precipitation

**Revised ice-microphysical properties (GA ticket #11)**

The representation of the ice PSD has been improved by adopting the parametrisation developed by Field et al. (2007). The mass-diameter relation of ice crystals is similarly updated to new, more accurate, measurements (Cotton et al., 2013) and the ice crystal fallspeeds are changed to be within the range of values reported in the literature. This PSD is derived from a much larger dataset of in situ cloud measurements than that used in GA6 and the data were corrected, as much as possible, for the effects of ice particle shattering during the measurement process. Similarly, the new mass-diameter relation was derived from measurements obtained with instruments designed to mitigate against the effects of shattering. Contamination of the GA6 PSD by shattering artefacts leads to an overestimation of small particle sizes; convective scale case studies suggest that this causes the microphysical characteristics of simulated clouds to be poorly predicted (Furtado et al., 2015).

The new PSD has several practical advantages. Firstly, it allows a unified representation of ice cloud in the microphysics and radiation schemes (see Sect. 3.2), an approach that was previously hampered by the effects of small particle sizes on SW reflectance from cloud tops. Secondly, case studies show that it works well with a realistic choice of particle fallspeeds (Furtado et al., 2015). By contrast, in GA6, fallspeeds that lay outside the range of available data were used in order to obtain realistic ice water contents. The main effects of the new parametrisation are on ice water content and specific humidity in the upper troposphere, which are shown to improve the simulation of the tropical tropopause layer (Hardiman et al., 2015). Moreover, when combined with the reduction in cirrus spreading discussed in Section 3.4, the new ice microphysics improves comparisons between modelled ice cloud radiative properties and satellite observations.

**New warm rain microphysics (GA ticket #52)**

In GA7, the warm rain part of the large-scale precipitation scheme has been almost completely re-written. The autoconversion and accretion parametrisations are now those of Khairoutdinov and Kogan (2000), following work by Boutle and Abel (2012) to demonstrate that this significantly improves the amount of precipitation produced by marine stratocumulus, and leads to improvements in the cloud cover, liquid water content, and boundary layer structure. In addition to this, improvements to the evaporation and sedimentation code have removed some undesirable consequences of the previous implementation, such as significant evaporation of rain inside cloud and an explicit non-conservation of rain water. Hill et al. (2015a) demonstrated that this new scheme significantly improves the representation of aerosol-cloud-precipitation interactions relative to the scheme used in GA6.

The new scheme also includes an explicit representation of how sub-grid variability affects microphysical process rates, based on Boutle et al. (2014a). The local process rates are upscaled to the grid box size based on parametrisations of the hydrometeor fractional standard deviation within a grid box, given by Hill et al. (2015b) for cloud and Boutle et al. (2014a) for rain. Note this means that for cloud water content, the same parametrisation of sub-grid variability is used consistently in the radiation and microphysics. Without parametrisation of the sub-grid variability, the model would underestimate autoconversion and accretion rates and it would not be possible to implement the Khairoutdinov and Kogan (2000) parametrisations. The





parametrisation of sub-grid rain fraction has also been improved, ensuring this is set consistently by either the fraction of autoconverting cloud or melting snow when rain is created. To avoid the need to advect this quantity, when rain is advected into a grid box which was previously rain-free, the rain fraction is set to the fraction of cloud directly above it, as that is likely to be the cloud from which the rain originated, and will have been advected by the cloud-scheme. The implementation of this

5  scheme in GA7 uses a fixed effective resolution of $\approx 100\,\text{km}$ rather than the grid box size dependency described in Boutle et al. (2014a). The reasons for this are discussed in Sect. 3.2.

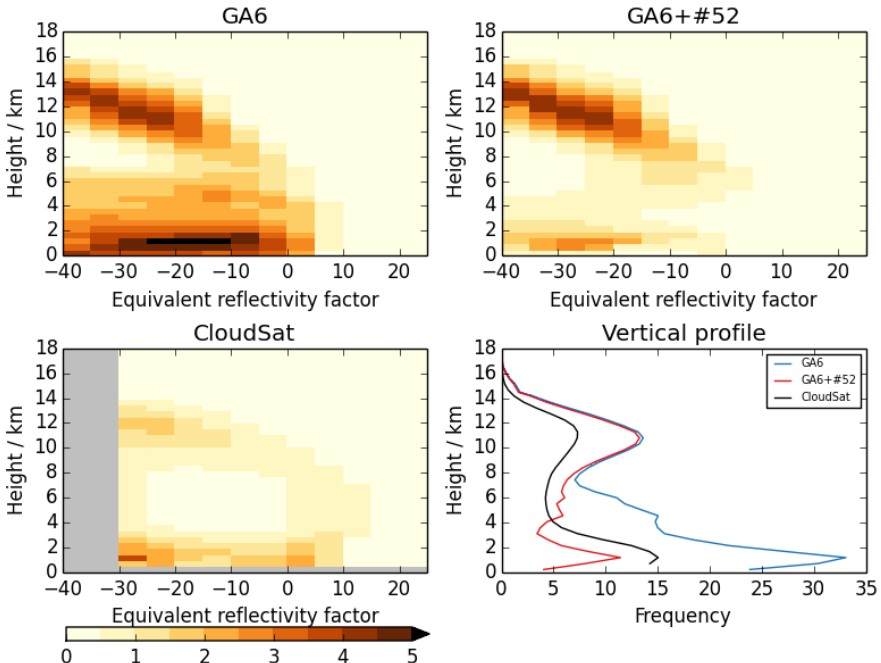

**Figure 2.** Histograms of height vs 94 GHz radar reflectivity over a trade cumulus region (130–160°W, 0–20°S), showing climatologies of CloudSat observations and simulated CloudSat data from 20 year N96 atmosphere/land-only climate simulations using GA6.0 and GA6.0 plus the new warm microphysics scheme.

Figure 2 summarises the affect of this change on low cloud and light rain. It has been noted elsewhere that previously, like many general circulation models (GCMs), the UM had too much rain in the lightest rain rate category (Bodas-Salcedo et al., 2008; Stephens et al., 2010). This is shown by the large frequency of radar returns in the $-30$–$0\,\text{dBZ}$ range below $2\,\text{km}$ for

10  GA6, in stark contrast to the observations from CloudSat. The inclusion of the new warm microphysics scheme considerably improves this bias, with simulated radar returns now a very good match to CloudSat observations below $2\,\text{km}$. The complete GA7 package shows similar improvements, effectively removing the long-standing model bias of excessive light rain.

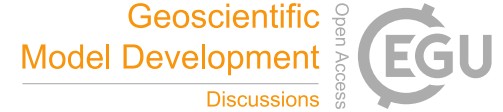

### 3.4 Large-scale cloud

**Including the radiative impact of convective cores (GA ticket #44)**

In the PC2 cloud scheme, the impact of convective cloudiness is represented by source terms that couple the convection scheme to PC2 in a manner following Tiedtke (1993) and Wilson et al. (2008a). As a convective plume rises and mixes with

its environment, it detrains condensate and cloud fraction into the environment and hence the prognostic cloud variables that get passed to the radiation scheme. As a result, it is only once condensate has detrained from the convective plume that it will have a radiative impact, whilst the radiative effect of the core of the convective updraught is ignored. This was originally justified by the fact that the fraction of the grid box occupied by the convective updraughts in a mass-flux convection scheme is assumed to be small. However, for some convective cloud types, such as shallow fair-weather cumulus, clouds may not detrain

much into the environment, but still have a significant radiative impact. To include the impact of these clouds, from GA7 we use a convective cloud model to include the radiative impact of the convective cores themselves. For shallow convection, the convective cloud amount (CCA) is calculated from the cloud-base mass-flux divided by the convective velocity scale following Grant and Lock (2004). This is then modified by a shape function that has its maximum value at cloud-base, its minimum at cloud top and decreases exponentially with height. For mid-level or deep convection, the CCA is calculated from the convective

precipitation rate, $P$, using CCA$= a + b \ln P$, following Slingo (1987), where $a = 0.3$ and $b = 0.025$. The convection scheme also calculates a profile of convective cloud water (CCW). The profiles of CCA and CCW are then combined with the PC2 cloud fields before being passed to the cloud generator used by McICA to calculate the radiative impact of the clouds. The models for CCA and CCW were originally developed to represent the cloud in the entire convective column (and not just the core of the convective updraught) so the values of CCA and CCW are scaled down before being combined with the PC2 cloud.

Note also that CCA and CCW are not added to the prognostic cloud fields themselves, and hence are not advected by the flow; instead, they are only radiatively active on the time steps in which convection has been diagnosed.

**Consistent treatment of phase change for convective condensate passed to PC2 (GA ticket #58)**

One benefit of using PC2 for modelling cloud created from detrained convective condensate is that this allows a consistent treatment of cloud, independent of the source of the cloud itself. The microphysical assumptions in the generation of cloud from

large-scale processes and convective processes, however, are currently independent. Whilst the microphysical assumptions in the convection scheme are far simpler than those used elsewhere in the model, it is still beneficial to ensure consistency at the level to which this is possible. One inconsistency identified in the GA6 treatment of convective cloud is in the phase of condensate passed from the convection scheme to PC2. To avoid an abrupt change of phase, in GA6 the phase of condensate detrained from convection scaled linearly from 100% liquid at 0°C to 100% ice at -20°C; in PC2, however, the maximum

temperature at which ice could form was -10°C. Here, we improve this consistency by reducing the upper limit at which ice can be formed by convection to -10°C, so that there are matching assumptions in the large-scale cloud and convection schemes.

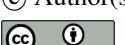



**Turbulence based critical relative humidity (GA ticket #89)**

The PC2 cloud scheme uses a critical relative humidity ($RH_\mathrm{crit}$) to determine when to initiate cloud in cloud-free grid boxes with increasing relative humidity ($RH$), and to remove cloud from fully cloud-filled grid boxes in which $RH$ is reduced. In previous GA configurations, $RH_\mathrm{crit}$ was a constant global value for each model level, which is a simplification, but is tunable to

global mean cloud distributions. This is undesirable for future climate projections, however, as these could show large changes in global cloud and $RH$ distributions, which might not be handled correctly in these cloud initiation and removal processes. Therefore, in GA7 we have implemented a method for calculating a variable $RH_\mathrm{crit}$ based on sub-grid turbulence.

The method is discussed in Van Weverberg et al. (2016), and involves parametrising the sub-grid variance and co-variance of temperature and humidity in terms of the resolved vertical gradients and sub-grid mixing length, eddy diffusivity and turbulent

kinetic energy (TKE) calculated by the boundary layer parametrisation. The TKE is diagnosed from the vertical velocity variance, $\sigma_w^2$, which is given by:

$$\sigma_w^2 = K_m \tau_{turb}^{-1}, \tag{2}$$

where $K_m$ is the eddy diffusivity for momentum and $\tau_\mathrm{turb}^{-1} = \max(\tau_\mathrm{surf}^{-1}, \tau_\mathrm{sc}^{-1}) + \tau_\mathrm{sbl}^{-1}$ is a turbulence timescale, calculated following Suselj et al. (2012) as a combination of convective and stable boundary layer timescales. The stable timescale is

given by $\tau_\mathrm{sbl} = N/0.7$ where $N$ is the Brunt-Väisälä frequency. The convective timescales are derived following the large-eddy simulations (LES) of Holtslag and Moeng (1991) and given by

$$\tau_\mathrm{surf} = \frac{C_{ws}^{2/3} \kappa z_h}{1.33 w_m}; \tag{3a}$$

$$\tau_\mathrm{sc} = \frac{g_1 \kappa z_{ml}}{1.33 V_{sc}}, \tag{3b}$$

where $\kappa$ is the von Karman constant, $z_h$ and $z_{ml}$ are the surface and cloud-top driven mixed layer depths, $w_m$ and $V_{sc}$ are

surface and cloud-top velocity scales, $C_{ws} = 0.25$ and $g_1 = 0.85$. Strictly speaking, Whilst $\sigma_w^2$ is the major component of TKE in a GCM, this is not a good approximation near the surface. The vertical velocity variance must tend to zero near the surface, but the TKE remains high due to continuity as horizontal fluctuations converge/diverge near the base of vertical fluctuations. To represent this, we set the TKE, $e$, equal to $\sigma_w^2$, but hold it constant below the maximum value of the surface driven non-local component to $K_m$.

To ensure numerical stability of the scheme, we constrain the calculated $RH_\mathrm{crit}$ value to lie between a maximum and minimum value. These values are calculated from aircraft observations of cumulus and stratocumulus clouds in the VOCALS (Wood et al., 2011) and RICO (Rauber et al., 2007) campaigns. Using all available flight data, Fig. 3 shows the mean $RH_\mathrm{crit}$ as a function of the flight leg-length (grid box size), and the 5th/95th percentiles of the data. We use fits to these as the maximum and minimum allowed values of $RH_\mathrm{crit}$.

Finally, in a change from the original implementation of Wilson et al. (2008a), because the $RH_\mathrm{crit}$ has been calculated based on the assumption of a triangular probability density function (PDF), we assume this shape when initiating cloud in PC2, rather than the previously used top-hat PDF. Van Weverberg et al. (2016) has shown that the parametrisation of $RH_\mathrm{crit}$ is a reasonable





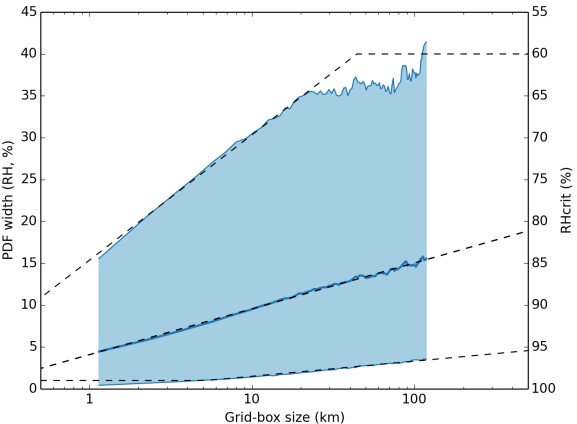

**Figure 3.** Mean and 5th/95th percentiles (central line and edges of the blue shaded region) of $RH_{\mathrm{crit}}$ as a function of flight-length from aircraft observations, and fits to the data (dashed lines) used in the model parametrisation.

match to independent lidar observations, and the implementation shows no degradation to GA6 performance, with the desired benefit of being less tuned to the present day climate.

**Removal of redundant complexity when dealing with ice cloud (GA ticket #98)**

PC2 deals with falling ice cloud condensate by increasing the ice cloud fraction in the layer into which the frozen condensate falls. The increased ice cloud fraction is larger than that in the layer above to represent some lateral displacement of the vertically-projected falling ice due to shear-generated fall-streaks. In GA4, this calculation was modified to use the local shear in the model's winds rather than a globally constant value. This led to an unrealistic reduction in mean ice cloud fraction, which was mitigated by introducing a cirrus spreading term that increased the frozen cloud fraction, $C_{\mathrm{ff}}$, via

$$\partial C_{\mathrm{ff}}/\partial t = 2r(C_{\mathrm{ff}} - C_{\mathrm{ff}}^2), \tag{4}$$

where $r$ is the cirrus spreading rate. The $C_{\mathrm{ff}} - C_{\mathrm{ff}}^2$ term ensures that ice cloud spreads more slowly when there is little cloud present, or as the grid box approaches an overcast state. In GA6, this rate was set to $r = 1.0 \times 10^{-3}\,\mathrm{s}^{-1}$.

Separately, in order to avoid the model producing regions of extensive ice cloud fraction when the ice water content was very low, GA6 included a term in the ice cloud fraction tendency, which ensured that if the in-cloud ice water content (grid box mean ice water content divided by ice cloud fraction) was less than $1.0 \times 10^{-6}\,\mathrm{kg\,kg}^{-1}$, the ice cloud fraction would be reduced accordingly. In GA6, both of these terms were necessary for the model to achieve a realistic distribution of ice cloud fraction, but in some regions, they were found to be acting in strong opposition. As part of the development of the package of cloud changes in GA7, we originally planned to reduce the cirrus spreading rate to a value very close to zero; in tuning the





final GA7 configuration as described in Sect. 3.11, this was increased to a final value of $r = 1.0 \times 10^{-5}\,\mathrm{s}^{-1}$. This is still small enough to allow us to remove the minimum in cloud ice water content.

**Turbulent production of liquid water in mixed phase clouds (GA ticket #120)**

Many atmospheric models are known to have problems producing and maintaining super-cooled liquid and mixed-phase
clouds, which instead are preferentially glaciated into ice-only clouds (Illingworth et al., 2007; Klein et al., 2009). A lack of liquid water in cold clouds has been implicated as a major contributor to severe model biases, particular in the southern hemisphere storm-tracks, where observations suggest that nature produces an abundance of super-cooled liquid water (Williams et al., 2013; Bodas-Salcedo et al., 2014). In this region, too little modelled super-cooled liquid leads to too little SW radiation reflected out to space and hence too much solar heating of the sea surface. This can lead to a host of problems in the simulation
of the coupled Earth system (see Hyder et al., submitted, for a review).

Motivated by these factors, Field et al. (2014) developed a new approach to parametrising the production of liquid water in mixed-phase clouds. They analytically solve the dynamics of supersaturation fluctuations in turbulent mixed-phase clouds under the action of adiabatic-lifting by turbulent air-motions, exchange of air between the cloud and its environment and the depletion of supersaturation by microphysical growth of the ice phase; this solution is used to calculate a probability
distribution of supersaturation. The liquid-cloud properties (water content and cloud fraction) are then calculated as moments of this distribution. The distribution is Gaussian, with mean and variance specified in terms of the parameters that describe the turbulence and the state of any pre-existing ice cloud. The parametrisation was tested against LES of mixed-phase clouds, with which it was found to be in good agreement (Hill et al., 2014).

The Field et al. (2014) parametrisation was implemented in the UM by Furtado et al. (2016). To close the model, the sub-
grid probability distribution is specified using the diagnostic of vertical velocity variance from the boundary-layer scheme, Equation (2), and the ice PSD from the microphysics scheme. The inclusion of the parametrisation was shown to increase the amount of super-cooled liquid and mixed-phase cloud, which improved the simulation of a case study of Arctic stratus and reduced biases in outgoing SW radiation over the Southern Ocean. However, the parametrisation performed poorly in the tropics, where it led to an over-production of liquid water in warm clouds. The was traced to assumptions in the model
which limit its validity to regimes where liquid-condensation is relatively small. Therefore, in GA7, the scheme is only used for temperatures below $0^\circ$C, where this approximation can be shown to be reasonable (Furtado et al., 2016). Above $0^\circ$C, liquid condensation is handled by the PC2 cloud initialisation scheme, which in GA7 uses the turbulence based $RH_{\mathrm{crit}}$ scheme described above.

## 3.5 Sub-grid orographic drag

**Introduction of heating due to gravity-wave dissipation (GA ticket #87)**

In GA7, we introduce terms for the conversion of kinetic energy to frictional heating, where drag is exerted on the flow, which were neglected in previous releases of the GA configuration. This includes heating corresponding to gravity-wave breaking (in





both the orographic and non-orographic schemes) and low-level flow blocking drag, thus improving the energy conservation of the model[7]. The frictional heating can be written as

$$\frac{\partial T}{\partial t} = -\frac{1}{c_p}\left(u\frac{\partial u}{\partial t} + v\frac{\partial v}{\partial t}\right), \qquad (5)$$

where $T$ is temperature, $c_p$ is the specific heat capacity at constant pressure and the $\partial/\partial t$ terms are the total tendencies due to the (orographic and non-orographic gravity-wave) drag schemes. The heating term is small in a global average sense. In the lower troposphere, where the dominant contribution comes from flow-blocking drag, global mean values are typically only $\sim 10^{-2}\,\mathrm{K\,day^{-1}}$ at N96 resolution, although locally, values can be as large as $10\,\mathrm{K\,day^{-1}}$ over the major mountain ranges. In the middle atmosphere, the heating comes from gravity-wave dissipation. Maxima associated with orographic gravity waves are typically between 10 and $20\,\mathrm{K\,day^{-1}}$ at heights of $50\,\mathrm{km}$ in the winter hemisphere over major orography. At higher levels, the contribution from the non-orographic gravity-wave drag provides more widespread heating with global mean heating rates at $65\,\mathrm{km}$ of $\sim 1\,\mathrm{K\,day^{-1}}$.

### 3.6 Atmospheric boundary layer

**Revised dependence of boundary layer entrainment on decoupling (GA ticket #13)**

The parametrisation of turbulent entrainment through the top of cloudy boundary layers involves sources from both cloud top (radiative and evaporative cooling) and the surface (positive buoyancy fluxes and wind shear). When the cloud layer is decoupled from the surface, this implies that the stratification associated with the decoupling inversion will restrict the surface driven turbulence from affecting the cloud layer, and in particular from driving entrainment at cloud top. Currently this impact of decoupling is diagnosed to occur when $\Delta\theta_{vl}$ exceeds $0.5\,\mathrm{K}$, where $\theta_{vl}$ is the adiabatically conserved virtual potential temperature in cloud-free air, which is used as a simplified measure of buoyancy. Single column model (SCM) comparisons with LES of the transition from stratocumulus to trade cumulus (e.g. Neggers et al., accepted) show that this leads to a sudden and substantial decrease in parametrised entrainment in the SCM that is not seen in the LES. Here, we make this abrupt transition more gradual by still including the entire impact of the surface contribution for $\Delta\theta_{vl} < 0.5\,\mathrm{K}$, but weighting this down linearly until there is no contribution above $\Delta\theta_{vl} = 1\,\mathrm{K}$. Note that this comparison with LES implies that some surface driven entrainment should continue for longer during the decoupling process and so will potentially lead to enhanced thinning of stratocumulus during the day.

**Forced convective clouds and resolved mixing across the boundary layer top (GA ticket #83)**

Prior to GA7, the parametrised boundary layer entrainment flux was implemented simply at the flux-level at the top of the mixed layer, implying that the vertical resolution was insufficient to resolve the distribution of this flux across the capping inversion. As vertical resolution becomes finer, this approach becomes increasingly untenable. In addition, for relatively weak

---

[7]Frictional heating from drag in the boundary layer scheme was already included, and has been since GA3.





inversions capping strongly surface-heated boundary layers (which occur commonly in desert regions) capping inversions can easily extend over 1 km and so should already be resolved (e.g. Garcia-Carreras et al., 2015).

To distribute the entrainment fluxes across the capping inversion, its thickness, $\Delta z_i$, must first be diagnosed. Assuming that this thickness is largely determined by the height to which turbulent thermals impinging into the stable stratification aloft can

penetrate, an energetic argument (Beare, 2008) implies:

$$6.3 w_m^2 = \int\limits_{z_h}^{z_h + \Delta z_i} b \, \mathrm{d}z. \tag{6}$$

where $w_m$ is the boundary layer velocity scale ($w_m^3 = u_*^3 + 0.25 w_*^3$, $w_*$ is the convective velocity scale and $u_*$ the friction velocity) and $b$ is the buoyancy, taken here from the convective diagnosis parcel. Note that the empirical constant in Equation (6) is actually the same as Beare's value of 2.5, because the definition of $w_m^3$ here differs by a factor of 4 (and $6.3 \approx 2.5 \times 4^{2/3}$).

The integration over the depth of the inversion is calculated working upwards from the level of neutral buoyancy, $z_h$, assuming piece-wise linear variation between grid-levels. Note that if this predicts that a boundary layer parcel has sufficient energy to penetrate any convective inhibition, and so reach the level of free convection, then the convection scheme is triggered. As a result, this represents a significant change in this triggering.

If $\Delta z_i$ is thicker than the model grid spacing (and the convection scheme has not been triggered), the entrainment fluxes

across the inversion are implemented as a standard down-gradient diffusive flux, with the diffusion coefficient $K_h$ between levels $k$ and $k + 1$ given by

$$K_h|_{k+\frac{1}{2}} = \frac{-\overline{w'\theta'_{vl}}}{(\theta_{vl\,k+1} - \theta_{vl\,k})/(z_{k+1} - z_k)}.$$

Within the inversion, $\overline{w'\theta'_{vl}}$ is assumed to decrease from the standard parametrised entrainment flux at the inversion base ($\overline{w'\theta'_{vl}}|_{entr}$) to zero at the inversion top following a cosine function, i.e.:

$$\overline{w'\theta'_{vl}} = \overline{w'\theta'_{vl}}|_{entr} \cos\left(\pi \frac{z'}{2}\right), \tag{7}$$

where $z' = (z - z_h)/\Delta z_i$ is scaled height within the inversion.

At points where the convection scheme is triggered, the top of the surface-based turbulently mixed layer was formerly capped at the lifting condensation level (LCL), so that the only mixing across the cumulus cloud base was through the convection scheme. This was seen to lead to errors in the mean profiles across the LCL, with the most extreme, if rare, examples including

superadiabatic lapse rates and large decreases in moisture. Using the boundary layer parametrisation to couple cloud and sub-cloud layers has the numerical advantage of being solved implicitly, but is also consistent with LES evidence that turbulence continues to show characteristics of the sub-cloud layer across the cloud base transition region (Grant and Lock, 2004). It is also consistent with the above resolved approach to boundary layer entrainment fluxes to implement a (potentially) resolved profile of fluxes across the top of the sub-cloud layer in cumulus-capped regimes. The approach adopted here is to extend the

algorithm used to diagnose boundary layer decoupling by adjusting the vertical extent of the surface-driven $K$ profile such





that the buoyancy consumption of turbulence kinetic energy is limited. In this case, however, no condensation is included in the diagnosed buoyancy flux because that part is handled by the convection scheme. Idealised clear-sky convective boundary layers (where the magnitude of the entrainment buoyancy flux is a fraction, $A_1$, of the surface flux) suggest that the ratio of the integrated buoyancy consumption to production is $A_1^2$ which, consistent with the entrainment parametrisation, is taken to be

5  0.05.

Now that the thickness of the capping inversion has been parametrised, this allows forced convective clouds to be represented. These clouds form in undulations of the top of convective boundary layers, but remain too shallow to reach their level of free convection (and become fully fledged cumulus clouds). They currently require special treatment because these sub-grid undulations of the capping inversion can typically imply a rather bimodal moisture distribution, consisting of moist boundary

layer domes surrounded by intrusions of very dry free-tropospheric air. From a survey at the Southern Great Plains ARM site, Zhang and Klein (2013) found almost 40% of summertime fair-weather shallow cumulus clouds were of this forced type and so would likely make a significant contribution to the radiation budget. A simple approach to represent them is adopted here. A profile of equilibrium forced cloud fraction is parametrised as varying linearly in height between a cloud-base value, at the LCL of the convection diagnosis parcel, and a cloud-top value top of 0.1. The cloud-base cloud fraction is parametrised as varying

linearly with cloud depth, between a minimum of 0.1 and a maximum of 0.3 for cloud depths between $100\,\mathrm{m}$ and $300\,\mathrm{m}$, based loosely on the observations of Zhang and Klein (2013). The cloud top is taken to be the top of the boundary layer inversion (at $z_h + \Delta z_i$) or, in cumulus-capped regimes, the top of the surface-driven $K$ profile. The in-cloud liquid water content at the top of the inversion is taken to be that of the adiabatic convection diagnosis parcel, with linear interpolation used between the lifting condensation level and inversion top. To allow for sub-adiabatic water content (due to lateral mixing or microphysical

processes) the in-cloud water content can be reduced by a factor that has been set to 0.5 in GA7. Increments are calculated, as necessary, to increase the prognostic cloud fraction and water content variables to these forced convective cloud fraction and liquid water content profiles at the end of the time step.

### 3.7 Convection

**Introduction of the 6A convection scheme (GA ticket #64)**

Major changes to parametrisations in the UM are indicated by incrementing the version of the scheme, with each version denoted by a number/letter combination. GA6 used the 5A version of the convection scheme; the parcel ascent calculations in the 5A scheme and its predecessors were originally developed at a time when model resolution, particularly in the vertical, and the demand for accuracy in the parcel ascent were lower both than they are currently and than they are expected to be in the future. Motivated by this, a review of the 5A convection scheme was undertaken and areas where improvements could be

made were identified. These improvements were implemented in the 6A convection scheme, which is used in GA7. The main improvements made between the 5A and 6A convection schemes are:

- The calculation of forced detrainment assumes that: *(i)* the detrained mass is saturated and is neutrally buoyant with respect to the environment; and *(ii)* that the remaining convective plume is buoyant and saturated. These conditions result





in implicit equations for the potential temperature of the detrained mass and the residual plume. In the 5A convection scheme, a single iteration is used to solve each of these implicit equations, whereas the 6A scheme uses three iterations;

– After the convective parcel is lifted from one level to the next in a dry ascent, it is brought to saturation. This involves solving for potential temperature at saturation at a given pressure, $\theta^{P,\mathrm{sat}}$, from

$$c_p \Pi \left( \theta^{P,\mathrm{sat}} - \theta^{P,\mathrm{dry}} \right) = L \left( q^{P,\mathrm{dry}} - q_{\mathrm{sat}} \left( \theta^{P,\mathrm{sat}} \right) \right), \tag{8}$$

where $c_p$ is the specific heat capacity at constant pressure, $\Pi$ is the Exner pressure, $\theta^{P,\mathrm{dry}}$ and $q^{P,\mathrm{dry}}$ are the potential temperature and specific humidity of the parcel after dry ascent and $q_{\mathrm{sat}}$ is the specific humidity at saturation. This calculation is performed iteratively, where the 5A convection scheme uses two iterations and the 6A scheme uses three iterations, to bring this closer to convergence. Unlike the 5A scheme, the 6A convection scheme also allows the evaporation of parcel condensate if the parcel becomes sub-saturated after entrainment and the dry ascent;

– The calculation of forced detrainment allows an ensemble of plumes with a distribution of buoyancies to be represented by a single convective plume, for which only the mean buoyancy is explicitly calculated (Derbyshire et al., 2011). The convective ascent is therefore terminated when the forced detrainment reduces the mass flux to a small value, i.e. when the large majority of the implied ensemble of plumes have terminated after becoming negatively buoyant. To that end, the ascent in the 6A scheme will terminate when the mass flux falls below 5% of its value at cloud base, which replaces the arbitrary small value in the 5A scheme, or when the forced detrainment needs to detrain more than 95% of its mass. As before, shallow convection will terminate at the top of its diagnosed parcel ascent;

– Building on the convective "safety checks" introduced at GA6, the 6A convection scheme applies additional checks to ensure that the parcel ascent is valid. In particular, the cloud base mass flux after closure needs to be greater than zero, convection needs to be at least 3-levels thick and there needs to be at least some latent heat release during the ascent (i.e. purely dry convection is not permitted);

– A turbulent heating term to account for the loss of kinetic energy due to the convective momentum transport is added, similar to that described for sub-grid orographic drag in Sect. 3.5;

– We introduce a local correction to conserve water and energy in the column. Without this, the convection scheme will introduce small errors in the water and energy budgets from *(i)* truncation error in the discretisation and *(ii)* the convection scheme assuming hydrostatic balance and a shallow atmosphere (i.e. that the horizontal area of the grid boxes do not increase with height) whilst the full model does not. To account for these small errors, the 6A scheme applies a correction to total column water and energy to ensure that the column integrals of these quantities is the same after the call to convection as they were before.

The 6A convection scheme diagnoses convective ascents that are usually deeper than those from the 5A scheme. It also removes occasional vertical grid-scale noise in its increments to the model's prognostic fields. This is demonstrated in the



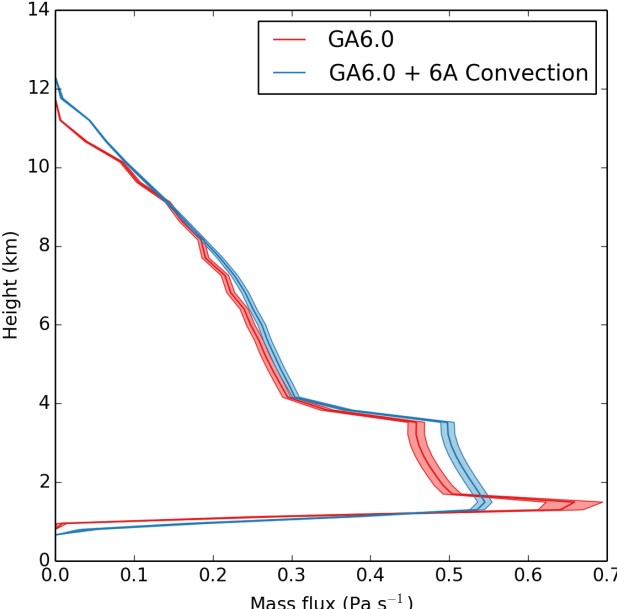

**Figure 4.** Profiles of mean convective mass flux from a pair of 100 member SCM ensembles using the GA6 configuration and the 6A convection scheme. The initial profiles and forcings are for an idealised diurnal cycle over tropical land (described in detail as experiment "r76" in Table 1 of Stirling and Stratton (2012)). The profiles are for 4 h means centred on local noon; the thick lines show the time-averaged ensemble-mean and the shading shows ±2 times the standard error in the ensemble-mean.

results of the SCM simulations presented in Fig. 4. Both of these affects are due primarily to the iterative convective parcel ascent calculations. Deeper convection translates into the tops of the deepest tropical cloud being beneficially higher. Figure 5 shows a comparison of the height of tropical cloud in an N96 atmosphere/land-only climate simulation and in CALIPSO observations, which shows that the cloud top height in the 6A convection scheme is in better agreement with observations than

5 in GA6; a similar improvement is seen throughout the tropics. Note, however, that this does not affect errors in other aspects of the cloud simulation, for which we rely on the other changes in GA7 (Williams and Bodas-Salcedo, 2017). Finally, in addition to the changes itemised above, the 6A convection scheme includes an amount of code tidying and refactoring that does not lead to any scientific differences, but means that this provides as a more suitable baseline for future development.

**CAPE closure for deep & mid-level convection dependent on large-scale vertical velocity (GA ticket #84)**

10 Prior to GA7, the deep and mid-level convection schemes used a fixed CAPE timescale, only shortened if very high vertical velocities were detected in a column. As discussed in Sect. 4.1.1 of Walters et al. (2017), the choice of a fixed timescale to suit both NWP and climate needs has been difficult, with NWP favouring a shorter timescale that improves the predictive skill of the model, and climate modellers preferring a longer timescale that reduces intermittent behaviour and improves the





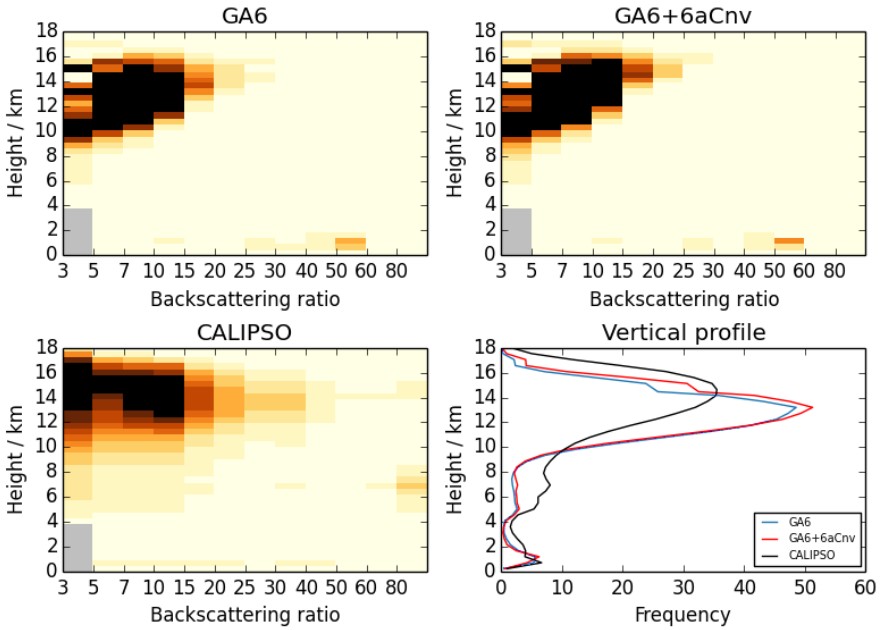

**Figure 5.** Histograms of height vs 532 nm lidar backscatter ratio over the tropical warm pool, showing CALIPSO observations and simulated climatologies of CALIPSO data from 20 year N96 atmosphere/land-only climate simulations using GA6.0 and GA6.0 plus the 6A convection scheme.

mean climatology. The UM convection scheme uses the dilute CAPE from the buoyant convective ascent, so we require a CAPE timescale for the dilute CAPE. An analysis of a few days of data from two $1.5\,\mathrm{km}$ resolution simulations from the CASCADE project (Pearson et al., 2014), one over West Africa and the other over the Indian Ocean, enabled us to investigate the deep convection by coarse gridding the data to $(150\,\mathrm{km})^2$ and $(30\,\mathrm{km})^2$ by area-averaging 100x100 or 20x20 grid points respectively. The fraction of buoyant and cloudy grid points and their properties relative to the mean were calculated, enabling us to estimate the dilute CAPE for deep convection. Figure 6 shows the distribution of CAPE timescales. It is clear that the CAPE timescale is not fixed and varies with resolution, with longer timescales being more frequent at coarser resolution; this explains why finding a fixed CAPE timescale for a model used across a range of resolutions is not easy. Our analysis of the data found a relationship between the mass-weighted mean vertical velocity and the dilute CAPE time scale such that

$$\tau_{\mathrm{CAPE}} = a w_{\mathrm{LS}}^{b}, \tag{9}$$

where $w_{\mathrm{LS}}$ is the mass-weighted mean vertical velocity over the depth of the deep or mid-level convection, and $a = 0.08$, $b = -0.7$ derived from a fit to the CASCADE data. The model $w_{\mathrm{LS}}$ can be negative or very small giving a very long CAPE timescale. In this case, for the deep convection scheme, an upper limit is derived from the surface-based closure used in the





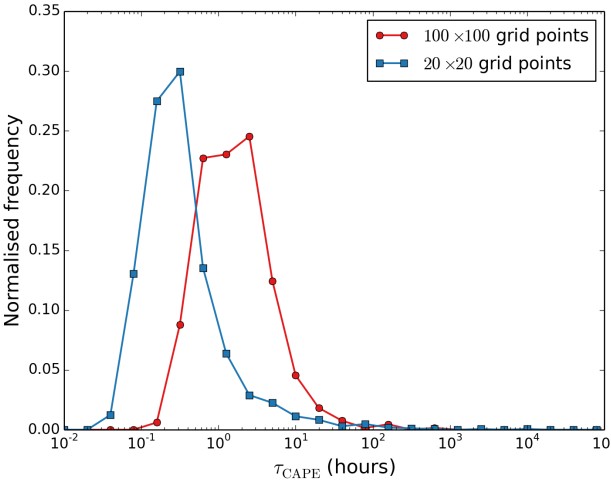

**Figure 6.** Normalised frequency distribution of convective timescale of deep convection from analysis of a small region of the Indian Ocean at resolutions of $100 \times 100$ and $20 \times 20$.

shallow scheme (Grant, 2001). In the mid-level scheme, an upper limit of $4\,\mathrm{h}$ is used. Both deep and mid-level convection have a lower limit set to the model convection time step.

### 3.8 Atmospheric aerosols and chemistry

**Introduction of the UKCA GLOMAP-mode aerosol scheme (GA ticket #60)**

In previous GA configurations, tropospheric aerosol was treated using either prognostic aerosol simulations or monthly mean climatologies from the CLASSIC aerosol scheme (Bellouin et al., 2011). In GA7, the simulation of prognostic aerosol is performed using the Global Model of Aerosol Processes (GLOMAP-mode[8], Mann et al. (2010)), with additional developments described in Mulcahy et al. (in prep.a). Glomap-mode is included in the UM as part of the UKCA coupled chemistry and aerosol code. Whilst CLASSIC was a simple mass-based bulk aerosol scheme, GLOMAP-mode models the aerosol number,

size distribution, composition and optical properties from a more detailed, physically-based treatment of aerosol microphysics and chemistry. This is expected to improve the representation of aerosol radiative effects and aerosol-cloud interactions (e.g. Bellouin et al., 2013) and was viewed as a requirement in GA7 so that GLOMAP-mode could be used as the atmospheric aerosol component of the UK's next Earth system Model, UKESM1.

   Speciated aerosol mass and number are simulated in 5 variable-size modes representing soluble nucleation, Aitken, accu-

15 mulation and coarse size ranges as well as an insoluble Aitken mode. As outlined in Sect. 2.10, the prognostic aerosol species represented are sulphate ($SO_4$), black carbon (BC), organic carbon (OC) and sea salt (SS) with species within each mode treated as an internal mixture. The size ranges covered by and aerosol species contributing to each mode are illustrated in

---

[8]GLOMAP-mode is included in the UM code base as part of the UK Chemistry and Aerosol (UKCA) code.





Table 3. The variable size distribution allows the median dry radius of each mode to change within these size ranges, whilst the standard deviation, $\sigma_g$, of each mode is fixed. In full chemistry-aerosol simulations such as those used for Earth system mod-

**Table 3.** The aerosol size distribution in GLOMAP-mode including aerosol modes represented, the range of radii these include, their geometric standard deviation and aerosol species contributing to each mode. Species represented are sulphate, black carbon, organic carbon and sea salt.

| Aerosol Mode | radii (nm) | $\sigma_g$ | Species |
|---|---|---|---|
| Nucleation sol. | $0 - 5$ | 1.59 | $SO_4$, OC |
| Aitken sol. | $5 - 50$ | 1.59 | $SO_4$, BC, OC |
| Accumulation sol. | $50 - 250$ | 1.40 | $SO_4$, BC, OC, SS |
| Coarse sol. | $250 - 5000$ | 2.00 | $SO_4$, BC, OC, SS |
| Aitken insol. | $5 - 50$ | 1.59 | BC, OC |

elling, aerosol precursor gases required for the production of $SO_4$ and secondary organic aerosols are provided by the UKCA stratospheric-tropospheric chemistry scheme (Morgenstern et al., 2009; O'Connor et al., 2014). In physical model simulations,

such as in the GA/GL climate simulations presented below, we use a simplified offline oxidant chemistry scheme (Mulcahy et al., in prep.a) in which the required chemical oxidant fields (such as $O_3$, OH, $NO_3$ and $HO_2$) are provided as monthly mean climatologies derived from an online chemistry simulation. GLOMAP-mode can simulate aerosol microphysical processes such as the nucleation of $SO_4$ aerosol, cloud processing, mode-merging, coagulation within and between modes and condensational growth of existing particles due to uptake from gas phase sulphuric acid and secondary organic vapours. Aerosol water

content is simulated prognostically, which combined with the internal mixing of aerosols within modes leads to a more realistic treatment of the aerosol optical properties than the previous CLASSIC scheme. The direct radiative impact of the aerosols is modelled using the UKCA-Radaer scheme outlined in Sect. 2.3. Cloud condensation nuclei are activated into cloud droplets using the UKCA-Activate aerosol activation scheme (West et al., 2014). In addition to the GLOMAP-mode species listed in Table 3, we continue to model mineral dust separately using the CLASSIC dust scheme (Woodward, 2011), although a modal

framework for the emission of mineral dust is being developed for future implementation.

One of the aerosol optical properties that is well observed in both ground-based and satellite observations is the aerosol optical depth (AOD). Figure 7 illustrates that the annual mean climatology of AOD in an N96 GA7/GL7 atmosphere/land-only climate simulation matches these observations well in both the mean value and its geographical distribution. A more detailed evaluation of the AOD including a comparison against the CLASSIC aerosol scheme included in GA6 is given in

Mulcahy et al. (in prep.a). To illustrate the suitability of using GLOMAP-mode simulations to diagnose the concentration of cloud condensation nuclei, Fig. 8 compares observed and simulated monthly mean aerosol number concentration of particles with diameter greater than $50\,\mathrm{nm}$ (N50) from the same simulation presented in Fig. 7a. The simulated aerosol number is in reasonable overall agreement with the observations, with approximately 60% of the data agreeing within a factor of 2. The





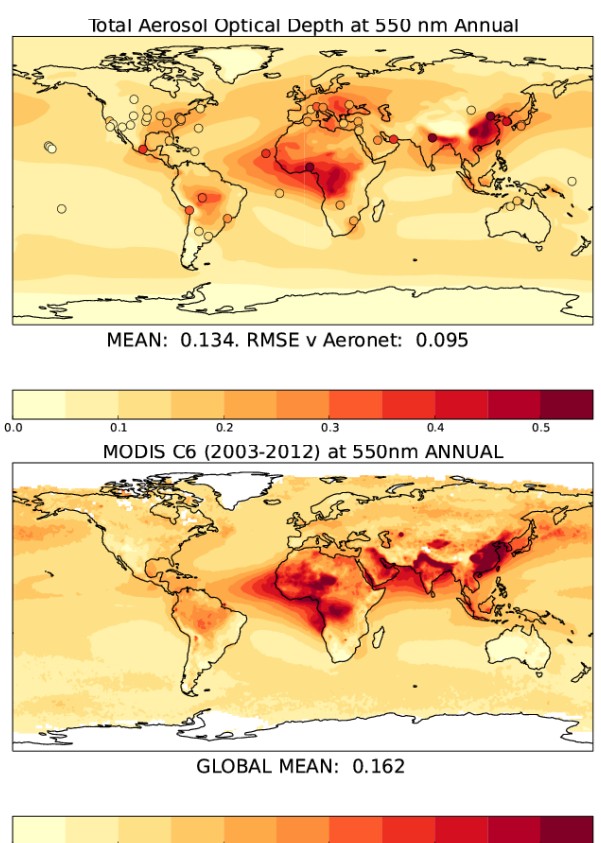

**Figure 7.** Annual mean AOD at 550 nm from a 20 year N96 atmosphere/land-only GA7/GL7 climate simulation (top) and satellite observations from the MODerate Resolution Imaging Spectrometer (MODIS) (bottom). The annual mean MODIS observations are derived from monthly mean level 3 products from MODIS AQUA collection 6 (Sayer et al., 2014) and cover the period 2003–2012. An annual mean climatology of Level 2 AOD observations at 500 nm from the ground-based AERONET sun photometer network (Holben et al., 2001) at 67 worldwide locations are overlaid on the GA7 spatial plot.

regional breakdown highlights good performance in relatively clean air regions such as the Arctic, Antarctica and ocean basins, although the model underestimates N50 in more polluted regions such as North America, Europe and Asia.

The two moment modal scheme represents a significant increase in complexity versus the single moment CLASSIC scheme. This additional complexity leads to improvements in the aerosol simulation, but does increase the computational cost of the model, with the run-time to solution in an atmosphere/land-only climate simulation at N96 resolution increasing by about 50%. In future, it may be desirable to develop a traceable hierarchy of aerosol complexity within the GLOMAP-mode framework, which would reduce the expense of some physical climate model simulations and allow implementation in systems not currently using prognostic aerosol such as NWP and seasonal forecasting systems. Another temporary limitation of GLOMAP-mode — and its interaction with the UM physics via Radaer and Activate — is that the code does not yet work with climatological





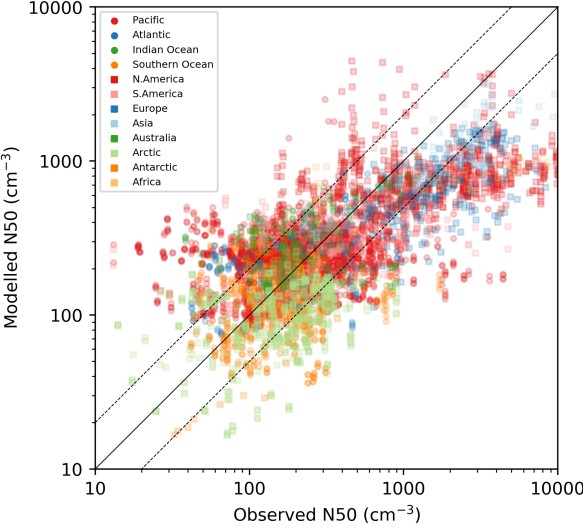

**Figure 8.** Monthly mean simulated N50 from the simulation presented in Fig. 7 versus Global Aerosol Synthesis and Science Project (GASSP, see references for details) observations from 2008. GASSP is a harmonised set of in situ measurements from ground-based networks, ship and aircraft campaigns, which we have regridded onto the model grid for the purpose of this comparison. Different colours and symbols denote data in different regions.

aerosol fields, which we currently require in these systems without prognostic aerosol. In these simulations, therefore, we continue to use climatologies from the CLASSIC aerosol scheme and the calculation of optical properties and cloud droplet concentrations that were used in GA6 as described in Sect. 2.3 of Walters et al. (2017). Whilst this is far from ideal, in parallel climate simulations using prognostic GLOMAP-mode and climatological CLASSIC aerosols (not shown, for brevity), the
5  model's climatological radiation fluxes are broadly similar. This traceability between prognostic GLOMAP-mode simulations and CLASSIC climatologies was an imposed constraint on the implementation of GLOMAP-mode in GA7 that will allow us to continue to support a seamless Global Atmosphere configuration across a variety of modelling systems. This constraint on further development of the configurations can eventually be relaxed once we can use aerosol climatologies derived from GLOMAP-mode in Radaer and Activate.

10  ### 3.9 Land surface and hydrology: Global Land 7.0

**Introduction of the multi-layer snow scheme (GL ticket #4)**

In GL6 and before, JULES used a simple snow scheme, described in Best et al. (2011), in which lying snow and the topmost soil layer are represented as a single thermal layer; because the snow in this scheme had no independent thermal store, this is labelled as a "zero-layer" scheme. This was a major deficiency in the configuration, as it allowed the surface layer of the





atmosphere direct access to heat within the soil and thus poorly represented the insulating effect of the snow pack in the real world (Slater et al., 2017). In GL7, we introduce a new snow scheme, based on the multi-layer scheme also described in Best et al. (2011). In standalone land surface simulations, where JULES is driven by near-surface meteorological fields, soil tempertures and permafrost extent are substantially improved by using the multi-layer scheme (Burke et al., 2013). However, the

additional degrees of freedom introduced by coupling the scheme to an atmospheric model impose more stringent constraints on the parametrisation and have led to a number of enhancements being introduced to improve the scheme in the coupled atmosphere-land system.

The multi-layer snow scheme works by accumulating snow in the top-most snow layer, until this reaches a specified maximum thickness, when the layer is split into two. The maximum permitted number of snow layers is set to 3, which earlier tests

suggested was sufficient to represent the snow pack with reasonable fidelity. This thickness of the first and second snow layers are set to $0.04$ and $0.12\,\mathrm{m}$ respectively, which is reduced from the values of $0.1$ and $0.2\,\mathrm{m}$ in the original scheme. Any snow beyond the combined $0.16\,\mathrm{m}$ thickness of the first two layers is held in the third layer. For reasons of numerical stability, the original zero-layer scheme is still used when the thickness of the snow pack is less than the thickness of the first snow layer. The reduction of this layer from $0.1\,\mathrm{m}$ to $0.04\,\mathrm{m}$, therefore, reduces this use of the zero-layer scheme and allows the snow pack

to respond more rapidly to changes in atmospheric conditions.

The density of fresh snow is set to $109\,\mathrm{kg\,m^{-2}}$, following Vionnet et al. (2012), but omitting their dependence on temperature and wind speed. The thermal conductivity is parametrised using the formula given by Calonne et al. (2011), replacing that of Yen (1981). Unlike the original parametrisation, this improved formula includes the affect of thermal conduction through the air in the snow pack, which is significant in newly fallen snow. Although the original scheme allows for temperature-dependent

mechanical compaction of the snow, temperature gradient metamorphism is not included. In the GL7 implementation, we introduce a parametrisation for rapid densification of fresh snow by equi-temperature metamorphism (i.e. the rounding of irregular snow grains due to the vapour pressure being higher on convex than on concave surfaces) based on Anderson (1976). This causes a significant increase in the density of fresh snow on the timescale of a few days and omitting it would result in prominent cold biases across the northern hemisphere. The effect of temperature gradient metamorphism is still omitted,

however, which remains a topic for future research. A wind-speed dependent rate of unloading of snow from needle-leaved tress has been introduced and adjusted to give a typical unloading timescale of $2\,\mathrm{days}$ in the Canadian boreal forest (MacKay and Bartlett, 2006). Rain water and melt water from snow on the canopy are also allowed to infiltrate into the snow pack, rather than bypassing it, as in the original version of the scheme. Finally, a new parametrisation of snow albedo, based on a two-stream model of radiative transfer through the snow pack and a prognostic snow grain size, has also been introduced.

The largest impact of the new scheme on the model's climatology comes from its insulation of the soil beneath the snow pack. Figure 9 shows that in December–February (DJF), the near-surface air over the northern hemisphere snow-pack is generally colder, as the heat flux from the soil into the atmosphere is reduced, whilst in March–May (MAM) the air is warmer, as the snow-melt over warmer soils leads to less cooling from below. It also shows that the annually integrated affect of this additional insulation is a significantly warmer soil layer, which improves a long-standing model bias and is expected to be highly desirable

for the simulation of permafrost.



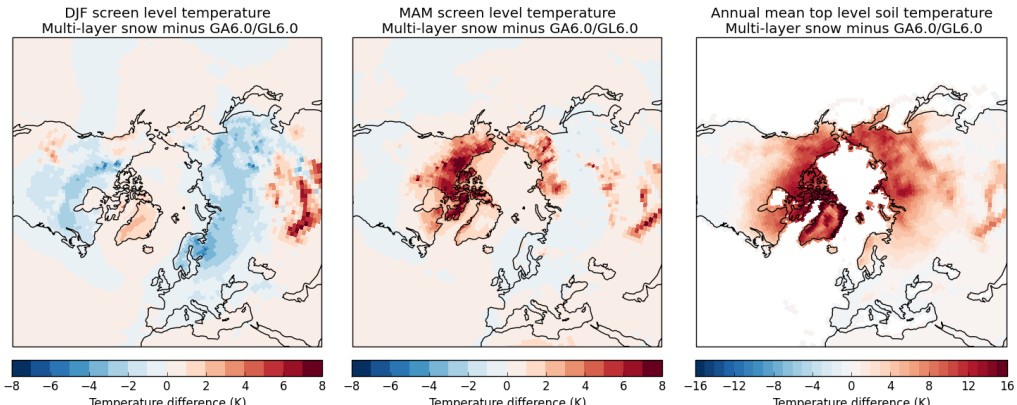

**Figure 9.** The difference in mean DJF screen-level temperature (left), mean MAM screen-level temperature (centre) and annual mean top-most level soil temperature (right) from a 20 year N96 atmosphere/land-only climate simulation with and without GL7's multi-layer snow scheme. The baseline simulation is GA6.0/GL6.0.

**Further improvements to land surface albedo (GL ticket #30)**

In both GL6 and GL7, the snow-free SW albedo of the land surface is calculated for each grid box, which is a weighted average of the albedos of the different surface types. The albedo of bare soil is input as ancillary data, described in Section 2.14, whilst the other non-vegetated surface types have a constant "first guess" albedo. The albedo of vegetated surface types varies between
5 the bare soil albedo and a full leaf albedo depending on the leaf area index (LAI) of that vegetation type. Additionally, when the land surface properties are fixed — rather than evolving as occurs in Earth system simulations using interactive vegetation — the albedos of each tile are scaled, within physical limits, so that the grid box mean snow-free albedo matches that of an observed climatology.

In GL6, these albedos were independent of wavelength, whilst in GL7, a limited spectral variability has been added by
10 applying separate calculations for the photosynthetically active radiation (PAR) and Near-InfraRed (NIR) using the canopy radiation model of Sellers (1985). When the albedos are scaled, they are compared to climatologies of the grid box mean PAR and NIR albedos, and the input albedos or leaf reflectivity and scattering coefficients are changed accordingly using a predictor-corrector step to account for the slight non-linearity in the canopy radiation model. Again, the albedos and reflectivities are kept within physical limits.

Alongside this change, the minimum albedo of the urban tile was reduced from 0.16 to 0.05 to better match observed albedos in urban areas. Finally, the generation of the land surface type data was slightly amended; this process takes the 17 land surface types from the the International Geosphere-Biosphere Programme dataset (IGBP) (Global Soil Data Task, 2000) and maps them to the 9 land surface types used in JULES. In GL7, the amount of bare soil present in the Grassland, Cropland and Crop/Natural Mosaic IGBP classes was reduced as we believe that the original mappings used increased bare soil values to





account for seasonally barren vegetation, which is now accounted for in the time-varying LAI. The mappings used in GL7 are shown in Table 4.

**Table 4.** Mapping from IGBP classification to JULES land surface types in GL7 (%). The abbreviated headers are broadleaf trees, needle-leaved trees, temperate C3 grass, tropical C4 grass, shrubs, urban areas, inland water, bare soil and land ice surface types respectively. Values that have changed are marked in bold with GL6 values in brackets.

| IGBP code | IGBP name | BL | NL | C3g | C4g | SH | UR | IW | BS | LI |
|---|---|---|---|---|---|---|---|---|---|---|
| 1 | EN forest | 0.0 | 70.0 | 20.0 | 0.0 | 0.0 | 0.0 | 0.0 | 10.0 | 0.0 |
| 2 | EB forest | 85.0 | 0.0 | 0.0 | 10.0 | 0.0 | 0.0 | 0.0 | 5.0 | 0.0 |
| 3 | DN forest | 0.0 | 65.0 | 25.0 | 0.0 | 0.0 | 0.0 | 0.0 | 10.0 | 0.0 |
| 4 | DB forest | 60.0 | 0.0 | 5.0 | 10.0 | 5.0 | 0.0 | 0.0 | 20.0 | 0.0 |
| 5 | Mixed forest | 35.0 | 35.0 | 20.0 | 0.0 | 0.0 | 0.0 | 0.0 | 10.0 | 0.0 |
| 6 | Closed shrub | 0.0 | 0.0 | 25.0 | 0.0 | 60.0 | 0.0 | 0.0 | 15.0 | 0.0 |
| 7 | Open shrub | 0.0 | 0.0 | 5.0 | 10.0 | 35.0 | 0.0 | 0.0 | 50.0 | 0.0 |
| 8 | Woody savannah | 50.0 | 0.0 | 15.0 | 0.0 | 25.0 | 0.0 | 0.0 | 10.0 | 0.0 |
| 9 | Savannah | 20.0 | 0.0 | 0.0 | 75.0 | 0.0 | 0.0 | 0.0 | 5.0 | 0.0 |
| 10 | Grassland | 0.0 | 0.0 | **85.0** (70.0) | **10.0** (15.0) | 5.0 | 0.0 | 0.0 | **0.0** (10.0) | 0.0 |
| 11 | Permnt wetland | 0.0 | 0.0 | 80.0 | 0.0 | 0.0 | 0.0 | 20.0 | 0.0 | 0.0 |
| 12 | Cropland | 0.0 | 0.0 | **85.0** (75.0) | 5.0 | 0.0 | 0.0 | 0.0 | **10.0** (20.0) | 0.0 |
| 13 | Urban | 0.0 | 0.0 | 0.0 | 0.0 | 0.0 | 100.0 | 0.0 | 0.0 | 0.0 |
| 14 | Crop/nat mosaic | **7.5** (5.0) | **7.5** (5.0) | **60.0** (55.0) | 15.0 | 10.0 | 0.0 | 0.0 | **0.0** (10.0) | 0.0 |
| 15 | Snow and ice | 0.0 | 0.0 | 0.0 | 0.0 | 0.0 | 0.0 | 0.0 | 0.0 | 100.0 |
| 16 | Barren | 0.0 | 0.0 | 0.0 | 0.0 | 0.0 | 0.0 | 0.0 | 100.0 | 0.0 |
| 17 | Water bodies | 0.0 | 0.0 | 0.0 | 0.0 | 0.0 | 0.0 | 100.0 | 0.0 | 0.0 |

### Improved parametrisation of the ocean surface albedo (GL ticket #43)

GL6 parametrised the ocean surface albedo (OSA) with the method of Barker and Li (1995), in which OSA is a function solely of the solar zenith angle, accounting for the glitter of the sea surface (Cox and Munk, 1954). In addition to this, hard-wired scaling factors were applied for each of the SW bands in Table 1 to enforce a spectral variation.

GL7 implements the OSA parametrisation of Jin et al. (2011), which has the advantage of including an additional dependence on wavelength as well as on the 10 m wind speed and chlorophyll content. The wavelength and chlorophyll content dependency of the optical properties are taken from lookup tables of data from Jin et al. (2011).

The wind speed dependency represents two effects. Firstly, an increased wind speed leads to an increase in the number and size of ocean waves on the sea surface, which in turn increases the variability of the incident angle of reflection at the surface (sea surface 'glitter'). This leads to a higher albedo at low solar zenith angles and a lower albedo it at higher zenith angles.





Secondly, an increased wind speed leads to the breaking of waves, which create whitecaps; these are represented by assigning a fraction of the model grid box as ocean foam with an albedo of 0.55. Finally, the spectral variability across the SW bands in Table 1 is represented by calculating the average of the OSA for 50 wavelengths within each band.

The chlorophyll content is prescribed via a new ancillary field, which contains a periodic monthly climatology based on
GlobColour ocean colour data. The GlobColour products merge ocean colour observations from the MERIS, MODIS-Aqua and SeaWiFS sensors, and derive sea surface chlorophyll-a concentration (Maritorena et al., 2010). The climatology was created by averaging the GlobColour data from 1998–2007 onto a $1\,^{\circ} \times 1\,^{\circ}$ grid. An extrapolation was then performed to fill grid points with no satellite observations. The creation of these climatology files is described in Ford et al. (2012).

**Use atmospheric rain fractions in surface hydrology (GL ticket #45)**

In previous GL configurations, the fraction of a grid box over which rain was assumed to fall on the land-surface was set to 1.0 for large-scale rain and 0.3 for convective rain (Best et al., 2011). This information is used in the calculations of canopy throughfall and evaporation, surface infiltration and runoff. Atmospheric changes introduced at GA7 have allowed these rain fractions to be more accurately calculated, and therefore we use this information consistently within the surface scheme. The large-scale rain fraction is now given by the scheme introduced in Sect. 3.3, whilst the convective rain fraction is given by the
maximum convective core area from the scheme introduced in Sect. 3.4.

**Implement surface roughness from the COARE4.0 Algorithm (GL ticket #31)**

In JULES, the momentum roughness length over the sea surface ($z_{0,\mathrm{m}}(\mathrm{sea})$) is given by

$$z_{0,\mathrm{m}}(\mathrm{sea}) = 0.11\frac{\nu}{u_*} + \alpha_{\mathrm{ch}}\frac{u_*}{g}, \qquad (10)$$

where $u_*$ is friction velocity. The first term, dominant in low-wind conditions, accounts for the kinematic viscosity of air
$\nu$ (Smith, 1988), whilst the second term is the Charnock relation in which $\alpha_{\mathrm{ch}}$ is the Charnock parameter (Charnock, 1955) and g is the acceleration due to gravity. Previous GL configurations used a constant Charnock parameter of $\alpha_{\mathrm{ch}} = 0.018$; observational evidence, however, suggests that younger waves are rougher (Donelan et al., 1993) such that the Charnock parameter should be correlated with the wind speed. Such a parametrisation is implemented in the COARE (Coupled Ocean–Atmosphere Response Experiment) family of bulk flux algorithms (Edson et al., 2013), so in GL7 we implement the roughness parametri-
sation used in COARE4.0 (Edson, 2009), in which Charnock parameter increases with wind speed up to $22\,\mathrm{ms}^{-1}$. Above this wind speed, the roughness length is capped as there are few observations and surface exchange at very high wind speeds is not well understood (Soloviev et al., 2014). JULES uses a single roughness length for scalars, for which we have adopted the COARE4.0 value for moisture, which is identical to that used in COARE3.0 (Fairall et al., 2003). Finally, note that we have only implemented the roughness lengths from the COARE algorithm and other aspects of surface exchange, such as the
similarity functions, follow those standard in JULES.

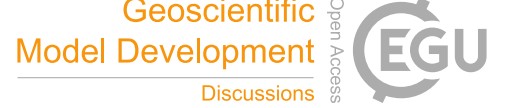



**Revised roughness lengths for sea ice (GL ticket #38)**

In the presence of sea ice, the roughness length of the non-land portion of a model grid box is taken as a weighted average of $z_{0,\mathrm{m}}(\mathrm{sea})$ (discussed above) and of fixed roughness lengths for pack ice and marginal ice, with the weightings taken as a function of the ice fraction. As discussed in Walters et al. (2017), for GL6.0, a pragmatic decision was made to adopt values
of $3.2\,\mathrm{mm}$ for pack ice and $100\,\mathrm{mm}$ for marginal ice, following earlier practice in operational weather forecasting. As further noted in that publication, these values are high compared to observational estimates, particularly for pack ice. Subsequent investigation has shown that performance is substantially unaffected if the value for pack ice is reduced to $0.5\,\mathrm{mm}$, which is more comparable with the observed values, whilst performance is degraded by reducing the value for marginal ice. In GL7.0, therefore, we use roughness lengths of 0.5 and $100\,\mathrm{mm}$ for pack ice and marginal ice respectively. Investigation of surface
exchange over marginal ice continues.

### 3.10    Stochastic physics

**Introduction of a standardised stochastic physics package (GA ticket #117)**

Prior to GA7, the use of stochastic physics schemes in the global UM was defined as a "system-dependent option", which remained outside of the definition of the Global Atmosphere configuration. This means that different ensemble prediction
systems (EPSs) have used different representations of stochastic physics as they have seen fit. The global component of the Met Office Global and Regional Ensemble Prediction System (MOGREPS-G, Bowler et al., 2008) has used the Stochastic Kinetic Energy Backscatter scheme version 2 (SKEB2, Tennant et al., 2011) to represent model error emerging from upscale transfers of energy from truncated or highly diffused scales. It has also used the Random Parameters scheme version 2 (RP2, Bowler et al., 2009) to represent the structural uncertainty of a parametrisation's key parameters, such as the convection scheme's
entrainment rate or coefficients in the gravity-wave drag. The Global Seasonal prediction system (GloSea, MacLachlan et al., 2014) has employed SKEB2 only, whilst to date, long-range climate projections have not used any stochastic physics schemes.

As discussed in Sect. 2.12, the role of stochastic physics in the performance of our prediction systems is increasing as the importance of EPSs across all timescales is growing and the impact of stochastic parametrisations on the model's climatology is better understood. For this reason, we have developed a standardised package of stochastic physics schemes that has been
included for the first time in GA7. The two main components of this package are an improved version of SKEB2 and the replacement of RP2 by the Stochastic Perturbation of Tendencies scheme (SPT), both of which are described in detail in Sanchez et al. (2016). The changes to SKEB2 are a replacement of the Smagorinsky numerical dissipation mask by a bi-harmonic formulation, which is more representative of the numerical dissipation caused by the semi-Lagrangian interpolation to the departure point in global models with horizontal resolutions of $O(10-100\,\mathrm{km})$. The convective mask also now includes a
resolution dependent coefficient, which reduces the impact of this term at higher resolutions. Both changes to SKEB2 improve its response to resolution, as with increased resolution, the model is less dissipative, so the error that SKEB2 is designed to overcome is itself reduced. This "scale awareness" is particularly improved in the tropics, where the original scheme was quite insensitive to resolution; in addition, at lower resolution, the changes lead to larger perturbations in the tropical free





troposphere and in the atmospheric boundary layer globally (see Sect. 3 of Sanchez et al. (2016)). The SPT scheme adds variability to the model parametrisations by perturbing their diagnosed tendencies, rather than internal parameters as is done in RP2. This is achieved by scaling increments in the radiation scheme, large-scale precipitation, sub-grid orographic drag and convective heating/moistening (although not convective momentum transport). Increments from the boundary layer scheme and

increments from all other schemes in the lowest 8 levels (i.e. below $\approx 660\,\mathrm{m}$) remain unscaled to maintain model stability. For similar reasons, the scaling is tapered between level 9 and level 15 (i.e. between $\approx 660\,\mathrm{m}$ and $1700\,\mathrm{m}$) and large perturbations are capped in grid boxes with large standard deviations in the sub-grid orography field. The potential impact of SPT is greater than that of RP2 as it can add fluctuations to represent sub-grid and structural uncertainties of the parametrisations rather than just including uncertainty in the inputs to deterministic parametrisations. The stochasticity of both SKEB2 and SPT is applied

via a first order autoregressive forcing pattern with a $6\,\mathrm{h}$ decorrelation time scale and applied spatially between horizontal wavenumbers 20 and 60. Finally, to allow SPT to be used in long climate integrations, where the conservation of energy and moisture are important, an additional constraint is applied to the scheme to conserve both water vapour and moist static energy in the column as described in Appendix B of Sanchez et al. (2016).

    The stochastic physics package has been tested in a 1-month trial (run from mid-November to mid-December 2012) of

short-range (3 day) ensemble forecasts with 12 members per 6-hourly cycle run at N216 horizontal resolution ($\approx 60\,\mathrm{km}$ in the mid-latitudes). The control ensemble uses GA6.1/GL6.1 (Walters et al., 2017) with the operational SKEB2 and RP2 stochastic physics, whilst the test replaces this with the standardised stochastic physics package described above. The largest impact of the new stochastic physics package is in the tropics. Figure 10 shows spread/skill relationships for $850\,\mathrm{hPa}$ tropical temperature and wind speed (i.e. the comparison of the ensemble control and ensemble mean root mean square errors (RMSEs) versus

radiosondes compared to the internal spread of the ensemble). In an ideal ensemble, the spread of the ensemble and the RMSE of the ensemble mean should match. For both parameters, the GA7 stochastic physics package shows increased spread without increasing the ensemble mean error, although the difference between the spread and error is still quite large in this small and simple ensemble test. However, the new configuration does produce slightly worse probabilistic scores on the mid-latitude boundary layer (not shown), as neither SKEB2 or SPT include perturbations to near-surface fields, which are perturbed in the

boundary layer parameters included in RP2. In climate simulations, the use of stochastic physics improves some long-standing climate biases, particularly, in the tropics, which indicates that the coarse representation of sub-grid fluctuations has a beneficial impact on the model's tropical climatology. Figure 11, which is reproduced from Sanchez et al. (2016), shows the reduction in errors in June–August (JJA) outgoing LW radiation (OLR) with the stochastic physics package at a number of resolutions. That publication also describes improvements to other tropical errors, such as precipitation and circulation, which is consistent with

the tropical improvements seen in other systems such as the European Centre for Medium-Range Weather Forecasts (ECMWF) Integrated Forecasting System (Weisheimer et al., 2014).

    Finally, whilst stochastic schemes such as SKEB2 and SPT aim to provide a reasonable representation of the main sources of model error and thus produce a better distribution of weather states, they do so by inflating the variability in the current over-deterministic formulations. Whilst this leads to improvements in the reliability of an EPS or a climate simulation, these schemes

are deficient in their representation of individual processes such that they will degrade the evolution of an individual forecast



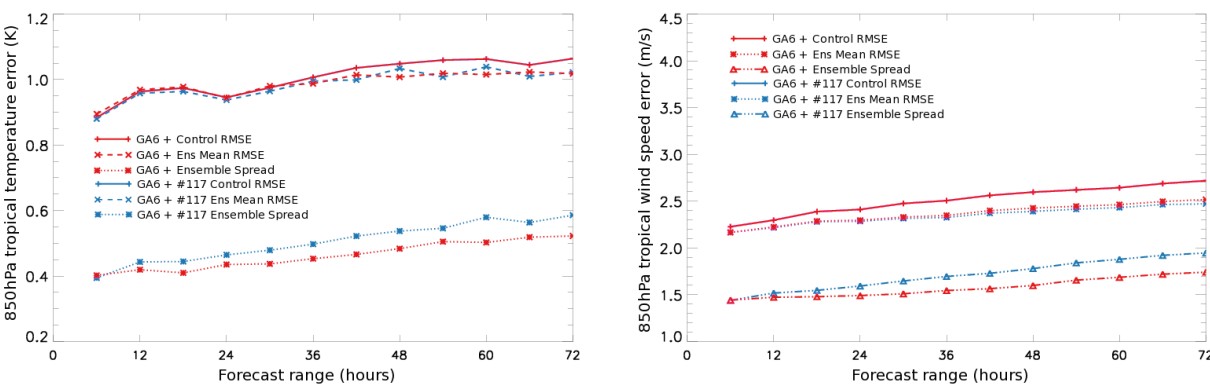

**Figure 10.** Ensemble error and skill for tropical temperature (left) and wind speed (right) versus radiosonde observations at $850\,\mathrm{hPa}$ in an N216 ensemble with the operational MOGREPS-G and GA7 stochastic physics package.

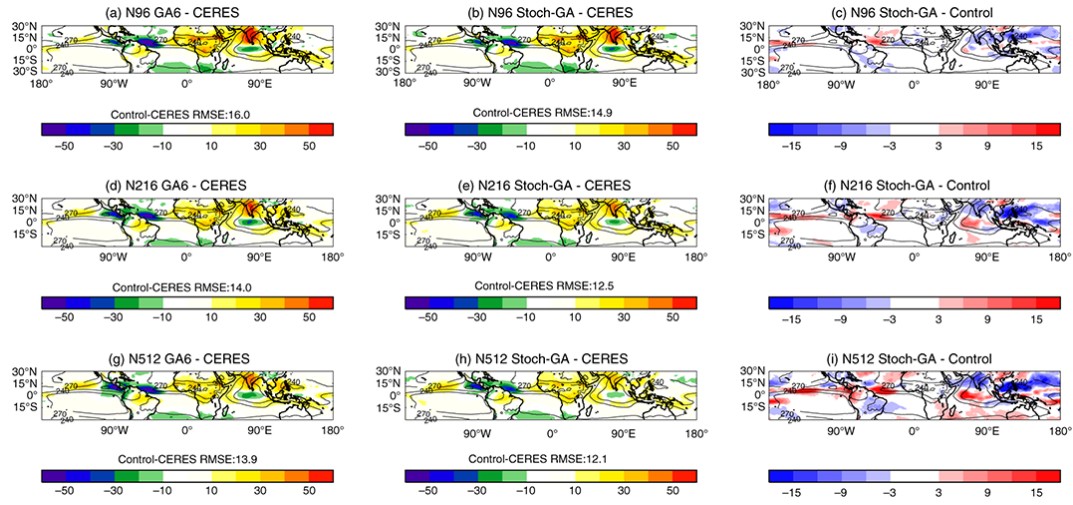

**Figure 11.** Mean June–August (JJA) top of atmosphere outgoing LW radiation (OLR) errors $(\mathrm{W\,m^{-2}})$ from $20\,\mathrm{year}$ atmosphere/land-only climate simulations compared to CERES EBAF (Clouds and the Earth's Radiant Energy System–Energy Balanced and Filled dataset, Loeb et al., 2009). The columns from left to right are GA6.0 controls, GA6 plus the stochastic physics package (labelled Stoch-GA) and the difference between two, whilst the rows from top to bottom are from simulations at N96, N216 and N512 resolution. This figure is reproduced from Sanchez et al. (2016).

event such as the trajectory of an individual mid-latitude cyclone (Sanchez et al., 2013). For this reason, the stochastic physics package is not currently planned for inclusion in deterministic forecast systems and the use (or lack of) stochastic physics remains a "system dependent option"; when a system uses GA7 with stochastic physics, however, it will now be expected to



use the schemes described above. In the longer term, we expect the use of more physically based parametrisation schemes with implicit stochastic elements — such as those presented in Eckermann (2011), Plant and Craig (2008) or Bengtsson et al. (2013) and Bengtsson and Körich (2016) — to overtake the use the current, more ad-hoc schemes. We would expect these more physically based stochastic schemes to be used in all prediction systems.

## 3.11 Tuning of the configuration

Tuning plays an important part in the process of developing any atmospheric model. Individual parametrisations will include input parameters that must be constrained using either direct observational estimates, evidence based on other modelling studies or by constraining their outputs to fall within the bounds of observational or theoretical uncertainty. For coupled climate models in particular, the balance of certain budget terms, such as the mean incoming and outgoing SW and LW radiation fluxes at the top of the atmosphere (TOA), need to balance to a precision that is greater than the uncertainty in its individual components to stop the model drifting into an unrealistic state (this is described briefly in Box 9.1 in Chapter 9 of Working Group I's contribution to the Intergovernmental Panel on Climate Change fifth assessment report: IPCC, 2013). Recently, the topic of tuning, the parameters tuned and the constraints that they are tuned to has received some attention in the literature (e.g. Hourdin et al., 2017; Schmidt et al., 2017) and is now accepted as an important aspect of model development that should be documented as part of a model description.

Here, we outline the tuning that took place in the development of the GA7.0 configuration. Because the development of the Global Atmosphere configurations is incremental, we limit this description to developments that took place in the increment documented herein; we do not document the tuning that took place in the development of GA6 or before, but the approach used in those developments was consistent with that described here. As discussed above, the bottom-up development of individual parametrisation schemes will also include a certain amount of parameter tuning and it is assumed that this is documented in the publications describing those parametrisations. Where we have deliberately altered an input parameter in an existing or proposed parametrisation, however, we do include this here.

The majority of tuning performed in the development of the GA configurations is motivated by taking a subjective overview of a large number of objectives measures. These primarily consist of NWP verification scores measured against observations and own/independent analyses, and climate metrics derived from comparing a large basket of mean fields and modes of model variability from present day climate simulations with observational and reanalysis datasets. Currently, the Met Office does not run climate change simulations which involve feedbacks (such as historical simulations with time-evolving forcing) during its model development process. It does, however, perform effective radiative forcing tests. Proposed scientific developments to the configuration have their individual impact tested by running low-resolution present day atmosphere/land-only simulations on both NWP and climate time scales. The impact of each change is assessed and recorded and any unexpected or severely detrimental impact may lead the GA development team to reject a change. At this stage, we might recommend the retuning of individual parameters, although no such tuning was performed on the changes accepted into GA7.

In the next stage of development, we study the combined impact of changes by building up collections (or packages) of changes into development configurations and assess these at both NWP and climate time scales. In the development of these





packages, and in particular in the final stages before the definition of the final configuration, there are two schemes that are routinely retuned because of their sensitivity to both uncertain input parameters and changes in other aspects of the model. The mineral dust emission scheme is highly sensitive to changes in surface roughness or near-surface winds. Whilst the parametrisation described in Woodward (2011) is based on detailed studies of the process of mineral dust lofting, the mapping of the

model's grid-box and time-step mean values of parameters such as the near-surface friction velocity or upper-most soil level moisture content onto point-like, instantaneous, surface layer values relies on the use of arbitrary scaling parameters. These are tuned to emergent observable quantities such as dust aerosol optical depth and near-surface concentrations in instrumented locations. Similarly, for the non-orographic gravity-wave drag (GWD) scheme described in Sec. 2.7, there is no a priori estimate for the amplitude of launched gravity waves required to represent the breaking of sub-grid waves and those on larger

scales that have been unrealistically damped by the model's large-scale dynamics. Instead, this it tuned so that the period of the model's QBO matches that observed in reanalyses. The tuning applied to the dust scheme and the non-orographic GWD scheme is describe in Table 5.

**Table 5.** Original GA6.0 and final GA7.0 values of parameters in schemes routinely tuned as part of GA development.

| Scheme | Variable | Parameter description | GA6 value | Tuned GA7 value |
| --- | --- | --- | --- | --- |
| Non-orographic GWD | `ussp_launch_factor` | Multiplicative tuning used to set the "launched spectrum-scale factor" compared to the original coded value of $3.42 \times 10^{-9}\,\mathrm{s}^{-2}$. | 1.2 | 1.3 (N96) 1.2 (N216+) |
| Dust emission | `horiz_d` | Global (multiplicative) tuning for dust emission. | 2.50 | 2.25 |
| | `us_am` | Multiplicative tuning applied to diagnosed friction velocity on input to dust emission scheme. | 1.45 | 1.45 (N96) 1.40 (N216+) |

When package configurations contain errors that are deemed not to be acceptable in the final configuration, or appear to be due to an imbalance between individual changes in the package, we consider either retuning the model's input parameters

or where appropriate, pulling through additional developments proposed for later model upgrades that were not ready (or not considered high enough priority) when planning the scope of the current release. In this tuning, however, we still insist that parameters are altered only within the estimation of their uncertainty and with the agreement of the parametrisation developers. When the problem being addressed is one of balance (e.g. the balance between TOA incoming SW and outgoing SW and LW radiation) we try to improve the balance by altering the component of the budget that we believe is most in error, ideally in the

geographical region that has the largest local error. We try not to bring the model into balance by adjusting standard "tuning knobs" that affect these balances, such as a global scaling to model cloud amounts. In the development of GA7, one measure that caused particular concern was the climatological temperature and humidity bias in and above the tropical tropopause layer (TTL). Hardiman et al. (2015) discuss the importance of this bias and the role of various physical and numerical processes in



its development. In developing GA7, the introduction of consistent ice optical and microphysical properties (GA ticket #17) was originally proposed via the scheme described in Baran et al. (2014) and the detrimental impact on the TTL temperature bias led to the additional developments made in Baran et al. (2016). The inclusion of cubic Hermite interpolation for moist prognostic variables (as part of GA ticket #135) was motivated primarily by its reduction of moisture biases above the tropical

5  tropopause. Also, the reduction of the adaptive detrainment in the deep convection scheme included in Table 6 was motivated by its improvement in both TTL temperature and moisture biases.

**Table 6.** Initially proposed and final values of parameters tuned to improve cloud/radiation interactions in GA7.0.

| Scheme | Variable | Parameter description | Initial value | Tuned value |
|---|---|---|---|---|
| Boundary layer | forced_cu_fac | Scaling factor applied to in-cloud water content diagnosed for forced convective clouds in GA ticket #83. | 1.0 | 0.5 |
| | dec_thres_cu | Buoyancy flux threshold for cumulus sub-cloud layers in GA ticket #83. | Not used. | 0.05 |
| | zhloc_depth_fac | Fraction of the cloud layer through which to continue a test on the Richardson number, used to diagnose a "shear dominated" BL (Bodas-Salcedo et al., 2012). | 0.3 | 0.4 |
| Large-scale cloud | two_d_fsd_factor | 1D-to-2D conversion factor used in calculation of the fraction standard deviation ($f$) in GA ticket #15. | 1.414 | 1.5 |
| | cff_spread_rate | Cirrus spreading rate ($r$) discussed in GA ticket #98. | $1 \times 10^{-9} \mathrm{s}^{-1}$ | $1 \times 10^{-5} \mathrm{s}^{-1}$ |
| | mp_dz_scal | Scaling factor for the mixing length in GA ticket #120 (labelled $\beta_1$ in Furtado et al., 2016). | 2.0 | 1.0 |
| Convection | r_det | "Adaptiveness" of the detrainment scheme used for deep convection (Derbyshire et al., 2011). | 0.9 | 0.8 |
| | cca_sh_knob | Fraction of diagnosed shallow convective cloud amount passed to the radiation scheme to represent the convective core in GA ticket #44. | 0.5 | 0.2 |

All other parameters tuned in the development of GA7.0 are also included in Table 6. The majority of these changes were designed to reduce the reflectivity of northern hemisphere low cloud (and hence reduce present-day climatological biases in reflected SW radiation) whilst maintaining the improved reduction in a positive surface SW radiation bias in the southern

10  hemisphere (described in Sect. 4). The asymmetry in the SW radiation biases in both GA6 and GA7 is discussed in more detail below. The tuning of the adaptive detrainment parameter r_det, however, highlights the final aspect of tuning, which





is tuning in the context of the coupled model. Like most coupled modelling centres, the Met Office primarily develop their atmospheric model to perform optimally when forced by observed sea surface temperatures (SSTs) and develop their ocean model to perform optimally with observed atmospheric fluxes. We run the coupled model routinely during the development of the component models, so as to monitor the impact of component model development on the coupled system, and we also

accept that there is often a requirement to tune the coupled model. At the point of this tuning, however, we aim not to degrade the fidelity of the uncoupled component models. The main atmospheric tuning motivated by the performance of the coupled model was the value of r_det=0.8. Reducing this as far as 0.7 improved the TTL temperature biases, but degraded the SSTs of the coupled model in the north Pacific due to its impact on the atmospheric turbulent fluxes. The final compromise value included in Table 6 was found to still give some improvement to the TTL biases, whilst keeping the north Pacific biases within

an acceptable range.

## 4   Model evaluation

### 4.1   General climatological assessment

The assessment of GA7.0/GL7.0 is performed by testing the new configuration in a large number of systems, many of which are listed in Table 8. For the purpose of this paper, we focus on assessing the model in atmosphere/land-only climate simulations at

N216 resolution and forecast-only NWP case studies at N768 resolution ($\approx 17\,\mathrm{km}$ in the mid-latitudes). A top-level summary of the impact of the new configuration on the model's climatology is presented in Fig. 12. This shows normalised assessment criteria (i.e. the ratio of spatial RMSE for a number of time-meaned fields) from a pair of atmosphere/land-only climate simulations at N216 resolution. In general, the quality of the simulation according to these measures is fairly similar to that in GA6.0, with the majority of measures that are outside the range of observational uncertainty changing by less than 15%.

The fields that lie furthest from the observations, i.e. where the difference between the model and the observations is much larger than the observational uncertainty, continue to include the model's tropical precipitation rates. This is in contrast to the extra-tropical precipitation rates, where the spatial disagreement between the model's mean precipitation and observations is smaller than the spatial difference between observational datasets. There is a small improvement in the errors over tropical land (where the error in GA6 is most in disagreement with the observations) but a small increase in the error over the ocean.

Figure 13 shows the annual mean precipitation rate compared to GPCP, which is the primary precipitation climatology used in Fig. 12. This shows a reduction in the intensity of oceanic precipitation along the inter-tropical convergence zone (ITCZ), with an increase in precipitation over tropical land. This redistribution of precipitation from tropical ocean to tropical land is generally an improvement, but the hydrological cycle of the model remains too active, with the global mean precipitation rate of $3.16\,\mathrm{mm/day}$ being outside the range of observational estimates of between $2.61\,\mathrm{mm/day}$ (Adler et al., 2003) and

$3.12\,\mathrm{mm/day}$ (Legates and Willmott, 1990b). Also, there are some regions, such as the maritime continent, where these changes are not an improvement; this will be discussed in more detail below.

One of the larger impacts on the model's climatological large-scale circulation comes from the introduction of the 6A convection scheme in GA ticket #64. As discussed in Sec. 3.7, the more accurate iterative detrainment calculation allows



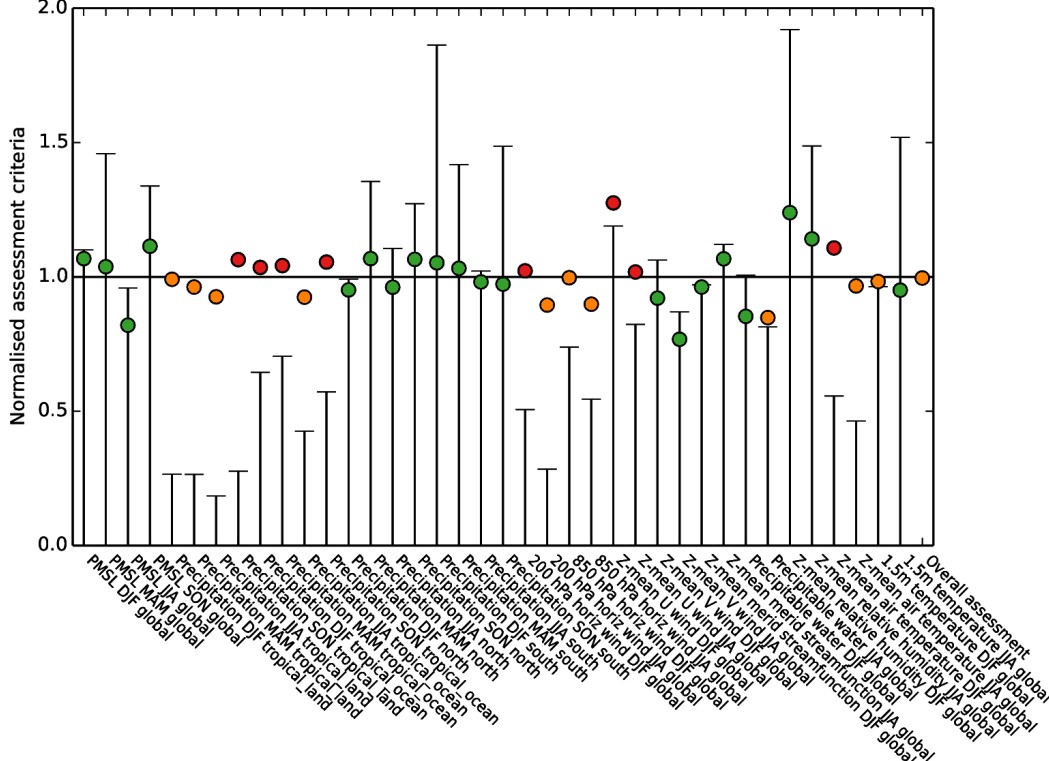

**Figure 12.** Normalised assessment criteria (ratio of mean field RMSEs) for a number of atmospheric fields from a GA7.0/GL7.0 atmosphere/land-only climate simulation at N216 horizontal resolution compared to an equivalent simulation using GA6.0/GL6.0. Statistics shown are for seasons DJF, MAM, JJA and September–November (SON) and for regions global, tropical land (land points between 30° N and 30° S), tropical ocean (sea points between 30° N and 30° S), north (30°– 90° N) and south (30°– 90° S). The observation datasets used are HadSLP2 pressure at mean sea level (Allan and Ansell, 2006), GPCP precipitation (Adler et al., 2003), SSMI precipitable water (Wentz and Spencer, 1998) and CRUTEM3 1.5 m temperature (Brohan et al., 2006), whilst the remaining climatologies are from ERA-interim re-analyses (Berrisford et al., 2009). The whisker bars are observational uncertainty, which is calculated by comparing these with alternative datasets; these are ERA-40 pressure at mean sea level and precipitable water (Uppala et al., 2005), CMAP precipitation (Xie and Arkin, 1997), Legates and Willmott (1990a) 1.5 m temperature and MERRA reanalyses for everything else (Bosilovich, 2008). Green circles denote fields for which the RMSE lies within observational uncertainty, whilst orange or red circles denote fields that do not, and for which the RMSE is improved or degraded respectively.

the deep convection to be deeper, and hence the upper branch of the Hadley circulation (i.e. the divergent flow returning air poleward from the top of convection in the ITCZ) reaches slightly higher up in the atmosphere. Figure 14 shows the zonal mean of the JJA meridional wind from the N216 atmosphere/land-only climate simulations discussed above. In this season, the ITCZ is in its northernmost position and the strongest branch of the Hadley circulation is to the south of this over the equator.
5 The difference between the two simulations shows that there is reduced southerly flow in the lower half of the returning



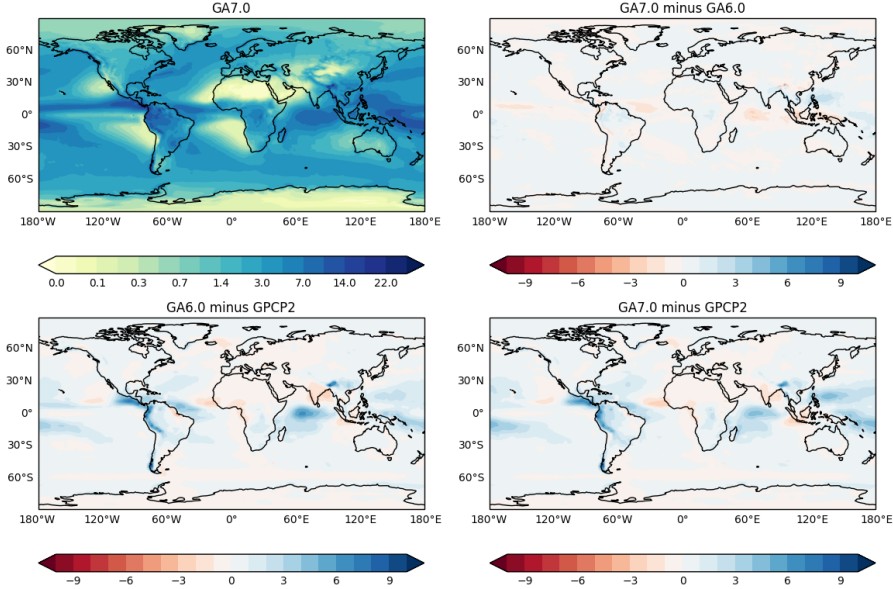

**Figure 13.** Annual mean precipitation rate ($\mathrm{mm/day}$) in the N216 atmosphere/land-only climate simulations presented in Fig. 12 showing GA7.0 (top left), the difference from GA6.0 (top right) and the bias compared to GPCP observations in GA6.0 (bottom left) and GA7.0 (bottom right).

circulation (i.e. below $200\,\mathrm{hPa}$), which improves the bias there vs ERA-interim. Above this, there is a small signal that the deepest convection is slightly deeper, as the weak bias in the retuning flow is reduced in GA7.0. However, there is also a signal that there is some reduction in the weak bias in the lowest regions of the returning flow below $300\,\mathrm{hPa}$, which suggests a beneficial increase in the variability in the height of the outflow from deep convection.

Most of the physics changes described in Sect. 3 have an impact on either the simulation of cloud or its interaction with the model's radiation schemes. Williams and Bodas-Salcedo (2017) present a detailed evaluation of the model's cloud fields by comparing a large number of cloud diagnostics with 3-dimensional satellite data and synoptic observations. They note that GA7.0 significantly reduces the amount of thin, often sub-visual, cirrus cloud which in GA6.0 were extended from thicker clouds through over-active cirrus spreading; this is improved in GA7.0 via GA ticket #98. There is a reduction in excessive

hydrometeor fraction in the mid-latitude boundary layer, which is largely due to the removal of excess drizzle by the new warm rain microphysics (GA ticket #52). Finally, they note that whilst the GA7.0 simulation of stratocumulus cloud is reasonable, there is too little moderately reflective cloud and too much optically thick cloud in these regions, which results in the in-cloud albedo being generally too high. Both the large amount of cirrus cloud and the excess drizzle in GA6 are problems that have previously been noted by operational forecasters using output from the UM, so we expect these problems to be improved when

a configuration based on GA7 is implemented.





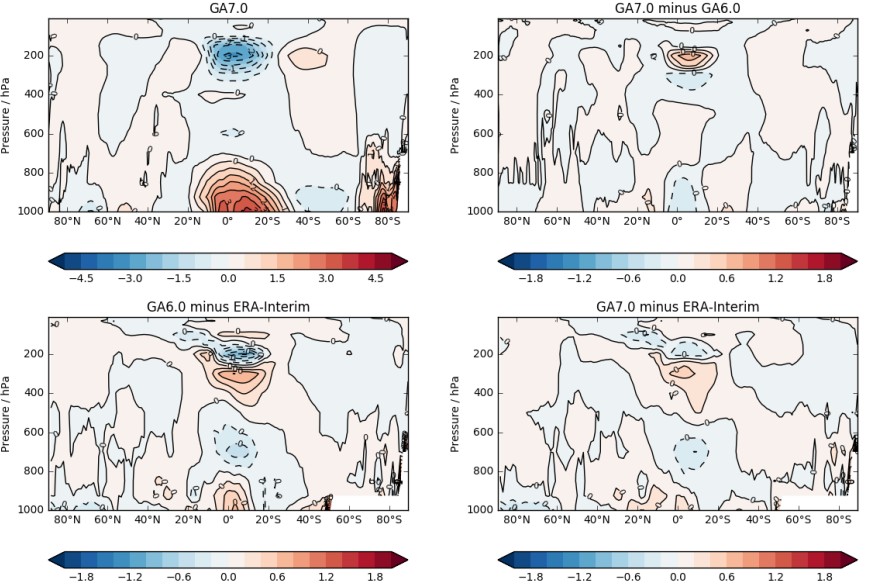

**Figure 14.** Zonal/time mean of the JJA meridional wind $(\mathrm{m\,s^{-1}})$ in the N216 atmosphere/land-only climate simulations presented in Fig. 12 compared to ERA-interim reanalysis. The layout is the same as in Fig. 13.

The net impact of these cloud changes on the top of atmosphere SW radiation fields is shown in Fig. 15, which compares the reflected SW from the N216 climate simulations discussed above. The impact of GA7.0 is generally beneficial with a reduction in both the largest positive biases in the tropics and the wide-spread negative bias over the Southern Ocean. The low bias in reflected SW over the Southern Ocean in both GA6.0 and GA7.0 contributes to a large SST bias in this region

in coupled model simulations. In coupled simulations using GA6.0 (not shown) these biases locally reach values as large as 6 K, which made us designate this as a critical error in GA6.0. This is significantly improved in coupled simulations using GA7.0, which are discussed in more detail in Williams et al. (under revision). Both the bottom panels of Fig. 15 highlight the issue of asymmetry between the SW errors in the northern and southern hemispheres; in general, simulations with GA6.0 show too much reflected SW in the northern hemisphere and too little in the southern hemisphere. This is slightly improved in

GA7.0 with a larger increase in reflected SW over the Southern Ocean than over the North Pacific and North Atlantic, but as discussed in section 3.11, this required some care during the tuning of the final configuration and there are still slight increases in error over these northern hemisphere oceans. An important component of the Southern Ocean bias is believed to be due to microphysical processes, in particular a shortfall in the amount of supercooled liquid water. This is represented slightly better in GA7.0 with the turbulent production of liquid water in mixed phase clouds (GA ticket #120) but the model still converts the

resulting liquid water too quickly into ice, which is a subject of ongoing research. More recently, the strength of the aerosol-cloud interaction has also been highlighted as an important contributor to this asymmetry. This is discussed in more detail in Sect. 4.5. Figure 16 shows the equivalent plot for outgoing LW radiation (OLR). Generally, the impacts are smaller than





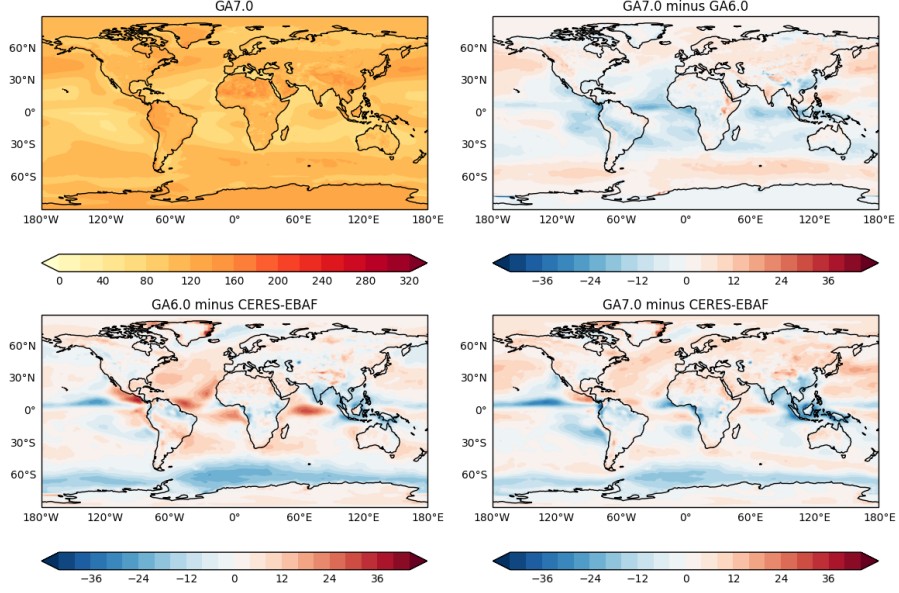

**Figure 15.** Annual mean top of atmosphere reflected SW radiation ($\mathrm{W\,m^{-2}}$) in the 27 year N216 atmosphere/land-only climate simulations presented in Fig. 12 compared to CERES EBAF (Clouds and the Earth's Radiant Energy System–Energy Balanced and Filled dataset, Loeb et al., 2009). The layout is the same as in Fig. 13.

in the SW, although there is a small increase in the global mean OLR (of about $1.5\,\mathrm{W m^{-2}}$); this improves the bias versus CERES EBAF over the sea, but increases the bias over tropical land. The exception to this is the Indian subcontinent, which is significantly improved and will be discussed in more detail below.

The combined impact of changes to the top of atmosphere radiation budget (and other processes) on the model's temperature
structure are shown in Fig. 17. This shows the annual mean GA6.0 and GA7.0 temperature biases versus ERA-interim both
in the N216 simulations discussed above, as well as in an equivalent pair of atmosphere/land-only climate simulations at N96
resolution. There is a consistent signal of GA7.0 being warmer than GA6.0 throughout the troposphere, which brings the
model's climatology into better agreement with a number of reanalysis datasets. This comes from a combination of a number
of changes in GA7.0, but the reduction in the upper-tropospheric cold bias, particularly in the tropics, is largely due to the
improved numerics in the 6A convection scheme. This removes a long-standing bias in the model's temperature structure, that
has been present for a number of development cycles. Near the tropical tropopause, the impact of GA7.0 on the temperature
structure is more complicated; we discuss this in more detail in Sect. 4.3.

## 4.2   Aspects of the climatology improved through stochastic physics

As discussed in Sec. 3.10, the inclusion of the stochastic physics package in GA7.0 means that for the first time, these schemes
will be used in long-range climate projections using a standard UM science configuration. In addition to improving the vari-





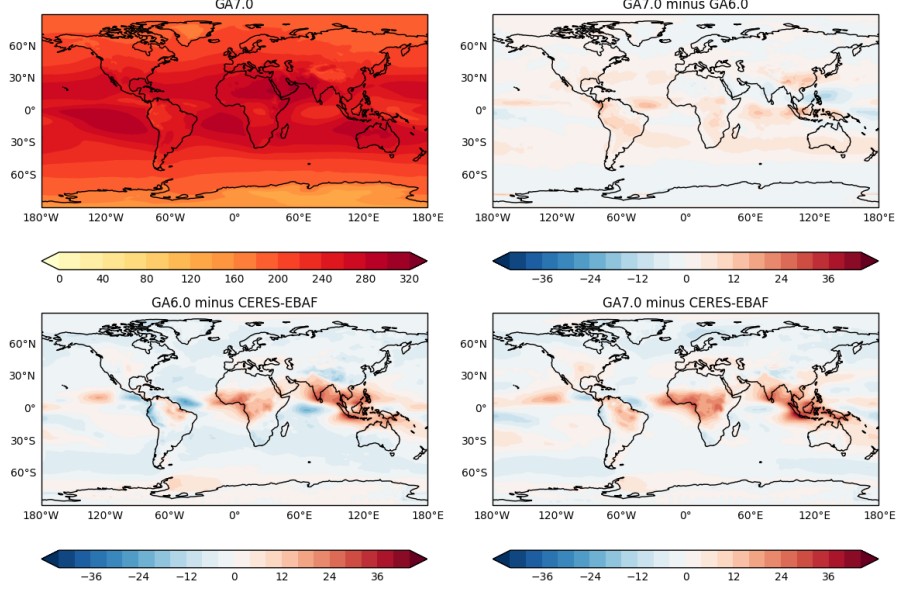

**Figure 16.** Annual mean top of atmosphere outgoing LW radiation ($\mathrm{W\,m^{-2}}$) using the same simulations, observational dataset and layout as Fig. 15.

ability of the model, particularly as measured by the spread in an ensemble of predictions, these schemes also have an impact on the mean climatology (Sanchez et al., 2016). Some of these improvements are highlighted by Fig. 18, which shows the mean JJA precipitation biases versus GPCP in the tropical eastern hemisphere. Over India and the Indian Ocean, we see an improvement in the long-standing bias of there being too much precipitation over the ocean and too little over the subconti-

nent. This improves another critical error in GA6.0, which has been particularly poor for a number of GA releases; whilst these regions stand out as needing further improvement, in N96 atmosphere/land-only simulations, these biases are now the smallest they have been since before the definition of Global Atmosphere 3.0 (Walters et al., 2011). Whilst the largest contribution to this improvement comes from the stochastic physics package, there were also contributions from other changes including the 6A convection scheme (GA ticket #64) and the improved warm rain microphysics (GA ticket #52). One region where we

see an increase in the dry bias over tropical land in GA7.0 is over the Maritime Continent. These degradations with GA7.0 appear to be mostly due to a general shift in the precipitation patterns, both the general (and elsewhere) beneficial decrease in precipitation in this region due to the inclusion of stochastic physics, and a general (and elsewhere) beneficial decrease in precipitation over tropical land from the interaction of the convective cores with the radiation scheme (through GA ticket #44). We do not believe that these changes are due to degrading the underlying processes that cause this bias, which has been shown

elsewhere to be related to more locally pertinent issues such as the representation of sea breezes and the interaction of the convection scheme with the model's smoothed mean orography (Birch et al., 2015; Rashid and Hirst, 2016).



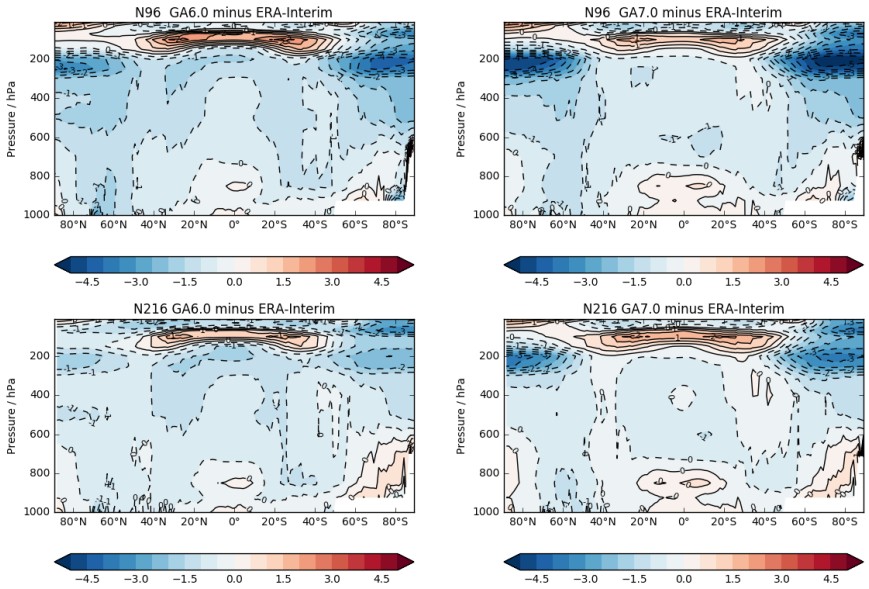

**Figure 17.** Annual mean zonal mean temperature biases (K) versus ERA-interim in the 27 year N216 atmosphere/land-only climate simulations presented in Fig. 12, alongside results from equivalent simulations performed at N96 resolution. The rows show data from N96 (top) and N216 (bottom), whilst the columns are data from GA6.0 (left) and GA7.0 (right).

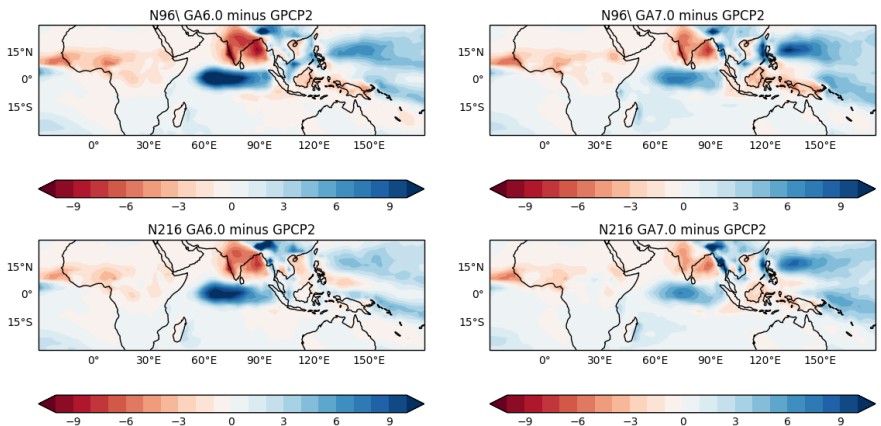

**Figure 18.** JJA precipitation rate biases versus GPCP over the Indian Ocean and the maritime continent (mm/day) from 27 year N96 and N216 atmosphere/land-only climate simulations of GA6.0 and GA7.0. The layout is the same as in Fig. 17.

Over Africa, the inclusion of stochastic physics leads not only to an improvement in the precipitation biases, as seen in Fig. 18, but also to a beneficial increase in the number of African Easterly Waves (AEWs). Figure 19 shows the number of AEWs identified per season (July–September) from the 27 year atmosphere/land-only climate simulations discussed above.





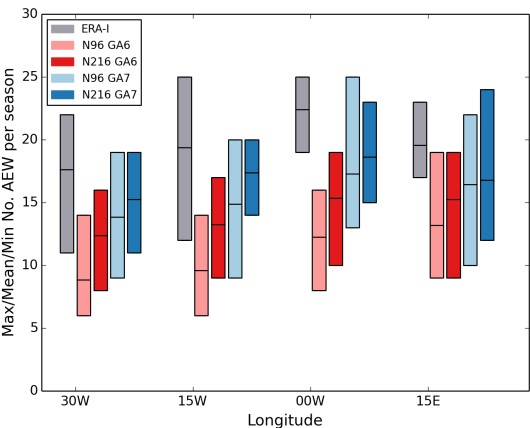

**Figure 19.** Average, maximum and minimum number of African easterly waves (AEWs) tracked per year (between July and September) in the 27 year N216 atmosphere/land-only climate simulations presented in Fig. 12, alongside results from equivalent simulations performed at N96 resolution. These waves are identified and tracked using the algorithm described in Bain et al. (2013).

This shows that in GA7.0, the number of AEWs is in better agreement with ERA-interim and also shows a slightly reduced resolution sensitivity; as a result, the mean number of AEWs identified in an N96 simulation with GA7.0 is greater than the number from an N216 simulation with GA6.0. Despite this improvement, there are still persistent biases in the mean state over northern Africa, such as too broad an African Easterly Jet and too southern a position for the continental ITCZ. This has an effect on the characteristics of AEWs and the convection parametrisation is still not sensitive enough to the dynamics of the wave and associated moisture convergence, which results in deficiencies in the representation of the convection-circulation interaction and related wave growth (Tomassini et al., accepted).

### 4.3 Tropical tropopause layer biases and energy conservation

In the troposphere, Fig. 17 shows a consistent impact on the climatological temperature profile at both N96 and N216, with both resolutions being warmer in GA7.0, reducing the cold bias versus ERA-interim. In the tropical tropopause layer (TTL), however, we see a contradictory signal, with a decreased warm bias at N96, but an increased warm bias at N216.

To understand this, let us first concentrate on the impact at N96. Hardiman et al. (2015) describe the TTL temperature and stratospheric water vapour biases in N96 GA6.0 simulations and show that the presence of a warm bias in the TTL was a common feature amongst many of the models that were submitted to CMIP5. This warm bias is closely associated with a moist bias in the stratosphere, due to not enough moisture being freeze-dried out of air ascending through the coldest point in the vertical temperature profile. This can prove a particular problem in models with interactive chemistry, as a stratospheric moisture bias can induce a bias in the ozone field, which in turn can feed-back to further increase the warm bias. This was designated as a critical error in GA6.0, particularly at N96 resolution, which is the resolution at which the CMIP6 simulations





of UKESM1 (with interactive chemistry) will be performed. Hardiman et al. (2015) developed a metric to measure the impact of model changes on TTL temperature and stratospheric moisture biases and demonstrated the impact of many of the changes proposed for inclusion in GA7.0. Reproduced for the simulations described above in Fig. 20, this compares the annual mean tropical temperature bias at $100\,\mathrm{hPa}$ (as a proxy for the bias in the "cold point" temperature) and the tropical specific humidity bias at $70\,\mathrm{hPa}$ (as a measure of the bias in moisture passing through the cold point into the lower stratosphere). A careful

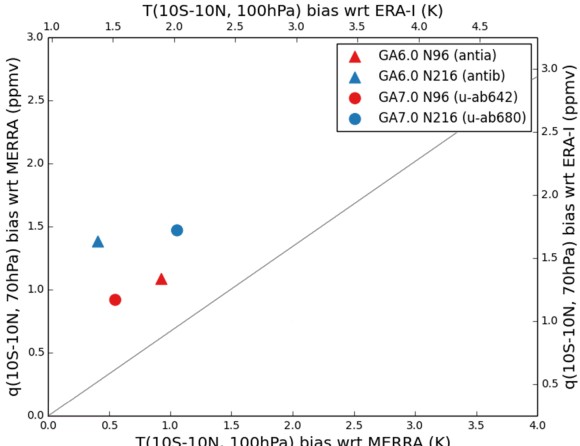

**Figure 20.** Tropical tropopause layer temperature biases and lower stratospheric humidity biases from $27\,$year N96 and N216 atmosphere/land-only climate simulations of GA6.0 and GA7.0 versus MERRA and ERA-interim reanalyses (following Hardiman et al., 2015).

monitoring of the impact of these changes, alongside the tuning described in Sect. 3.11, allowed us to ensure that the bias is improved in N96 simulations of GA7.0 to the extent that we do not expect a strong ozone feedback in UKESM1.

At resolutions of N216 and above, the impact of GA7.0 on the TTL bias is complicated by the inclusion of the conservation of mass-weighted potential temperature (GA ticket #146). As discussed in Sect. 3.1, this change removes a spurious positive

10 source of energy, which was known to increase in magnitude with increasing model resolution. In GA6.0 climate simulations, this energy source was compensated by the global energy correction described in Sect. 2.13. The magnitude of this energy correction term is monitored in all simulations, and is usually expected to be $O\pm(0.5\text{-}1.0)\,\mathrm{Wm^{-2}}$. Table 7 shows that whilst this is true at N96, in GA6.0 simulations at higher resolutions, the response of the energy correction scheme to the error in the temperature advection leads to the scheme being much more active, which was also designated as a critical error in GA6.0. The

15 improved temperature advection reduces this sensitivity and hence addresses this problem. When the energy correction scheme stays within its expected bounds, its impact (not shown) is small enough not to significantly affect the model's climatology. In N216 simulations of GA6.0, however, the atmospheric cooling associated with the $\approx -2.5\,\mathrm{Wm^{-2}}$ correction was big enough to cool the TTL, and spuriously reduce the warm bias. This means that the net impact of all the GA7.0 changes at N216 as seen in Fig. 17 and Fig. 20, is to increase the TTL warm bias at this resolution.



**Table 7.** Mean value of the global energy correction applied over the 27 year simulations at N96 and N216 presented above. *We also include the results from some shorter N512 resolution simulations using GA6.0 and GA7.0, which ran for 13 years and 10 years respectively. These should be comparable, as this measure is stable to $\mathrm{O}(0.1\,\mathrm{W\,m^{-2}})$ in simulations of only a few years in length.

| | Global energy correction ($\mathrm{Wm^{-2}}$) | | |
| --- | --- | --- | --- |
| Config. | N96 | N216 | N512* |
| GA6.0 | -0.79 | -2.72 | -3.68 |
| GA7.0 | -0.61 | -0.49 | -0.27 |

### 4.4 Predictive skill in short-range forecasts

The primary assessment of the predictive skill of NWP forecasts using GA7.0 has been performed using forecast-only "case studies" initialised from independent analyses. Figure 21 is a top-level summary of the difference in RMSE between GA7.0 and GA6.0 for a number of fields and forecast ranges in a set of N768 resolution case studies initialised from operational ECMWF analyses on 24 synoptically-independent dates. The RMSE for each field and forecast range is calculated against synoptic surface and radiosonde observations at the observation location and against verifying analyses on a common $2.5°$ global grid. When verifying against analyses, the ECMWF analysis fields are first converted into native UM prognostics, which are then used to calculate the various diagnostic fields; this ensures consistency both with the way in which the forecasts are initialised and with the way that particular diagnostics (e.g. PMSL) are calculated, which is particularly important in regions of high orography. The figure shows that the number of fields improved outweighs the number of fields degraded. In particular, there are improvements in northern hemisphere PMSL and $500\,\mathrm{hPa}$ geopotential height, which is primarily due to improved insulation of the surface under lying snow in DJF. There are also improvements to temperatures at $250\,\mathrm{hPa}$, but mixed signals at lower levels. As with the model's long term climatology, the impact of GA7.0 on the model's short-range temperature error structure is complicated. Figure 22 shows the tropical temperature error profile calculated from both radiosonde observations and verifying analyses. At $100\,\mathrm{hPa}$ we see the same improvement in the TTL warm bias seen in N96 climate simulations in Fig. 17; although these NWP forecasts are at much higher horizontal resolution than N96, they do not include the energy correction term that cooled higher-resolution GA6.0 climate simulations and hence we do not see a compensating warming in moving from GA6.0 to GA7.0. Below this, again as in Fig. 17, GA7.0 is warmer than GA6.0 throughout most of the troposphere. Between 100 and $200\,\mathrm{hPa}$ this warming appears too great, but throughout most of the rest of the troposphere this is an improvement to a long-standing tropospheric cold bias. The exception to this is at $700\,\mathrm{hPa}$, which remains too cool in GA7.0. Ujiie et al. (2017b) show that this cold "spike" is due to an overly simple representation of the melting of convective precipitation at the freezing level and propose improvements to this that will be implemented in a future GA release. The impact of GA7.0 on the general circulation is also similar between the model's long range climatology and these short range NWP forecasts. Figure 23 shows the tropical horizontal wind error profile from radiosonde observations and verifying analyses. There is a decrease in the





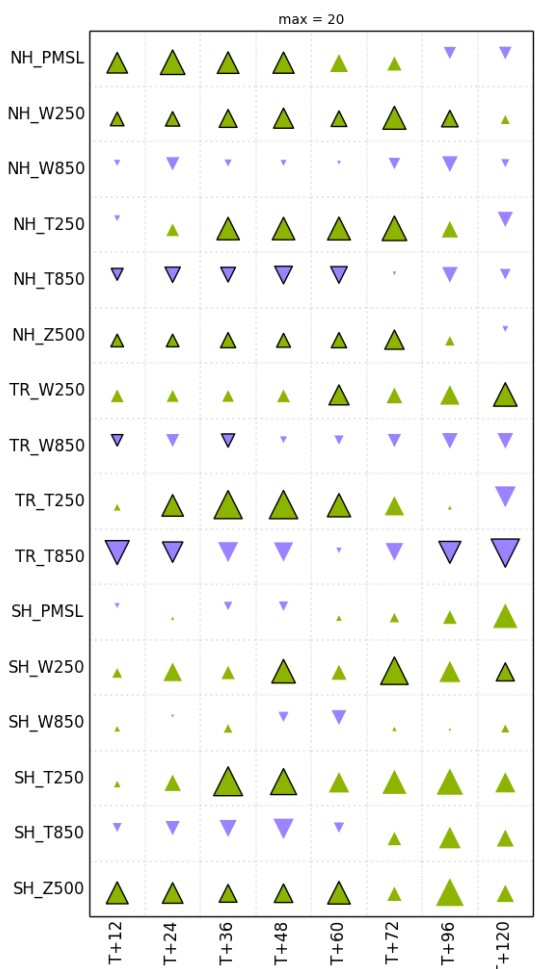

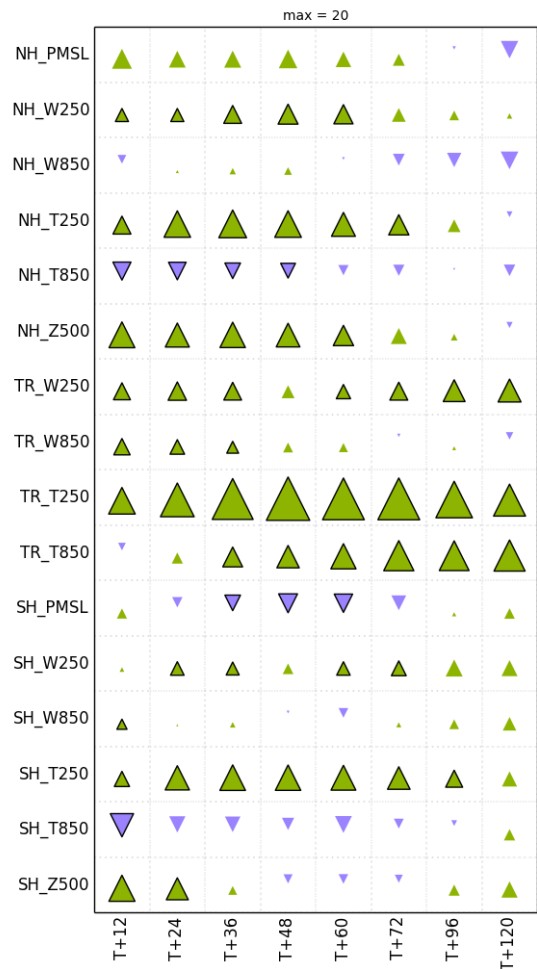

**Figure 21.** Percentage change in RMSE between GA7.0 and GA6.0 when verified against synoptic and radiosonde observations (left) and analyses (right) in a set of 24 N768 resolution forecast case studies run from operational ECMWF analyses. The rows represent RMSE differences for particular parameters in the northern hemisphere (NH), the tropics (TR) and the southern hemisphere (SH). The boundaries between these regions are at latitudes of $\pm20°$ for verification vs observations and $\pm18.75°$ for verification vs analyses, which are calculated on a standard $2.5°$ grid. The parameters are pressure at mean sea level (PMSL), vector wind errors and temperature errors at 250 hPa and 850 hPa (W250, W850, T250, T850) and geopotential height at 500 hPa (Z500). The columns represent forecast ranges from 12 h (T+12) to 5 days (T+120). Green up arrows represent a decrease in RMSE (i.e. an improvement) whilst purple down arrows represent a degradation. The area of the arrow denotes the size of the change with a 20% change filling the width of the column and a solid outline denotes that the change is statistically significant.





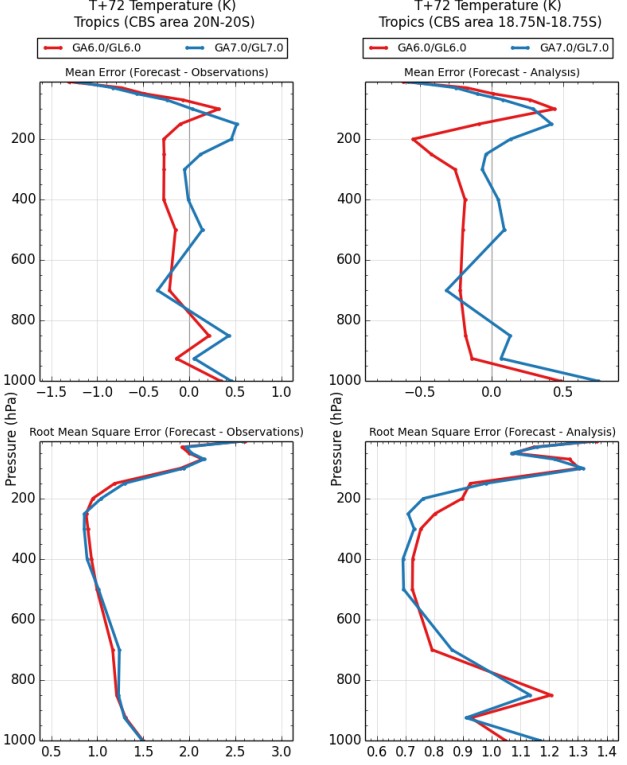

**Figure 22.** Profiles of 3 day (T+72) tropical temperature bias (top) and RMSE (bottom) versus radiosonde observations (left) and ECMWF analyses (right) from the N768 GA6.0 and GA7.0 case studies presented in Fig. 21.

horizontal wind speed at 200-250 hPa and an increase in the horizontal wind speed at 100 hPa, which is consistent with higher divergent outflows due to deeper convection in the 6A convection scheme. This leads to an apparently large improvement in the 250 hPa vector wind error against analyses in Fig. 21 and a degradation in the 100 hPa vector wind error in Fig. 23. It is likely, however, that these apparent improvements and degradations are due, at least in part, to the characteristics of the UM
5  becoming more or less like those of the forecast model used in generating the ECMWF operational analyses.

To make a more quantitative assessment of the performance of GA7.0 in short-range NWP forecasts, we would usually perform "assimilation trial" forecasts using a 4D-Var or hybrid 4D-Var data assimilation cycle based on the Met Office operational global analysis system (Clayton et al., 2013). Indeed, such trials were used during the early-to-middle stages of developing GA7.0/GL7.0. Later on, however, we encountered problems in making a fair comparison between GA7.0 and GA6.0, which
10 meant that they were not used to constrain the final stages of development. The first problem was that prior to March 2016, the Met Office analysis at 850 hPa had a tendency to be warm compared to radiosonde observations, due to the details of the bias correction scheme used in assimilating satellite radiances. This has since been addressed by introducing a variational bias correction scheme (VarBC), which brings the analysed global mean 850 hPa temperature both closer to that of other NWP



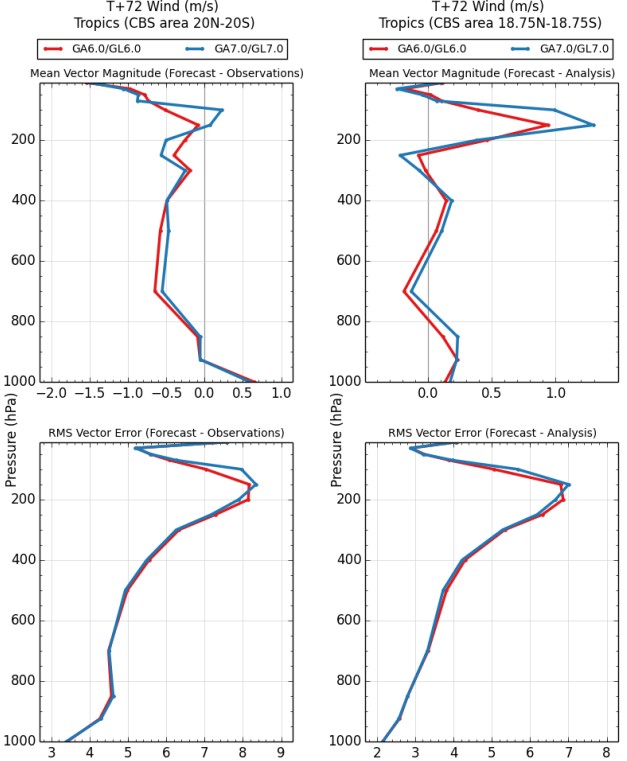

**Figure 23.** Profiles of 3 day (T+72) tropical wind speed bias (top) and vector wind RMSE (bottom) versus radiosonde observations (left) and ECMWF analyses (right) from the N768 GA6.0 and GA7.0 case studies presented in Fig. 21.

centres and closer to that of radiosonde observations (Ujiie et al., 2017a). The combination of a warm analysis and a climatological cold bias in the operational GA6.1/GL6.1 meant that the model's mean 850 hPa temperature would drift towards the radiosonde observations with increasing forecast range. As a result, test forecasts of prototype GA7 configurations, with their reduced cooling at lower levels, would verify warm against radiosonde observations when initialised from either opera-

5 tional Met Office analyses or pre-VarBC assimilation trials. The pre-operational trials of VarBC using GA6.1 showed that the cooler analyses reduced the rate of cooling towards the model's climatology, and hence significantly improved its verification against own analysis. Trials of GA7.0 using VarBC brought the model's climatology closer to its analysed value, and hence reduced this drift even more. This is illustrated in Fig. 5 and Fig. 16 of Ujiie et al. (2017a). The second problem encountered in producing trial forecasts based on GA7.0 was in the specification of the stationary background error covariance matrix $B_c$.

10 Data assimilation relies on having a suitable representation of these forecast errors, incorporating model, observation, initial condition and representivity errors. $B_c$ is calibrated from a training dataset representing typical forecast errors and in hybrid 4D-Var, it makes an important contribution to the background error penalty and ideally should be consistent with the model used in the assimilation cycle. In practice, however, it is usually sufficient to use a $B_c$ based on the control model when testing





a model upgrade and later produce a $\boldsymbol{B}_c$ based on the new model for consistency. In our original tests of GA7.0, this appeared not to be the case and a series of tests (not shown) suggested that the interaction between the operational $\boldsymbol{B}_c$ and GA7.0 was detrimental to forecast performance, particularly at lead times of 2 days or less. The operational $\boldsymbol{B}_c$ was based on a training dataset produced by the ECMWF forecast model but evolved using the UM (Wlasak and Cullen, 2014), so blending the forecast

error characteristics of the two models. The impact of the covariance/model inconsistency appears to be smaller when $\boldsymbol{B}_c$ is calculated using training data produced by an experimental ensemble of four-dimensional ensemble variational data assimilations (Bowler et al., 2017; Wlasak et al., in prep.), even when based on the previous GA6.1 configuration. A new $\boldsymbol{B}_c$ generated in this way will be made operational in late 2017, which we believe will allow us to revert to the strategy of staggered model and $\boldsymbol{B}_c$ updates described above. In time, the replacement of the stationary covariance matrix with one based solely on the

history of recent forecasts may make it easier to test the impact of model upgrades on forecast quality in an operational-like system.

### 4.5 Problems identified with GA7.0/GL7.0

#### 4.5.1 Overly low snow albedo in extremely cold conditions

The albedo of snow over land in GL7.0 is linked to the snow grain size as specified by the parametrisation of Marshall and

Oglesby (1994). This leads to a rapid growth from a $50\,\mu\mathrm{m}$ grain size for fresh snow to $150\,\mu\mathrm{m}$ over a few days even in very cold conditions below $-30^\circ$ C, followed by slow grain growth for aged non-melting snow. GL7.0 thus has a lower albedo for cold snow than the temperature-dependent albedo in GL6.0. Whilst this appears to reduce the reflected SW bias versus CERES-EBAF over Antarctica, as seen in Fig. 15, other evidence such as the observed radiative fluxes used in the GABLS4 intercomparison (Bazile et al., 2014) suggests that the model's Antarctic albedo of $\sim 0.78$ is too low. This is consistent with

snow grains growing too large, which is supported by measurements of snow specific surface area that imply grain sizes of $\sim 80\,\mu\mathrm{m}$ (Gallet et al., 2011), compared to the values of $\sim 150\,\mu\mathrm{m}$ seen in GL7.0. In NWP forecasts, this leads to a warm bias at low levels over Antarctica during austral summer, as shown in Fig. 24. In future GL releases, we plan to improve this by using the equi-temperature snow grain growth scheme of Taillandier et al. (2007).

#### 4.5.2 Overly strong negative aerosol effective radiative forcing

As discussed in Sect. 3.11, the climate simulations that contributed to the development and tuning of GA7.0 were all simulations of the present-day climate. Atmosphere/land-only simulations were performed with time-varying emissions, trace gas concentrations and observed SSTs for years after 1981 and coupled simulations were performed over much longer periods, but using perpetual year 2000 emissions and trace gas concentrations. Following the definition of GA7.0, however, its anthropogenic Effective Radiative Forcing (ERF) was assessed over the historical period (1850–2000), in order to determine its

suitability for climate change simulations. This revealed an extremely strong, negative aerosol ERF of $\approx -2.7\,\mathrm{Wm}^{-2}$, which





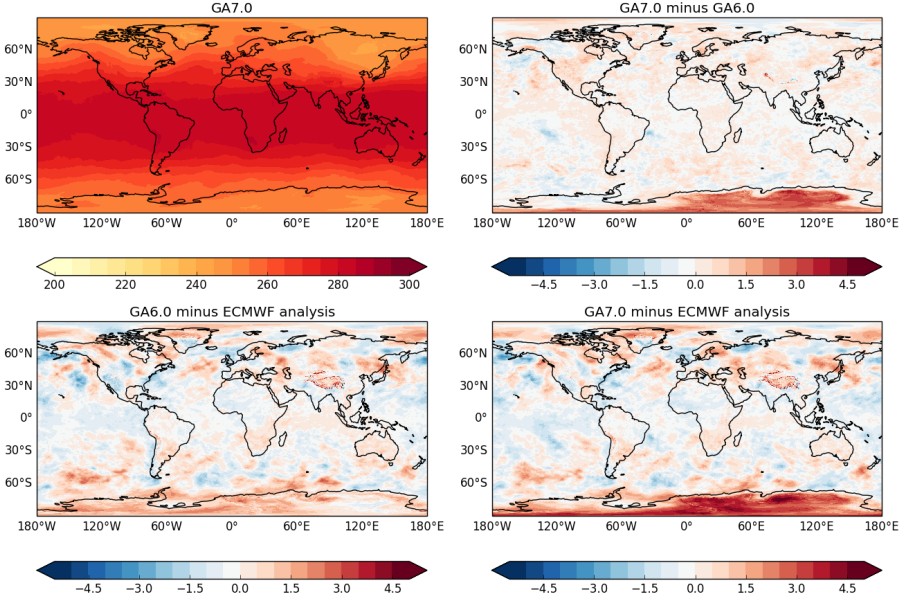

**Figure 24.** T+120 temperature (K) at $700\,\mathrm{hPa}$ meaned over the 12 pairs of forecasts included in Fig. 21 that were initialised in DJF. The panels show the mean GA7.0 forecast (top left), the difference between GA7.0 and GA6.0 forecasts (top right) and the bias compared to the verifying ECMWF analyses in GA6.0 (bottom left) and GA7.0 (bottom right).

contributed to a total anthropogenic ERF (from greenhouse gasses, aerosols, ozone and land use change) that was negative[9], with a value of $\approx -0.4\,\mathrm{Wm^{-2}}$. This would have had an extremely detrimental impact on historical simulations using GA7.0. For CMIP6, this is addressed by developing the GA7.1 "branch" configuration (discussed below); beyond this, we aim to address this shortcoming by building the majority of the GA7.1 developments back into the Global Atmosphere "trunk" in a future release.

## 5   Additional developments and tunings included in the CMIP6 climate configuration Global Atmosphere 7.1

As discussed above, the very strong negative aerosol ERF in GA7.0 make it unsuitable for use in historical climate change simulations, and hence in submissions to CMIP6. For this reason, we developed the GA7.1 "branch" configuration that reduced anthropogenic aerosol ERF, whilst maintaining a present-day simulation similar to that from GA7.0 (Mulcahy et al., in prep.b). GA7.1/GL7.0 is the atmosphere/land component of the Global Coupled model 3.1 (GC3.1) configuration (Williams et al., under revision), which will underpin the HadGEM3-GC3.1 physical model submission to CMIP6. GC3.1 will also be the physical model component of the UKESM1 Earth system model, which will also be submitted to CMIP6.

---

[9]Chapter 8 of Working Group I's contribution to the Intergovernmental Panel on Climate Change fifth assessment report (IPCC, 2013) states that "It is certain that the total anthropogenic ERF is positive."

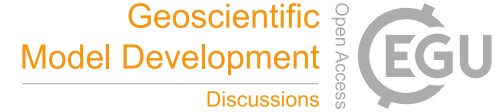



The differences between GA7.1 and GA7.0 are:

1. A new parametrisation of spectral dispersion for the cloud droplet size distribution following Liu et al. (2008). This parametrises the impact of the droplet number (and hence the aerosol loading) on the dispersion relation, as opposed to the simpler approach adopted in GA7.0, in which the spectral dispersion in continental and maritime clouds is specified according to the model's land sea mask (Martin et al., 1994).

2. An update to the complex refractive index of black carbon from $1.75 - 0.44i$ to the more recent estimate of $1.85 - 0.71i$ (Bond and Bergstrom, 2006; Bond et al., 2013).

3. Inclusion of more detailed look-up tables for aerosol optical properties in UKCA-Radaer, enabling more accurate spectral resolution of aerosol solar absorption.

4. Replacement of the climatological oceanic dimethlysulphide (DMS) concentration of Kettle et al. (1999) with the up-dated climatology of Lana et al. (2011).

5. Multiplicative scaling of the parametrised marine emission of DMS in GLOMAP-mode by $(1 + 0.7)$, where the factor 0.7 is designed to account for a missing source of primary marine organic aerosol in GLOMAP-mode.

6. Several improvements to the calculation of the TKE data passed to UKCA-Activate: *(i)* an explicit bug in the level indexing was corrected; *(ii)* the minimum value of TKE used was reduced by an order of magnitude following Boutle et al. (2017) and *(iii)* an explicit estimate of TKE in cumulus clouds was introduced.

7. Retuning of the parameters mp_dz_scal and cca_sh_knob in Table 6 back to their originally proposed values of 2.0 and 0.5 respectively.

Of the changes listed above, 1, 2, 3, 4 and 6 are model developments that we aim to include in a future release of the GA "trunk", whilst 5 and 7 are tunings that may or may not be revisited in the future. The development of GA7.1, including the scientific justification for these differences from GA7.0, is documented in more detail in Mulcahy et al. (in prep.b).

Figure 25 compares the annual mean reflected SW from an N216 resolution present-day climate simulation using GA7.1 with an equivalent simulation using GA7.0. This shows that whilst the changes above reduce the aerosol ERF, the impact on the model's present-day climatology are small, particularly when compared with the impact of GA7.0 in Fig. 15. Where there are differences, these are generally beneficial and in particular, the reduced contrast between cloud brightness in polluted and pristine airmasses reduces the asymmetry in the reflected SW biases discussed in Sec. 4.1.

## 6 Conclusions

The Global Atmosphere 7.0 and Global Land 7.0 configurations consolidate a number of important incremental improvements to the UM and JULES into the GA and GL trunks, including an improved treatment of gaseous absorption in the radiation scheme, improvements to the treatment of warm rain and ice clouds and an improvement to the numerics in the model's





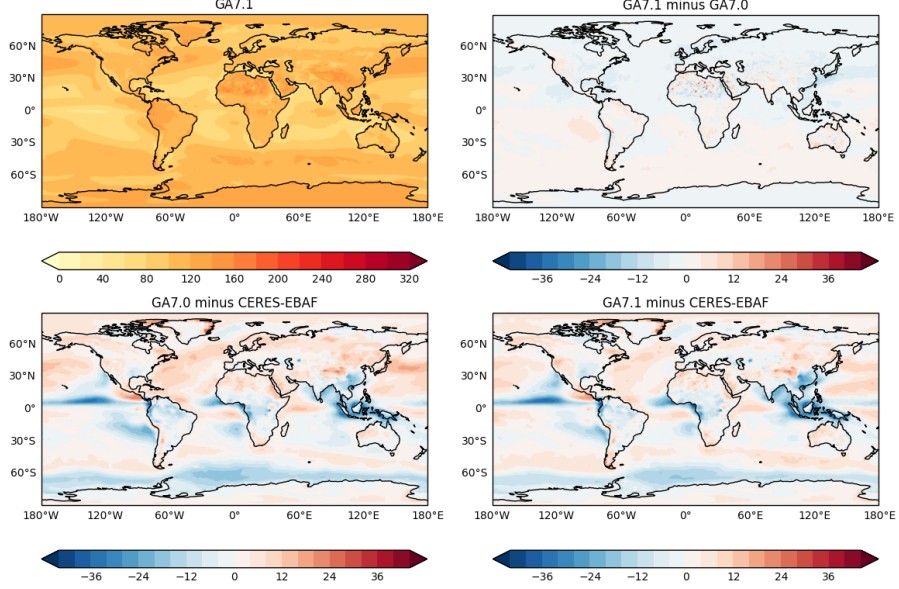

**Figure 25.** Annual mean top of atmosphere reflected SW radiation ($\mathrm{W\,m^{-2}}$) in a pair of 20 year N216 atmosphere/land-only climate simulations using GA7.0 and GA7.1. The panels show GA7.1 (top left), the difference from GA7.0 (top right) and the bias compared to CERES-EBAF observations in GA7.0 (bottom left) and GA7.1 (bottom right).

convection scheme. It also includes additional changes motivated by their impact on the model's largest errors as well as significant structural improvements that increase the complexity and improve the fidelity of climate simulations, namely the UKCA GLOMAP-mode aerosol scheme and the JULES multi-layer snow scheme. An assessment of GA7.0/GL7.0 (and separately of the GC3.0 coupled model configuration that uses it: Williams et al., under revision) shows that the new configurations significantly reduces the four "critical" errors identified in their precursors. Specifically, we see:

- A reduction in the climatological dry precipitation bias over the Indian subcontinent;

- A reduction in the tropical tropopause layer warm bias in climate simulations at the N96 horizontal resolution that will be used in UKESM1 and in NWP simulations at all resolutions;

- The removal of a source of energy non-conservation, reducing the size of the global energy correction term required in higher-resolution climate simulations and

- A significant reduction in the warm SST bias in the Southern Ocean in coupled model simulations.

An initial assessment of the configuration on NWP timescales highlights a number of benefits, although there will need to be an upgrade to the data assimilation system's climatological background error covariances before a configuration based on GA7.0/GL7.0 can be implemented operationally. Finally, an estimate of the model's effective radiative forcing highlighted





that the contribution due to anthropogenic aerosols was too strongly negative, which has been addressed by a small number of changes in the GA7.1 "branch" configuration, most of which are suitable for future inclusion in the GA "trunk". This will allow GA7.1 and GL7.0 to be used as the physical atmosphere and land components in the HadGEM3-GC3.1 and UKESM1 climate models that will be submitted to CMIP6.

**Table 8.** Identifiers for a set of GA7.0/GL7.0 reference simulations across a number of UM code versions and systems/applications. These suites are held on the Met Office Science Repository Service, which also holds the UM and JULES code. Identifiers marked in bold denote those used in the original assessment of the GA7.0 configuration.

| UM code base | Atmosphere/land-only climate | | | Coupled climate | | | Seasonal forecast | NWP case study suite | | | | |
|---|---|---|---|---|---|---|---|---|---|---|---|---|
| | N96 | N216 | N512 | N96 | N216 | N512 | N216 | N216 | N320 | N512 | N768 | N1280 |
| vn10.3 | **u-ab642** u-ab747 | **u-ab680** u-ab770 | u-ab261 | **u-ab673** | **u-ab674** | | | **u-ac441** | **u-ac231** | **u-ac443** | **u-ac445** | **u-ad613** |
| vn10.4 | u-ac283 | u-ad442 | | u-ac349 | u-ac695 | u-ac699 | **mi-an938** | | u-ac493 | | | |
| vn10.5 | u-ae523 | u-ae955 | | u-af082 | u-af206 | | | | u-ae530 | | | |
| vn10.6 | u-ah389 | u-ai936 | | u-ah815 | | | | | u-ah414 | | | |
| vn10.7 | u-ak497 | | | | | | | | u-ak926 | | | |

**Table 9.** Identifiers for a set of GA7.1/GL7.1 reference climate simulations across a number of UM code versions. These suites are held on the Met Office Science Repository Service, which also holds the UM and JULES code. Here, ORCA0.25 denotes a $0.25° \times 0.25°$ NEMO ocean model resolution.

| UM code base | Atmosphere/land-only climate | | Coupled climate (ORCA0.25) | (ORCA0.25) |
|---|---|---|---|---|
| | N96 | N216 | N96 | N216 |
| vn10.6 | u-ai955 | | u-ah981 | u-ah984 |
| vn10.7 | u-al613 | u-al616 | u-ak896 | u-al016 |

5   *Code availability.* Due to intellectual property right restrictions, we cannot provide either the source code or documentation papers for the UM or JULES.

*Obtaining the UM.* The Met Office Unified Model is available for use under licence. A number of research organisations and national meteorological services use the UM in collaboration with the Met Office to undertake basic atmospheric process research, produce forecasts, develop the UM code and build and evaluate Earth system models. For further information on how to apply for a licence see http://www.

10   metoffice.gov.uk/research/modelling-systems/unified-model.

*Obtaining JULES.* JULES is available under licence free of charge. For further information on how to gain permission to use JULES for research purposes see http://jules-lsm.github.io/access_req/JULES_access.html.



*Details of the simulations performed.* UM/JULES simulations are compiled and run in suites developed using the Rose suite engine (http://metomi.github.io/rose/doc/rose.html) and scheduled using the cylc workflow engine (https://cylc.github.io/cylc/). Both Rose and cylc are available under v3 of the GNU General Public License (GPL). In this framework, the suite contains the information required to extract and build the code as well as configure and run the simulations. Each suite is labelled with a unique identifier and is held in the same revision controlled repository service in which we hold and develop the model's code. This means that these suites are available to any licensed user of both the UM and JULES. We also document a more complete set of reference GA7/GL7-based simulations in Tables 8 and 9.

*Competing interests.* The authors declare that they have no conflict of interest.

*Acknowledgements.* MD, NG, SH, AJ, BJ, CJ, JM, AS and RS were supported by the Joint BEIS/Defra Met Office Hadley Centre Climate Programme (GA01101). MU was a visiting scientist working at the Met Office on secondment from the Japan Meteorological Agency under agreement number P014013. The development and assessment of the Global Atmosphere/Land configurations is possible only through the hard work of a large number of people, both within and outside the Met Office, that exceeds the list of authors.

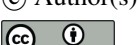



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
