# Peer review of "The Met Office Unified Model Global Atmosphere 7.0/7.1 and JULES Global Land 7.0 configurations"

_Geoscientific Model Development, 2017_

## Short Comment (SC1) · 29 Nov 2017

Apologies for nit-picking but I beleive that in Table 2 and elsewhere in the manuscript the reference given for the IGBP Land Usage is not correct:

Land usage IGBP; Global Soil Data Task (2000) Mapped to 9 tile types

The given reference (GSDT) is for IGBP soils not vegetation. I use Loveland et al as the reference for the IGBP land cover dataset.

Global Soil Data Task: Global soil data products CD-ROM (IGBP-DIS), CD-ROM, https://daac.ornl.gov/SOILS/guides/igbp.html, International Geosphere-Biosphere Programme, Data and Information System, Potsdam, Germany. Available from Oak Ridge National Laboratory 5 Distributed Active Archive Center, Oak Ridge, TN, last access: 25th October 2017, 2000.

Loveland, T.R., Reed, B.C., Brown, J.F., Ohlen, D.O., Zhu, Z., Yang, L. and Merchant, J.W. 2000: Development of a global land cover characteristics database and IGBP DISCover from 1 km AVHRR data. International Journal of Remote Sensing, 21(6-7) 1303–1330.

---

## Author Comment (AC1) · 29 Nov 2017

Thanks for pointing this out. I've had a quick look through the reference provided (Loveland et al., 2000) and agree that *(i)* this is a useful reference and *(ii)* this is a more suitable reference than the one included in the discussions paper.

I will update this ahead of resubmission, following the review of the discussions paper.

---

## Referee Comment (RC1) · P. M. Caldwell (Referee) · 3 Jan 2018

Wow, this is the best-written manuscript I've ever reviewed. I made some minor comments below which the authors should feel free to accept or reject.

Summary: this paper starts with a brief overview of the UKMO atmosphere model versions 7.0 and 7.1 and the JULES land model version 7.0. It then goes into detail about each change from version 6 to version 7. This is followed by a discussion of the model's tuning strategy. At this point, the model is evaluated briefly in climate and weather modes. Remaining problems are then discussed, particularly related to overly-strong aerosol indirect effect. Finally, modifications to improve the aerosol effect are

[Figure]

described. Improvements to the model have beneficial impact and make sense. I was struck by how methodical and focused the UKMO team is at getting the right answer for the right reasons.

Minor comments:

1. The thing that bugged me most in this paper was that the model timestep and target horizontal resolutions were never described or were described late in the paper in a tangential way. It would be nice to include target resolution info (as a table?) in the intro and timestep info in section 2.2.

2. p2 line 1-3: citations in this sentence seem like they're for v7 of the model, which is odd since this manuscript is the overview for this model version. Delete citations here?

3. section 2.1: I'm curious why UKMO continues to use a lat/lon grid instead of a cubed-sphere grid. Also, are polar filters used?

4. I'm curious about model performance. How many cores do you typically use and what is your typical throughput on the machine you use most? This might fit in section 2.2... or might be out-of-scope.

5. section 2.2: Using parallel splitting creates opportunities for water and aerosol species to go negative. How do you handle this and the resulting conservation errors?

6. page 4 line 13: When using the prognostic aerosol scheme, this is included..." is awkward. Replace with "The prognostic aerosol scheme is included..."?

7. page 7 line 12: "We also make use of..." sentence is awkward.

8. It seems odd to talk about Large-scale precipitation (sect 2.4) before large-scale cloud (sect 2.5) since the latter creates the condensate which the former acts on.

9. page 8, top: This is a research idea: combining CCA and CCW with PC2 cloud fraction and condensate before computing subcolumns seems like it would smear out differences between convective and stratiform cloud properties. Wouldn't it be better

to just reserve a subcolumn for convection?

10. section 2.8: if your PBL scheme is really a turbulence scheme, why not call the section and the parameterization "turbulence scheme" instead of PBL scheme?

11. p. 9 lines 16-22: I found this explanation to be unclear. Are you saying you're lifting parcels moist adiabatically by 1 grid cell and computing the TKE consumption required by that motion?

12. p. 9, line 28: "stable stability dependence": is stable stability redundant?

13. p. 9, line 29: what is the "sharp" function?

14. p. 10, line 9: "surface layer is conditionally unstable"?

15. p. 11, lines 28-end: I don't understand how you use the canopy model to simulate lake and urban surfaces. Reword?

16. p. 12, line 8: it seems like you'd only want to nudge to climatology for weather simulations, not climate change simulations? Also, how is albedo "further modified in the presence of snow"?

17. p. 12, line 10: does the canopy affect LW emission?

18. p. 12, line 22: how is "Excess water" defined?

19. p. 14 table and elsewhere: When you say "system dependent", I think of computer system (e.g. a particular intel KNL machine) but I think you mean "model configuration".

20. p. 16, line 10: This is a research idea: could you make solver tolerance a function of vertical level to get the best of both computational efficiency and accuracy where needed?

21. p. 17 line 2-3: I don't think there should be a paragraph break here.

22. p. 20 line 1: "by either" sounds awkward.

23. p. 20, lines 1-3: how is the fraction of autoconverting cloud different than the cloud fraction? It also seems awkward that you avoid advecting rain by doing something special when rain advects into a grid cell... how is rain advecting when you're avoiding advecting it? I think you mean you avoid horizontal advection by handling rain sedimenting into cells below by setting rain area fraction equal to the cloud fraction above it.

24. p. 21, line 19: "are scaled down before being combined..." How are they scaled down?

25. p. 22, line 29 or so: what fraction of the time is RHcrit at its max or min value?

26. p. 25 top: what is the % reduction in energy fixer magnitude from GA ticket #87?

27. p. 25 line 25: is enhanced thinning of stratocumulus a good thing in your model? In my experience models tend to thin Sc too much during the day.

28. p. 26-27: I must admit I still don't understand how convection interacts with the turbulence scheme after reading this section. I think turbulence acts as the trigger for convection and when convection acts the turbulence scheme ignores condensational heating (i.e. is a "dry turbulence" scheme) in order to avoid double-counting moist plumes? In any case, this could be explained better.

29. p. 30: the relationship between CAPE timescale and resolution would make a good topic for a standalone paper. What is written here is fine, but more details (in a follow-on paper) would be welcome.

30. p. 42 line 25: "own/independent analyses" this sounds awkward. Is a word missing?

31. section 3.11: I'm surprised there's no mention of land model tuning? Are there no tunable parameters in the land model? If so, you should mention that.

32. p. 45, line 14: Isn't it customary to include a table where it is first mentioned? Why

is Table 8 left to the end of the paper?

33. p. 45 line 15: in section 4.1-4.3 you only talk about N96 and N216. You should mention both of these and NOT mention N768 until section 4.4. Also, it would be nice to get the rough dx values in km here for N96 and N216.

34. p. 47 line 2: I think you mean "returning" rather than "retuning"

35. section 4.1-4.3: how long are the simulations you discuss in the climate section. The end of section 4.3 mentions 27 years. . . if this is the case, it should be mentioned at the beginning of section 4.1 because otherwise the reader is left wondering about statistical significance of the results you show. . .

36. Fig. 12: it's really hard to tell orange from red dots. Perhaps replace orange with yellow dots?

37. Fig 16 top left panel: why have OLR color levels start at 0 W/m2 so the plot basically just looks like a red box? I think some other figures could benefit from this change too.

38. Figure 21: 3rd from last line of caption: I think you mean blue instead of purple?

39. p. 57 line 7: I think you mean "OUR own analysis"?

40. section 4: I'm surprised you don't evaluate the land model skill at all. I know decent land model behavior is needed for reasonable atm metrics, but shouldn't you also evaluation soil moisture and temperature at each layer, river runoff, vegetation albedo, etc?

41. p. 59 end of line 9: "prep.b)" typo?

42. p. 60: differences between GA7.1 and 7.0 bullets 2-4: does CMIP6 specify these input datasets? More generally, you don't really say anywhere whether or not you follow the CMIP6 specifications for input dat

---

## Referee Comment (RC2) · Anonymous Referee #2 · 11 Jan 2018

The paper provides a lucid description of the development of the most recent versions of the Met Office global atmosphere and land models. The structure, an overview of changes from earlier versions with motivations, followed by implementation details, is an effective means of organizing this complex discussion. The inclusion of both climate and prediction characteristics presents a comprehensive view of model behavior. The discussion is naturally in the context of earlier Met Office models, a knowledge of which will increase readers' appreciation of the material greatly, but sufficient information is generally available (though note the issues raised in RC1) to get the gist of the scientific developments for readers with less background in earlier model versions. Specific

points for revision or consideration follow.

1. Given that GA7.1 will be be the UK contribution to CMIP6, ideally GA7.1 would have been the model described with changes required to reduce the magnitude of the aerosol effective radiative forcing (ERF) integrated into the main deveopment sequence. The point when it was realized the GA7.0 ERF was inconsistent with observed climate change may have not made this practical, and the authors note that many of the GA7.0 results presented change either little or are improved in GA7.1. Still, if this is so, it would have been better to have seen the GA7.1 results consistently through the paper.

2. Section 5 omits some very important information which is essential, and its inclusion is recommended as a major revision:

(a) The paper reports ERFs for GA7.0 but not GA7.1. It is the latter values that are critically important to understanding GA7.1. The GA7.1 aerosol and total ERFs must be reported.

(b) Provide references to justify the 70% increase in marine DMS emissions (p. 60, l. 5).

(c) Provide some indication of the relative importance of the changes designed to reduce the magnitude of the aerosol ERF (p. 60) in doing so.

3. Even if the changes discussed on p. 60, which can be viewed as ERF tunings, are empirically and physically justified, an important point emerges. Even a skillfully designed model like GA7.0 can produce a manifestly unrealistic behavior (global cooling,

instead of warming, during the 20th century). It is not the only model for which this is an issue. Four of the six U.S. climate models discussed in Schmidt et al. (2017) use aerosol indirect effect in their tuning. To this reviewer, this should impel research to understand more robustly the magnitude of the aerosol ERF. Further comments on the implications of the need to develop GA7.1 distinctly from GA7.0 should be included in the paper.

4. Section 3.11 (p. 42, ll. 6-8) notes the importance of tuning with principles for selection of parameters based on observations, other models, and constraints based on theory or observations. Yet, some parameterizations seem designed primarily to reduce model bias with structural designs difficult to justify otherwise. The particular example here is the parameterization for cirrus spreading (p. 23). The parameterization form (4) is not evidently related to the physics of shear-generated fall streaks but rather to preventing unrealistic cloud fractions in the model.

5. Figs. 7, 13, 14, 15, 16, 17, 18, 24, and 25: Include summary statistics (mean bias, RMSE, correlation coefficient) to compare model fields with observations.

6. p. 7, ll. 5-10: Indicate the number of moments in the microphysics parameterization.

7. p. 12, ll. 7-8: Nudging toward climatology presumably does not take place with interactive vegetation. This is somewhat unclear.

8. p. 16, ll. 23-26: Quantify the magnitude of the radiation improvements.

9. Numerous important references are described as in preparation. Unless these have at least reached the submitted stage when the final version of this manuscript is ready,

[Figure]

I suggest removing them.

p. 32, l. 22-23: N50 concentrations are described as in reasonable agreement with modeled results, but modeled concentrations are as much as an order of magnitude lower than observed as observed N50 approaches 10,000 cm$^{-3}$.

p. 45, ll. 5-10: Tunings to improve the coupled simulation are described. How much do these tunings change the global, annual-mean top-of-atmosphere OLR, SW, and net radiative fluxes if imposed in the uncoupled model?

p. 8, l. 3: Cores do not detrain into *variables.*

p. 22, l. 20: "Whilst" $->$ "whilst"

p. 23, l. 6: Should "GA4" be "GA6"?

p. 35, l. 16: "m$^{-2}$" $->$ "m$^{-3}$"

p. 43, l. 12: "describe" $->$ "described"

---

## Author Comment (AC2) · 15 Feb 2018

**1   Response to general comments**

We would like to thank the reviewer for their thorough and complementary review. The list of minor comments is appreciated as addressing these has led to improvements to the paper. We respond to these in turn below.

[Figure]

**2   Responses to specific questions**

1. *"The thing that bugged me most in this paper was that the model timestep and target horizontal resolutions were never described ..."*

   Because the GA configuration is designed for use across a range of horizontal resolutions, we do not really have the concept of a "target resolution". Instead, we expect the documentation for each system using a GA configuration to specify their own target resolutions. The fact that the time step information is only included as supplementary information, however, is an oversight; we've added this in a table in Sect. 2.2.

2. *"p2 line 1-3: citations in this sentence seem like they're for v7 of the model ..."*

   The Unified Model is a very general code base that can be used is a large number of ways; the Global Atmosphere configuration described in the paper is a single science configuration of this model. The citations in these lines are for general descriptions of the models, rather than the configurations described in this paper. We have tried a number of different ways of citing the general model descriptions and the structure used in lines 1–3 is the best we can come up with.

3. *"section 2.1: I'm curious why UKMO continues to use a lat/lon grid instead of a cubed-sphere grid. Also, are polar filters used?"*

   We originally chose a lat/lon grid for the UM (rather than taking, say, a spectral approach) because this allowed us to use the same code base for a global models and limited area models. The move to semi-Lagrangian advection in the early 2000s meant that the cost of these grids was not too great and a more recent upgrade to use the ENDGame dynamical core has improved the scalability to be suitable for current HPC architectures. With the move to increased parallelism, the UM will suffer from scalability issues due to the incredibly fine resolution as

one approaches the pole and it has been known for some time that this approach is not suitable for Exascale HPCs. The next-generation model (due to be operational in the mid-2020s) will run on a cubed-sphere grid, but with the aim of the majority of code being agnostic to the grid, which will make it easier to swap to alternative grid structures later on, if required.

Since the introduction of ENDGame in GA6, the GA configurations of the UM have run without using a polar filter. Whilst this improves the model's scalability, it is not without problems as it does expose the noise at the pole discussed on p16 of the paper and alleviated by ticket #135. We have not ruled out the reintroduction of some very limited polar filtering in future releases, although we do not want to do this without first understanding the source of the noise in the wind field.

4. *"I'm curious about model performance. How many cores do you typically use and what is your typical throughput on the machine you use most?"*

As the reviewer suggests, we do see this as out of scope of the paper. However, here are some very rough numbers for completeness. An N96 atmosphere/land-only climate simulation run on about 450 cores of our Cray XC40 HPC completes just over 3 years of simulation in 1 day (ignoring any queuing time between monthly resubmissions). At the other end of the spectrum, an operational N1280 NWP simulation uses 530 nodes ($\approx$19,000 cores) and completes a 7+ day forecast in 40-45 minutes. In research mode, we tend to run on fewer nodes, which increases runtime, but allows us to run a larger number of simulations in parallel.

5. *"section 2.2: Using parallel splitting creates opportunities for water and aerosol species to go negative..."*

Although this is possible in principle for moist species, there are checks within the code to ensure not only positivity, but internal consistency between differ-

ent moist/cloud prognostics. No aerosol processing takes place with the parallel physics.

6. *"page 4 line 13: "When using the prognostic aerosol scheme, this is included. . ." is awkward"*

   Agreed. We have made the change suggested.

7. *"page 7 line 12: "We also make use of. . ." sentence is awkward."*

   Agreed. We have restructured the sentence and split it in two.

8. *"It seems odd to talk about Large-scale precipitation (sect 2.4) before large-scale cloud (sect 2.5)"*

   Yes, we can see the reviewer's point. However, the ordering of physics schemes in Sect. 2 is based on their order in the UM time step. In the UM, the LSP scheme is run from the start of the time step, whilst the PC2 cloud scheme is distributed through the time step, so we have decided to include this afterwards.

9. *". . . Wouldn't it be better to just reserve a subcolumn for convection?"*

   Yes, we could allocate one column for convective cloud amount. In the current scheme, this does implicitly set the convective cloud fraction as 1/(number of sub-columns), so we would need some sensible way of working out its weight. Related to this, but of equal priority in the current scheme, is further investigation of how best to determine the amount of convective cloud water.

10. *"section 2.8: if your PBL scheme is really a turbulence scheme, why not call the section and the parameterization "turbulence scheme" instead of PBL scheme?"*

    Most of this turbulence diagnosed by the scheme is in the boundary layer, so this is still its main role. In addition, there is also plenty of turbulence in and

around cumulus clouds which this scheme does not handle and so we find this a helpful distinction. Finally, we want the paper to be consistent with our internal documentation and code naming, in which this is labelled the BL scheme.

11. *"p. 9 lines 16-22: I found this explanation to be unclear."*

We have updated the text to make this clearer: *"The existence and depth of unstable layers is diagnosed initially by two moist adiabatic parcels, one released from the surface, the other from cloud-top. The top of the K profile for surface sources and the base of that for cloud-top sources are then adjusted to ensure that, from the resultant buoyancy flux, the magnitude of the buoyancy consumption of turbulence kinetic energy is limited to a specified fraction of buoyancy production, integrated across the boundary layer."*

12. *"p. 9, line 28: . . . is stable stability redundant?"*

By stable stability dependence, we mean the stability function used in stable boundary layers. We have altered the text to make this clearer.

13. *"what is the "sharp" function?"*

We have added a reference in which this is described.

14. *"p. 10, line 9: "surface layer is conditionally unstable"?"*

By unstable we mean where the surface buoyancy flux is positive and so have now written that instead.

15. *"p. 11, lines 28-end: I don't understand how you use the canopy model to simulate lake and urban surfaces."*

Here, we take advantage of the capability of the "vegetation canopy" code to model a layer above the top soil layer with an additional heat capacity that is

coupled radiatively to the soil. All other aspects of the vegetation canopy code is switched off on these surface tiles. This is described in the references included."

16. *"p. 12, line 8: it seems like you'd only want to nudge to climatology for weather simulations"*

Agreed. The lack of clarity on this point was also picked up by the second reviewer in reviewer comments RC2. We have added a comment to clarify this.

17. *"p. 12, line 10: does the canopy affect LW emission?"*

No, the emission is based on the surface temperature and the emissivity of the surface type.

18. *"18. p. 12, line 22: how is "Excess water" defined?"*

For clarity, we have changed this to "Outflow at inland basin points with saturated soils is distributed...". This is required, as otherwise the outflow would go into runoff, which would go into the rivers and then turn back into outflow.

19. *"When you say "system dependent", I think of computer system (e.g. a particular intel KNL machine) but I think you mean "model configuration"."*

As with the response to Question 1, this is complicated because the GA configuration is not a model in itself, but a specific configuration of a more flexible model. For this reason, we have come up with the definitions below:
**Model** (e.g. The Unified Model): The underlying code base used for modelling one component (i.e. the atmosphere in the case of the UM). Major iterations of this code base are described as model "versions".
**Science configuration:** A set of scientific options and settings used to "configure" a model simulation. The Global Atmosphere configuration is essentially a set of options designed for use at resolutions requiring parametrised convection.

Major iterations of these configurations are described as "releases" (i.e. GA7 is the 7th GA release, not the 7th GA version).

**System:** The use of the word system is quite common in describing short-range prediction tools. e.g. the Met Office global NWP system uses a GA/GL configuration of the UM/JULES for its deterministic forecast. Similarly, it is quite common to describe an ensemble prediction system or a seasonal forecasting system. It is less common to describe a production climate model as a system, but by analogy, it is, with its system design essentially dictated by the experimental design of the activity for which it is being used.

20. *". . . could you make solver tolerance a function of vertical level. . . ?"*

This would be more difficult in the UM than in some other global models, because its dynamical core is non-hydrostatic. We have also had discussions about making the solver tolerance a more localised function of latitude, although this would also lead to load balancing issues. In line to our response about polar filtering, we would still like to understand the root cause of the noise in the wind fields at the poles.

21. *"p. 17 line 2-3: I don't think there should be a paragraph break here."*

Agreed.

22. *"p. 20 line 1: "by either" sounds awkward."*

23. *"how is the fraction of autoconverting cloud different than the cloud fraction?"*

*"It also seems awkward that you avoid advecting rain by doing something special when rain advects into a grid cell. . ."*

As a joint response to points 22–23, we have rewritten this paragraph, which hopefully makes this clearer. The quantity that we don't want to advect is the

"rain fraction" rather than the rain mass mixing ratio, which we do advect. The rewritten paragraph is: *"We also improve the parametrisation of sub-grid rain fraction, i.e. the fraction of the grid box in which rain is held. When rain is created by autoconversion, this is set to be equal to the grid box liquid cloud fraction and when rain is created by melting snow this is set to be equal to the grid box ice cloud fraction. When rain is advected horizontally into a previously rain-free column, the rain fraction is set to the fraction of cloud directly above it. This avoids the requirement to advect a rain fraction in addition to the rain amount."*

24. *"p. 21, line 19: "are scaled down before being combined. . ." How are they scaled down?"*

    Now clarified in the text.

25. *"p. 22, line 29 or so: what fraction of the time is RHcrit at its max or min value?"*

    We don't have a measure of how often this is at its max or min value within a model run, but we have previously looked at what the model would have diagnosed using offline calculations, which suggests that these max and min values will be being activated, but not too often. Another piece of evidence that the impact of these on the model cloud is not large comes from some diagnostics we have looked at (but not shown here) showing histograms of cloud amount vs RH. At GA6, these show clear clustering on the old RHcrit curve, even though RHcrit is only being used for initiation from zero cloud and erosion from a fully cloudy state. In GA7 simulations there is no evidence of any unsmooth peaks, suggesting that the impact of the parametrisation is itself relatively smooth.

26. *"p. 25 top: what is the % reduction in energy fixer magnitude from GA ticket #87?"*

    This is discussed in the results section 4.3.

27. *"p. 25 line 25: is enhanced thinning of stratocumulus a good thing in your model?"*
    You raise a good point, that here, the word "enhanced" could be misread as
    "improved", so we will replace this with "additional". However, in GA6, the diurnal
    cycle of our SCu LWP was previously too weak and got neither thin enough by day
    nor thick enough by night. This change (GA:#13) and the warm rain microphysics
    change (GA:#52) do both act to improve this.

28. *"p. 26-27: I must admit I still don't understand how convection interacts with the
    turbulence scheme after reading this section . . . "*

    The reviewer's understanding of this (based on their comments) is correct, but
    we have updated the text to make this clearer.

29. *"p. 30: the relationship between CAPE timescale and resolution would make a
    good topic for a standalone paper"*

    We have decided not to write up the relationship between the CAPE timescale
    and resolution, as any closure based on this assumption will break down when
    going to the convective "grey zone". The development of a grey zone convection
    scheme for the Unified Model is currently an area of active research, which will
    itself require a new approach to the convective closure.

30. *"p. 42 line 25: "own/independent analyses" this sounds awkward. "*

    This was originally trying to clarify that we verify both against the model's own
    analyses (as generated through a cycling data assimilation system) and independ-
    ent analyses. As it's difficult to express this elegantly, we have simply replaced
    this with "analyses".

31. *"I'm surprised there's no mention of land model tuning?"*

    The tuning section only describes tuning performed between GA6.0/GL6.0 and
    GA7.0/GL7.0, which did not include any *new* tuning of the land surface model.

32. *"p. 45, line 14: Isn't it customary to include a table where it is first mentioned?"*

    The main reason for including this table is to meet the requirements of the journal for "Details of the simulations performed", so it is included at that point. Given that this table is there, however, it is worth referencing at the start of the results section. We have clarified this a bit in the text.

33. *"p. 45 line 15: in section 4.1-4.3 you only talk about N96 and N216. You should mention both of these and NOT mention N768 until section 4.4. Also, it would be nice to get the rough dx values in km here for N96 and N216."*

    Agreed.

34. *"p. 47 line 2: I think you mean "returning" rather than "retuning""*

    Yes, thank you.

35. *"section 4.1-4.3: how long are the simulations you discuss in the climate section?"*

    We have clarified that these are 27 year simulations at the start of the section.

36. *"Fig. 12: it's really hard to tell orange from red dots."*

    The majority of figures in the paper have been created using a standard 8 class paired colour palette, so I have replaced the orange with the "light orange" colour in that palette.

37. *"Fig 16 top left panel: why have OLR color levels start at 0 W/m2 so the plot basically just looks like a red box? I think some other figures could benefit from this change too."*

    The aim of this is for Figs 15, 16 and 25 to use the same colour scales (as part of a budget). As the main points of discussion are in the differences, we would rather leave this as it is, unless the reviewer/editor feels very strongly about this.

38. *"Figure 21: 3rd from last line of caption: I think you mean blue instead of purple?"*

Comment from the lead author: This comment made me chuckle because whilst this colour was deliberately chosen to have the same tone as the green arrows, exactly what colour it actually is has caused great debate! A quick straw poll of those around me has led to "purple", "blue" and "purple/blue", so I've decided to leave this as it is.

39. *"p. 57 line 7: I think you mean "OUR own analysis"?"*

Verifying model changes against an analysis created using that model is tricky to describe, because the change in the model leads to changes in the analysis. For this reason, "own analysis" is a bit of a standard phrase, although I agree it does not read well. We've replaced this with "its own analysis".

40. *"I'm surprised you don't evaluate the land model skill at all."*

We routinely perform a rudimentary analysis of the land surface model, but improved assessment of the GL configuration is something in our future development plans. This should include routinely assessing the performance of the land surface component in JULES-only simulations as well as coupled UM/JULES ones.

41. *"p. 59 end of line 9: "prep.b)" typo?"*

This is standard BibTeX behaviour because "in prep." is used as the year. We need to wait for a doi for the "a" and "b" papers before final resubmission (see our response to reviewer's comments RC2), so this will go away then.

42. *"p. 60: differences between GA7.1 and 7.0 bullets 2-4: does CMIP6 specify these input datasets? More generally, you don't really say anywhere whether or not you follow the CMIP6 specifications for input data."*

[Figure]

No, CMIP6 specifies the emissions of the aerosol precursors, not the optical properties of the aerosols themselves. Whilst DMS emissions over land are included, marine emissions from natural sources are dependent on the model. The emissions used in our assessment simulations are outlined in Table 1, which are mostly based on CMIP5; the CMIP6 protocols had not been published at the point that these simulations were first performed. The performance of the UKESM1 and HadGEM3-GC3.1 climate models with CMIP6 emissions will be the subject of subsequent publications.
* * *

---

## Author Comment (AC3) · 15 Feb 2018

**1   Response to general comments**

We thank the referee for reviewing our manuscript and for their insightful comments. The most significant point raised is the lack of detail in the description of GA7.1. The authors fully recognise the importance of these developments, and for this reason, a detailed description of the rationale behind these is being documented in full in an additional paper (already referenced in the paper under review as Mulcahy et al. (in prep.b) – hereafter Mulcahy et al.). Mulcahy et al. will address comments 2(a-c), documenting

more fully the aerosol ERFs of both GA7.0 and GA7.1. It will also document in full the relative importance of each change in terms of impact on ERF (addressing Comments 2(b and c)). The authors feel that to address these issues here would go beyond the scope of this paper and would significantly detract from the importance of that highly relevant work.

The authors propose to raise the awareness of the Mulcahy et al. paper at the start Sect. 5 by including the following sentence: *". . . whilst maintaining a present-day simulation similar to that from GA7.0. The aerosol ERF and subsequent development of the GA7.1 configuration as well as the scientific justification for those developments are fully documented in Mulcahy et al. (in prep.b)."* In response to comment 9, however, this does mean that we will need to request that the final publication of this paper is delayed until a doi is available for Mulcahy et al.

We hope that this addresses the main concerns of the referee; we address the individual points raised in more detail below.

**2   Responses to specific questions**

1. *"Given that GA7.1 will be be the UK contribution to CMIP6, ideally GA7.1 would have been the model described . . . it would have been better to have seen the GA7.1 results consistently through the paper."*

   Whilst GA7.1 will be the GA configuration used in the UK submissions to CMIP6, we try to make clear in the paper that this is still seen as a "branch" to the GA "trunk" and that the latest release of that trunk is still GA7.0. Aside from CMIP6, GA7.0 will still be used as the basis for a number of applications including operational NWP and the GloSea seasonal forecasting system. Furthermore, GA7.0 and not GA7.1 forms the baseline for GA8 development and the eventual documentation of a GA8.0 configuration will be made by comparison with GA7.0.

2. (a)*"The paper reports ERFs for GA7.0 but not GA7.1."*

As discussed in the response to general comments, this is out of scope of the current publication, but will be documented in full in Mulcahy et al.

(b)*"Provide references to justify the 70% increase in marine DMS emissions"*

The increase in marine DMS emissions is a first attempt to represent a marine organic aerosol source in the model, guided by the observations in McCoy et al. (2015) and work of O'Dowd et al. (2004). Again, this work and the motivation for this change is documented in detail in Mulcahy et al.

(c)*"Provide some indication of the relative importance of the changes designed to reduce the magnitude of the aerosol ERF"*

As discussed above, a quantitative response to this will be included in Mulcahy et al. However, for a more qualitative response, we are also happy to list the changes in Sect. 5 in order of importance. The list will therefore read:

1. Liu cloud droplet spectral dispersion
2. Aerosol absorption update
3. RADAER lookup changes
4. Scaling of DMS
5. UKCA Activate
6. Lana
7. Retuning

3. *"Even if the changes discussed on p. 60, which can be viewed as ERF tunings, are empirically and physically justified, an important point emerges. . . . To this reviewer, this should impel research to understand more robustly the magnitude of the aerosol ERF."*

Again, we agree with the referee's comments that this highlights an important point of general interest to the climate modelling community. Further research is required to more robustly understand the magnitude of the aerosol ERF across global climate models. However, once again, we feel these discussions go well beyond the scope of the current paper. The implications of the need to develop a GA7.1 configuation will be discussed in Mulcahy et al. and the larger scientific questions raised as a result of this research are already the focus of further investigations with the UKESM and HadGEM3-GC3.1 models.

4. *"Some parameterizations seem designed primarily to reduce model bias with structural designs difficult to justify otherwise. The particular example here is the parameterization for cirrus spreading (p. 23)."*

Whilst this does sometimes occur in the development of new parametrisations, it is something that we try hard to avoid, to be clear about when this has been done and to address in the longer term by replacing these with more physically motivated approaches.

The cirrus spreading term highlighted by the reviewer is a good example of this. It was first introduced in GA4.0 as a late tuning, with an acknowledgement from the developers that this was poorly motivated. Because of this, when the package of cloud, radiation and microphysics changes for GA7.0 was first proposed, the parametrisation developers proposed removing this altogether by setting the spreading rate to a vanishingly small value. In the later development of more complete GA7.0 test configurations, comparisons with observations showed that in general this was a good thing, but that reducing it to the extent originally proposed pushed the biases too far. For this reason, we have had to leave this in the final configuration, albeit with a significantly reduced rate compared to GA6.0, but have acknowledged in the documentation that this is a tuning and that there is still a desire to eventually remove this term altogether.

5. *"Figs. 7, 13, 14, 15, 16, 17, 18, 24, and 25: Include summary statistics"*

   We will consider this ahead of producing plots for the final publication.

6. *"Indicate the number of moments in the microphysics parameterization."*

   Our current microphysics scheme is single moment, holding prognostic variables for the mass of a number of independent species (i.e. cloud liquid water, cloud ice water and liquid rain). We have altered the text to make this explicit in the description in Sect. 2.4.

7. *"Nudging toward climatology presumably does not take place with interactive vegetation. This is somewhat unclear."*

   Agreed. The lack of clarity on this point was also picked up by the first reviewer in reviewer comments RC1. We have added a comment to clarify this.

8. *"Quantify the magnitude of the radiation improvements."*

   Some evidence for this will come through the statistics added in response to Question 5. In line with our response to Question 1, however, we would like to reiterate that simulations presented here are designed to be indicative of the performance of the configuration in general and not to represent its final performance in any particular target system or application. For this reason, we believe that a detailed quantitative discussion of the magnitude of improvements in one particular simulation is beyond the scope of this paper; instead this is for discussion in subsequent papers describing particular applications.

9. *"Numerous important references are described as in preparation."*

   We agree that a final resubmission should not contain "in prep" references. The most important of these, which is needed to address questions 1–3 of this review,

is Mulcahy et. al (in prep.b). Good progress is being made on that paper, but we will make a request to the editor of this paper that its publication is held back until a doi is available for Mulcahy et al.

10. *"N50 concentrations are described as in reasonable agreement with modeled results..."*

The word "reasonable" was used here because whilst we agree that it would be an overstatement to say that this "agrees well" or "is in good agreement", we do believe that the agreement is reasonable given the current expectations for physical climate models and the limitations of both our modelling systems and the observational datasets. In the paper, we highlight good performance in the clean air regions, but acknowledge the underestimation in the polluted regions of North America, Europe and Asia. The observations used in this comparison have their limitations, as these are from specific campaigns at particular points in time, compared to a free-running 20 year model climatology. In many instances, these campaigns deliberately track the polluted plumes and so the observations are likely to be weighted towards highly polluted events (e.g. see Reddington et al., 2017; Schutgens et al., 2017). These papers highlight the inherent uncertainty in model-to-obs comparisons due to spatio-temporal sampling and note that this can lead to an overestimation of high pollution episodes and underestimation of clean episodes in the observations.

We propose adding the following sentence to the paper to address this: *"It is a point of ongoing discussion whether the current approach of targeting high-pollution events with observational campaigns leads to biases in comparisons of models to observations for these events (Reddington et al., 2017; Schutgens et al., 2017)."*

11. *"Tunings to improve the coupled simulation are described. How much do these tunings change the global, annual-mean top-of-atmosphere OLR, SW, and net*

*radiative fluxes if imposed in the uncoupled model?"*

As discussed in Sect. 3.11, the most significant tuning aimed at improving the simulation in the coupled model was the tuning of the r_det parameter. This was primarily motivated by its impact on local SSTs rather than any global mean parameters, so the impact on global OLR, SW and net radiative fluxes in the uncoupled simulations was less than $0.1\,\mathrm{Wm}^{-2}$.

12. Additional suggested corrections:

    We agree with all 5 of these suggestions and have updated the manuscript appropriately.

**References**

McCoy, D. T., Burrows, S. M., Wood, R., Grosvenor, D. P., Elliott, S. M., Ma, P.-L., Rasch, P. J., and Hartmann, D. L.: Natural aerosols explain seasonal and spatial patterns of southern ocean cloud albedo, Science Advances, 1, doi:10.1126/sciadv.1500157, 2015.

O'Dowd, C. D., Facchini, M. C., Cavalli, F., Ceburnis, D., Mircea, M., Decesari, S., Fuzzi, S., Yoon, Y. J., and Putaud, J.-P.: Biogenically driven organic contribution to marine aerosol, Nature, 431, 676, doi:10.1038/nature02959, 2004.

Reddington, C. L., Carslaw, K. S., Stier, P., Schutgens, N., Coe, H., Liu, D., Allan, J., Browse, J., Pringle, K. J., Lee, L. A., Yoshioka, M., Johnson, J. S., Regayre, L. A., Spracklen, D. V., Mann, G. W., Clarke, A., Hermann, M., Henning, S., Wex, H., Kristensen, T. B., Leaitch, W. R., Pöschl, U., Rose, D., Andreae, M. O., Schmale, J., Kondo, Y., Oshima, N., Schwarz, J. P., Nenes, A., Anderson, B., Roberts, G. C., Snider, J. R., Leck, C., Quinn, P. K., Chi, X., Ding, A., Jimenez, J. L., and Zhang, Q.: The Global Aerosol Synthesis and Science Project (GASSP): Measurements and modeling to reduce uncertainty, BAMS, 98, 1857–1877, doi:10.1175/BAMS-D-15-00317.1, 2017.

Schutgens, N., Tsyro, S., Gryspeerdt, E., Goto, D., Weigum, N., Schulz, M., and Stier, P.: On the spatio-temporal representativeness of observations, Atmos. Chem. Phys., 17, 9761–9780, doi:10.5194/acp-17-9761-2017, 2017.